# Engineered heart muscle allografts for heart repair in primates and humans

Ahmad-Fawad Jebran[1,2,29], Tim Seidler[2,3,4,29], Malte Tiburcy[2,5,29], Maria Daskalaki[2,6], Ingo Kutschka[1,2], Buntaro Fujita[7,8], Stephan Ensminger[7,8], Felix Bremmer[2,9], Amir Moussavi[2,10], Huaxiao Yang[11,12], Xulei Qin[11,12], Sophie Mißbach[2,6,13], Charis Drummer[2,6], Hassina Baraki[1,2], Susann Boretius[2,10], Christopher Hasenauer[14], Tobias Nette[14], Johannes Kowallick[2,14], Christian O. Ritter[2,14], Joachim Lotz[2,14], Michael Didié[2,3], Mathias Mietsch[2,13], Tim Meyer[2,5], George Kensah[1,2], Dennis Krüger[15], Md Sadman Sakib[15], Lalit Kaurani[15], Andre Fischer[2,15,16,17], Ralf Dressel[2,18], Ignacio Rodriguez-Polo[2,6], Michael Stauske[2,6], Sebastian Diecke[19,20], Kerstin Maetz-Rensing[21], Eva Gruber-Dujardin[21], Martina Bleyer[21], Beatrix Petersen[2,22], Christian Roos[22], Liye Zhang[22], Lutz Walter[2,22], Silke Kaulfuß[23], Gökhan Yigit[2,23], Bernd Wollnik[2,17,23], Elif Levent[2,5], Berit Roshani[24], Christiane Stahl-Henning[24], Philipp Ströbel[9], Tobias Legler[2,25], Joachim Riggert[2,25], Kristian Hellenkamp[3], Jens-Uwe Voigt[26], Gerd Hasenfuß[2,3], Rabea Hinkel[2,13], Joseph C. Wu[11,12], Rüdiger Behr[2,6] & Wolfram-Hubertus Zimmermann[2,5,17,27,28 ✉]

Cardiomyocytes can be implanted to remuscularize the failing heart[1–7]. Challenges include sufficient cardiomyocyte retention for a sustainable therapeutic impact without intolerable side effects, such as arrhythmia and tumour growth. We investigated the hypothesis that epicardial engineered heart muscle (EHM) allografts from induced pluripotent stem cell-derived cardiomyocytes and stromal cells structurally and functionally remuscularize the chronically failing heart without limiting side effects in rhesus macaques. After confirmation of in vitro and in vivo (nude rat model) equivalence of the newly developed rhesus macaque EHM model with a previously established Good Manufacturing Practice-compatible human EHM formulation[8], long-term retention (up to 6 months) and dose-dependent enhancement of the target heart wall by EHM grafts constructed from 40 to 200 million cardiomyocytes/stromal cells were demonstrated in macaques with and without myocardial infarction-induced heart failure. In the heart failure model, evidence for EHM allograft-enhanced target heart wall contractility and ejection fraction, which are measures for local and global heart support, was obtained. Histopathological and gadolinium-based perfusion magnetic resonance imaging analyses confirmed cell retention and functional vascularization. Arrhythmia and tumour growth were not observed. The obtained feasibility, safety and efficacy data provided the pivotal underpinnings for the approval of a first-in-human clinical trial on tissue-engineered heart repair. Our clinical data confirmed remuscularization by EHM implantation in a patient with advanced heart failure.

Myocardial remuscularization can be achieved by cardiomyocyte (CM) implantation[1–3], with increasingly strong evidence from large animal studies[4–7,9,10]. Similar data were obtained in rodent and rabbit models with tissue-engineered allografts and xenografts[11–18]. Small animal studies are important for early proof of concept, but they may not reliably predict outcomes in larger animals and ultimately in patients with heart failure, particularly when conducted as xenograft studies[4–7,19]. In addition, xenograft studies are compromised by strong immune responses, and to our knowledge, there is no evidence for long-term engraftment of CM xenografts beyond 3 months in immunocompetent animals under clinically acceptable immune suppression for patients with heart failure.

The observed therapeutic effects in xenograft models, but also in allograft adult stem cell studies with no evidence for remuscularization and limited retention, suggest that the outcomes in these studies are at least partially mediated by immune responses[20] or paracrine mechanisms[21,22]. Our own data, on human xenografts in nude rats with notable macrophage infiltration and similarly improved outcome after implantation of viable and lethally irradiated engineered heart muscle (EHM), are in agreement with these muscularization-independent mechanisms[18]. Importantly, allograft studies with effective immuno-suppression demonstrated that CM-containing EHM is superior to non-myocyte or non-viable grafts, suggesting functional remuscularization as the main mode of action[11,12].

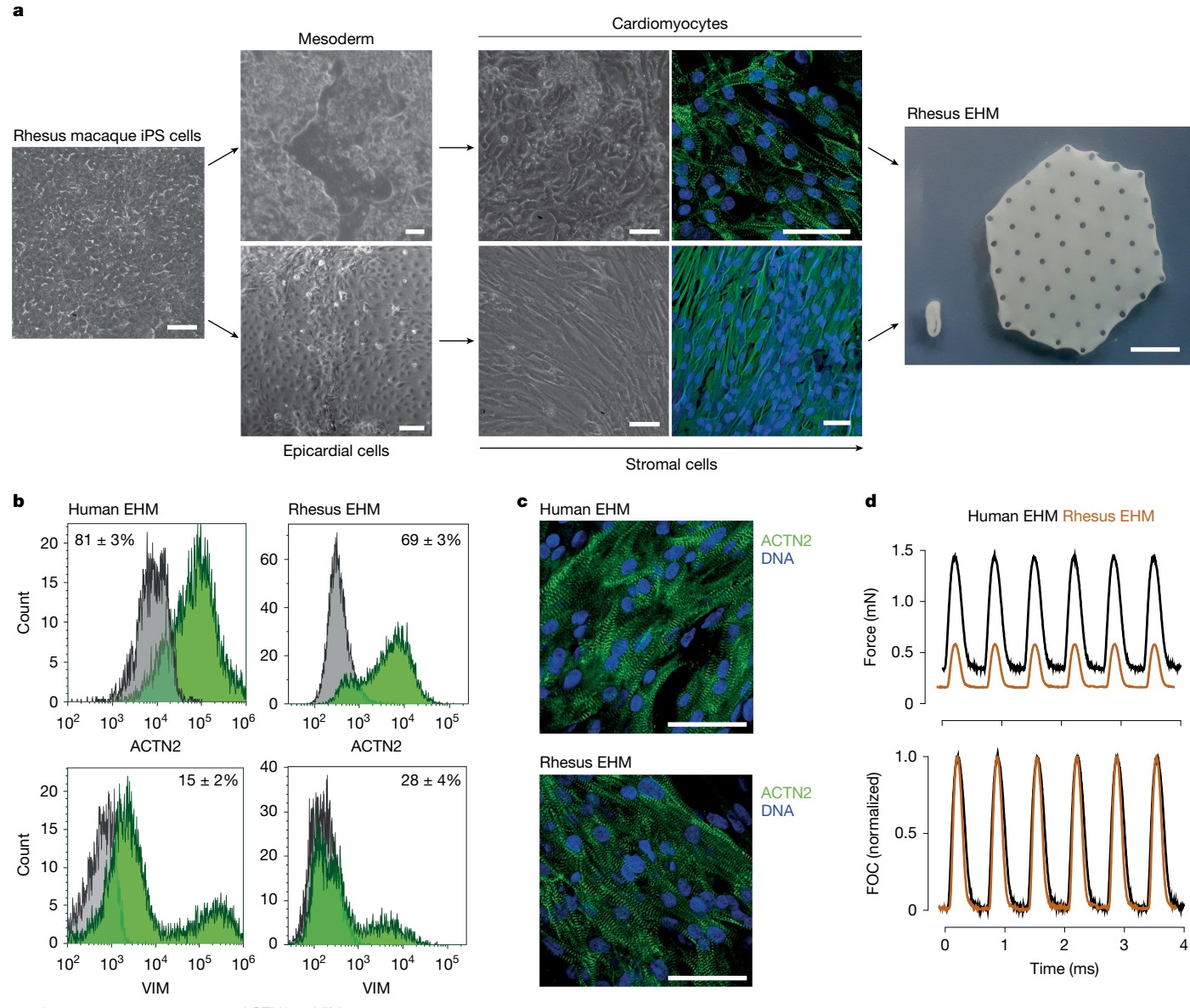

**Fig. 1 | Rhesus macaque EHM formulation and characterization.**
**a**, Illustration of the CM and StC differentiation process starting from undifferentiated rhesus macaque iPS cells (left, bright field image; middle upper, bright field images 10 days (left) and 22 days (right) after mesoderm induction with immunofluorescence staining for sarcomeric actinin (ACTN2 in green; nuclei in blue); middle bottom, bright field images 20 days after mesoderm and subsequent epicardial induction (left) as well as after additional 20 days (right) of epithelial-to-mesenchymal transition induction and expansion with immunofluorescence staining for vimentin (VIM in green; nuclei in blue). Subsequent to their directed differentiation, CMs and StCs were embedded in bovine collagen type I hydrogels, cast into custom-made moulds and cultured for 28 days to obtain EHM (right) in a ring format (450-µl reconstitution volume)

for quality control by contractility measurements and in a patch format (8-ml reconstitution volume) for implantation in rhesus macaques (refer to Supplementary Video 1). A similar protocol was used for human EHM formulations. **b**, Cellular composition of human ($n = 18$) and rhesus ($n = 4$) EHM assessed by flow cytometry for ACTN2$^+$ (CMs) and VIM$^+$/ACTN2$^-$ (StCs); refer also to snRNA-seq and additional flow cytometry data in Extended Data Fig. 1. Data are presented as mean ± s.e.m. **c**, Representative whole-mount stainings for ACTN2 (green) and nuclei (blue) in human ($n = 1$) and rhesus macaque ($n = 3$) EHM for comparison. **d**, Contractile function measured under isometric conditions and electrical field stimulation at 1.5 Hz; for details, refer to Supplementary Table 2. Scale bars, 1 cm (**a** (rightmost image), 50 µm (**a** (all other scale bars), **c**).

To advance our previous study on tissue-engineered heart repair[8,11,12,18] and ultimately allow clinical testing of myocardial remuscularization, we obtained guidance from the responsible regulatory authority in Germany, that is, the Paul-Ehrlich-Institut, as to the most adequate animal model to inform clinical translation. Considering the limitations of xenografting and the technical challenges associated with clinical autografting, homologous allograft implantation studies in a large animal model were recommended. The paucity of stable pluripotent stem cells from large animal species led us to focus

on the rhesus macaque (*Macaca mulatta*), for which we had already generated induced pluripotent stem (iPS) cells[23,24]. Here we tested four different rhesus macaque iPS cell lines, including two newly generated lines, to obtain insight into in vivo autograft responses (Supplementary Table 1).

All applied rhesus macaque iPS cell lines could be differentiated into CMs and stromal cells (StCs) with fibroblast properties (Fig. 1a) at high purities (identified by flow cytometry: 92 ± 2% ACTN2$^+$ CMs ($n = 7$ batches optimized protocol); >99% VIM$^+$ StCs ($n = 2$ batches);

Extended Data Fig. 1a,b) using similar protocols established for human iPS cells[8]. Purity was further confirmed by single-nucleus RNA sequencing (snRNA-seq; Extended Data Fig. 1a,b and Supplementary Note 1). In addition, snRNA-seq (9,994 rhesus macaque and 5,515 human nuclei) provided no evidence of residual pluripotent stem cell contamination. EHM constructed from rhesus macaque iPS cell-derived CMs and StCs displayed similar cell composition, structure and contractile performance as human EHM developed for clinical use (Fig. 1b–d and Extended Data Fig. 1c), with, however, lower maximal force of contraction (FOC) in rhesus EHM and species-specific characteristics (Supplementary Table 2), including a higher beating rate[25].

Before embarking on a non-human primate implantation studies, we first assessed the feasibility of rhesus EHM implantation in a widely used athymic nude rat model with ischaemia–reperfusion (I/R) injury[8,18,24] (Extended Data Fig. 2a). Fifteen rats were implanted with viable ($n = 7$) or lethally irradiated ($n = 8$) rhesus EHM ($2\times$ EHM loops constructed from a total of $5 \times 10^6$ cells per animal (that is, approximately $17 \times 10^6$ cells per kilogram body weight). In agreement with our earlier data[18], rhesus macaque CMs could be identified 4 weeks after implantation of viable but not lethally irradiated EHM (Extended Data Fig. 2b). Residual pluripotent stem cells and teratoma formation were not identified (Extended Data Fig. 2c). This was paralleled by an increase in ejection fraction (mean ± s.e.m. of BL (baseline) and 28-day values in percent: 47 ± 4 versus 59 ± 3; $P = 0.0174$ by two-sided paired $t$-test) and stroke volume (mean ± s.e.m. of BL and 28-day values in microlitres: 102 ± 10 versus 173 ± 13; $P = 0.0016$ by two-sided paired $t$-test) in rats with viable EHM grafts (Extended Data Fig. 2d).

To test whether iPS cell-EHM could be safely and efficaciously used in a translationally more relevant rhesus macaque model (Supplementary Table 3), we first implanted rhesus EHM in healthy macaques at a low dose (cohort 1: $1\times$ EHM patch constructed from a total of $40 \times 10^6$ cells, that is, approximately $4 \times 10^6$ cells per kilogram body weight; $n = 6$ allografts and $n = 1$ autograft; 3-month follow-up; Extended Data Figs. 3 and 4). In the first four animals of cohort 1, different immune suppression regimens, that is, tacrolimus ($n = 2$) versus tacrolimus with methylprednisolone ($n = 2$), were tested with evidence for better cell retention under combined calcineurin inhibition and steroid administration (tacrolimus: starting dose of $0.02$ mg kg$^{-1}$ d$^{-1}$ and target trough levels of 5–15 ng ml$^{-1}$ (Supplementary Data 1); methylprednisolone: $0.15$ mg kg$^{-1}$ d$^{-1}$) before extending the study with two more animals under combined tacrolimus and methylprednisolone, as well as an autograft study ($n = 1$) with no concurrent immunosuppression. Histopathological investigations identified similar EHM patch/CM retention in the allografts and autograft under these conditions (Fig. 2).

We next asked whether combined immunosuppression with tacrolimus and methylprednisolone ($n = 3$) would support long-term retention (6 months) of a high dose (cohort 2: $5\times$ EHM patches constructed from in total $200 \times 10^6$ cells, that is, approximately $20 \times 10^6$ cells per kilogram body weight) and whether withdrawal of immune suppression after 3 months would result in rejection of the allografts ($n = 2$). The latter experiments tested a rescue strategy that may be applied to reject CM grafts in case of unwanted effects. To further investigate whether tacrolimus could be substituted for another calcineurin inhibitor, cyclosporin (starting dose: $2$ mg kg$^{-1}$ d$^{-1}$; target trough levels: 140–250 ng ml$^{-1}$; Supplementary Data 1) was administered with methylprednisolone ($n = 1$). Similar to cohort 1, an autograft was implanted in one macaque. In agreement with the findings in cohort 1, CM allografts were retained under tacrolimus and methylprednisolone for 6 months (Fig. 2). Withdrawal of immune suppression after 3 months resulted in rejection of allograft CMs. Unexpectedly, we observed rejection of the autograft (no. 2500) and allograft (no. 2915) with cyclosporin and methylprednisolone. Donor-specific antibody (DSA) analyses revealed no evidence of autograft immunization in no. 2500 (Fig. 2 and Supplementary Data 2). A detailed analysis of the leukocyte infiltrate in the corresponding 3-month autograft model (cohort 1, no. 2483) identified

T cell-mediated rejection with concomitant B-cell accumulation and no evidence of an innate immune response (Extended Data Fig. 5). In contrast, strong allograft immunization was observed in no. 2915 (6-month allograft) after reduction of cyclosporin to target trough levels (140–250 ng ml$^{-1}$), suggesting rejection upon dose adjustment (Fig. 2 and Supplementary Data 1 and 2).

An unanticipated finding in the initial EHM implantation studies was the formation of terminally differentiated (Ki67$^{neg}$) osteochondral cells in five of the 14 implanted animals (0.7–35 mm$^2$). This finding was EHM dose-dependent (low-dose cohort 1: one of seven animals; high-dose cohort 2: four of seven animals) and was not associated with notable side effects. We had never made such an observation in previous rodent studies with human EHM grafts[8,18], including a Good Laboratory Practice toxicity, tumorigenicity and biodistribution study (20 nude rats implanted with $1\times$ EHM with 6 months of follow-up) performed as part of the investigator medicinal product dossier for EHM patches in clinical heart repair. We reasoned that the observation of osteochondral differentiation was a consequence of a suboptimal response in rhesus iPS cells to the iPS cell CM differentiation protocol developed and optimized for human use. Our snRNA-seq data, in agreement with this hypothesis, indicated a greater heterogeneity in the rhesus iPS cell derivatives (Extended Data Fig. 1a) with osteochondral cells in the CM pool (seven in 3,874 sequenced nuclei; Supplementary Note 1). By extending metabolic CM selection from 4 to 7 days, the heterogeneity of rhesus macaque derivatives could be reduced (Extended Data Fig. 1a).

As to structural and functional consequences of EHM grafting in healthy macaques, we identified the anticipated EHM dose-dependent thickening of the target heart wall ($1\times$ EHM: $+1.4 \pm 0.3$ mm ($n = 7$); $5\times$ EHM: $+4.5 \pm 0.6$ mm ($n = 7$) in end-diastole; $P < 0.001$ by two-sided unpaired $t$-test; Extended Data Fig. 4) with no evidence for functional impairment in EHM-implanted animals, irrespective of the study conditions (Supplementary Table 4). Continuous electrocardiogram (ECG) monitoring (Extended Data Fig. 6a) and thorough pathological analyses did not raise safety concerns regarding the EHM patch implantation. Clinical chemistry analyses revealed expected calcineurin inhibitor-related side effects (transaminitis (no. 2520) and hyperglycaemia (nos. 2909 and 2913)); in addition, eosinophilia after withdrawal of immune suppression (nos. 2869 and 2887) and elevation of N-terminal prohormone of brain natriuretic peptide (NT-proBNP) elevation (no. 2506) were observed (Supplementary Data 3).

Collectively, the adaptive experimental design in cohorts 1 and 2 allowed us to (1) gain insight into EHM dose-dependent effects ($1\times$ versus $5\times$ EHM; 4 versus $20 \times 10^6$ cells per kilogram body weight), (2) identify an appropriate immunosuppressive regimen (tacrolimus + methylprednisolone) and (3) confirm a rescue strategy (withdrawal of immunosuppression for controlled allograft rejection). Most importantly, we did not observe arrhythmia, tumour formation, EHM-related morbidities or mortality, and thus confirmed the maximal feasible dose (MFD) of $5\times$ EHM as a safe maximal dose in the healthy macaque model.

With the goal of investigating the safety and efficacy of EHM in heart failure, we set up a model of chronic heart failure in rhesus macaque by I/R injury in a new cohort 3 (Supplementary Table 3) with stably decreased local and global heart function (Extended Data Fig. 7 and Supplementary Table 5). Macaques were randomized to EHM implantation ($2\times$ EHM (approximately $8 \times 10^6$ cells per kilogram body weight), $n = 3$; $5\times$ EHM (approximately $20 \times 10^6$ cells per kilogram body weight), $n = 4$) or control groups with ($n = 3$) or without ($n = 4$) immunosuppression. In the $5\times$ EHM group, one macaque died in the post-anaesthesia recovery phase because of low cardiac output syndrome. The remaining EHM-implanted macaques had a clinically uneventful follow-up with tacrolimus-induced side effects (transaminitis and γ glutamyl transferase elevation (nos. 2819 and 2884), hypertriglyceridaemia (nos. 2884 and 2907), hyperglycaemia (nos. 2907, 2819 and 2884) and combined hyponatraemia and hypochloraemia (nos. 2907, 2819 and 2884) in some animals; Supplementary Data 3). We attributed the higher incidence

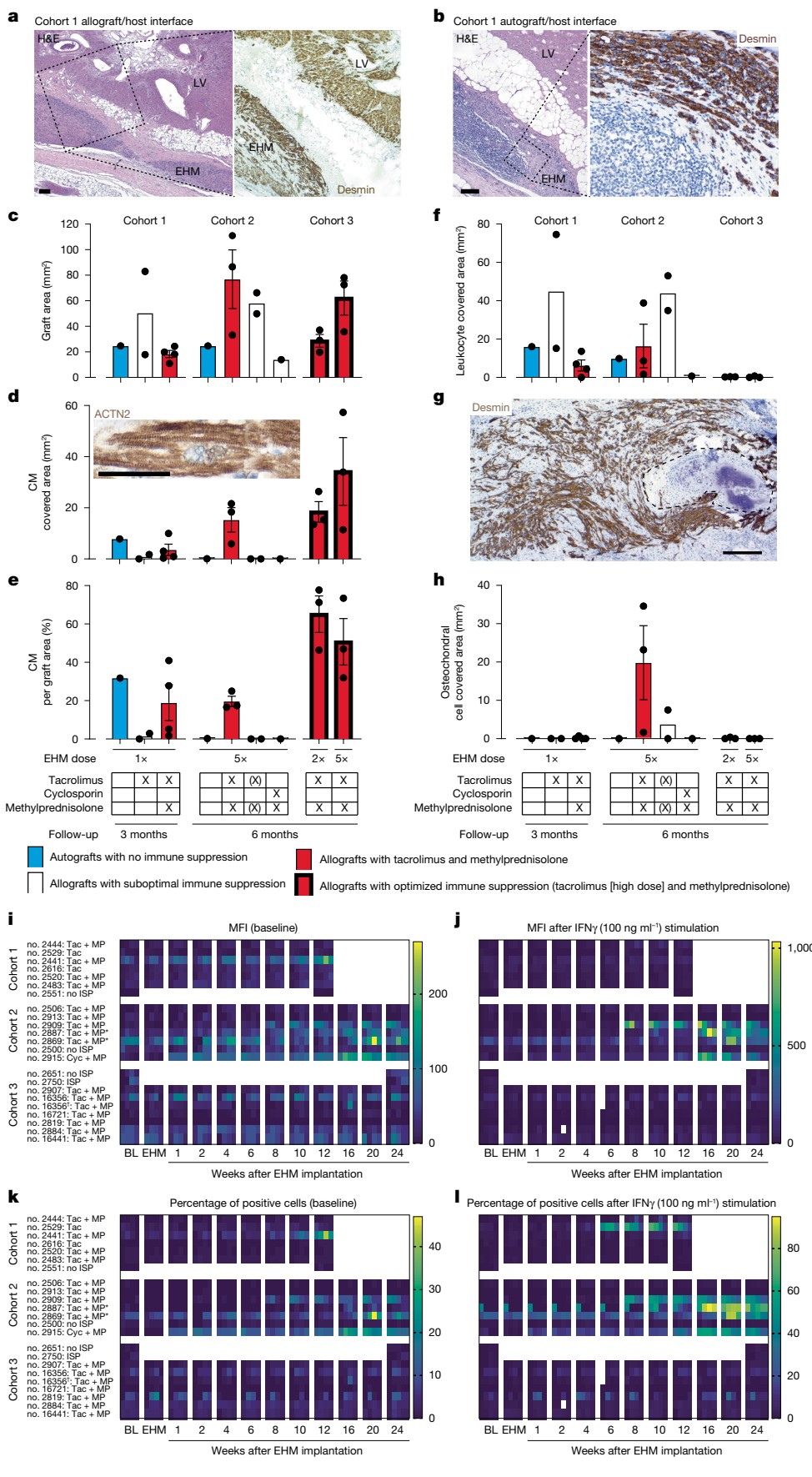

**Fig. 2 | Histopathology and allosensitization. a,b,** Haematoxylin and eosin (H&E) and desmin (brown with haematoxylin-stained nuclei) stains highlighting the host left ventricle (LV)/graft (EHM) interface in rhesus macaque no. 2520 (with allograft) (**a**) and no. 2483 (with autograft) (**b**). **c,** Quantification of the EHM graft area. **d–f,** CM (desmin⁺) engrafted area (inset: representative CM with regularly registered Z bands in brown (sarcomeric actinin (ACTN2)) (**d**), ratio of CM/EHM graft area (**e**) and area covered by inflammatory cells (leukocytes) (**f**). **g,** Desmin (brown) with haematoxylin-stained nuclei highlighting the largest osteochondral differentiation observed in the study (in rhesus macaque no. 2506; cohort 2 allograft). **h,** Quantification of cartilage/bone structures; note that in cohort 3, target tacrolimus trough levels were increased from approximately 10 ng ml⁻¹ (cohorts 1 and 2) to approximately 20 ng ml⁻¹ (details in Supplementary Data 1), and metabolic CM selection was extended to reduce potential osteochondral cell impurities (refer to Extended Data Fig. 1a (ii) versus (iii)). Bar graphs (n = 1, 2, 4, 1, 3, 2, 1, 3 and 3 animals from left to right): autograft data are in blue; allograft data obtained under immune suppression with tacrolimus and methylprednisolone are in red; additional experimental groups in white; s.e.m. included in groups with three or more biological replicates. Immune suppression protocols are summarized below the bar graphs. (X) indicates withdrawal of immune suppression after 3 months to induce allograft rejection. **i–l,** Detection of DSAs in serum from EHM-implanted macaques directed against iPS cell-derived CMs or StCs (marked with †) from the individual EHM preparations at the indicated time points. Cells were left unstimulated (**i,k**) or stimulated with interferon-γ (IFNγ) to enhance major histocompatibility complex (MHC) I expression (**j,l**). Mean fluorescence intensity (MFI) (**i,j**) and the proportion of stained cells in percentage of total cell number determined by flow cytometry (**k,l**) (details in Supplementary Data 2). Scale bars, 200 μm (**a,b,g**) and 50 μm (**d**).

and prevalence of tacrolimus-induced side effects in cohort 3 to the higher tacrolimus dosing, which we administered for better allograft retention (average trough concentration in cohorts 1 and 2 versus 3: 12 ± 4 versus 18 ± 6 ng ml⁻¹ (mean ± s.d.)). Under this revised protocol, no DSAs were observed at any time during the 6-month study period (Fig. 2 and Supplementary Data 2). Analysis of peripheral blood-derived

T cells (CD3$^+$, CD4$^+$ and CD8$^+$), B cells (CD20$^+$), natural killer cells (CD8$^+$/CD159a$^+$) and dendritic cells (CD11c$^+$) indicated no significant difference in frequency and activation (CD69$^+$, CD80$^+$ and HLA-DR$^+$) of the respective leukocyte subsets in the investigated groups (Source Data: Flow cytometry). Finally, negligible leukocyte infiltration was observed in the autopsy material with higher (approximately twofold) CM retention than that observed in cohorts 1 and 2 (Fig. 2d). EHM graft identity was further confirmed by genomic microsatellite analysis (Extended Data Fig. 8a).

In line with the now better controlled CM purity, only minute foci with osteochondral differentiations (0.015 and 0.4 mm$^2$) were observed in two animals implanted with 2× EHM and not in the animals implanted with 5× EHM. Similarly, as in the healthy macaques, we observed a sustained dose-dependent augmentation of the target heart wall thickness (mean ± s.e.m.: +1.5 ± 0.8 ($n = 3$) and +2.2 ± 0.1 ($n = 3$) mm in end-diastole in the 2× and 5× EHM groups versus −0.03 ± 0.07 mm ($n = 7$) in controls; $P = 0.0128$ and $P < 0.00001$ by two-sided unpaired $t$-test; Fig. 3a–c). Three (nos. 2819, 2884 and 16721) of six treated macaques (two from the 5× EHM group) presented with a sustained enhancement of target heart wall contractility (thickening fraction; mean ± s.e.m.: +21 ± 0.2% ($n = 3$) versus +0.6 ± 3.3% ($n = 7$; controls); $P = 0.0039$ unpaired two-sided $t$-test); one (no. 2907) of the EHM-treated macaques demonstrated a similar but apparently transiently enhanced heart wall contractility (Fig. 3d). Three (nos. 2819, 2884 and 2907) of the four macaques with enhanced target heart wall function also demonstrated an improved left ventricular ejection fraction (mean ± s.e.m.: +7 ± 3% ($n = 3$) versus −2 ± 2% ($n = 7$; controls); $P = 0.0389$ unpaired two-sided $t$-test; Fig. 3e and Supplementary Table 5). In agreement with cohorts 1 and 2, pathological analyses of cohort 3 heart explants (Extended Data Fig. 9) and ECG monitoring (Extended Data Fig. 6b) did not raise safety concerns related to EHM implantation.

Sustainable and impactful remuscularization requires functional vascularization. Gadolinium-based perfusion magnetic resonance imaging (MRI) measurements in a macaque (no. 2819) with clear separation of the host and graft (5× EHM) myocardium showed stable but attenuated perfusion compared to the remote myocardium during follow-up (Fig. 4a). In agreement with this observation, vascularization of EHM grafts was confirmed by histopathological analyses with no differences at 3 months (cohort 1) and 6 months (cohorts 2 and 3), that is at the study end points (281 ± 37 versus 1,262 ± 71 (left ventricle) and 1,145 ± 59 (right ventricle) blood vessels per square millimetre; $n = 20$ animals; Fig. 4b). Engrafted CMs were terminally differentiated (Ki67$^{neg}$) and smaller (1,678 ± 45 μm$^2$; $n = 13$ animals) than left ventricle (4,804 ± 172 μm$^2$) and right ventricle (3,685 ± 226 μm$^2$) CMs of the recipient animals ($n = 20$; Fig. 4c). Identification of TNNI1 (troponin I isoform in immature myocardium) and TNNI3 (troponin I isoform in adult ventricular myocardium), as well as stronger staining for MYL4 (myosin light chain isoform in immature myocardium) compared to MLY2 (myosin light chain isoform in adult ventricular myocardium), indicated a relative immaturity of the implanted versus host CMs. Engrafted CMs showed evidence of intercalated disk formation (CDH2), with sparse expression of the gap junction protein connexin 43 (GJA1; Extended Data Fig. 8b).

Owing to the availability of a heart from a patient who successfully underwent heart transplantation in the BioVAT-HF Phase 1/2 first-in-patient trial (Fig. 5a), we can provide clinical proof of CM retention after EHM implantation (10× EHM constructed from 400 × 10$^6$ CMs and StCs; Fig. 5b,c). As observed in the rhesus macaque model, engrafted CMs remained smaller (947 ± 35 μm$^2$; $n = 62$) than the recipient heart CMs (3,632 ± 168; $n = 33$) (Fig. 5d). Histological analyses confirmed a similar relative immaturity as observed in the rhesus macaque model (Extended Data Fig. 8d) and lower capillary density (187 ± 5 mm$^{-2}$) in the EHM graft than in the recipient heart (963 ± 12 mm$^{-2}$; $n = 3$ regions of interest analysed; Fig. 5e). No differences in capillary densities in the remote myocardium and in close proximity to the EHM suggest

that angiogenic paracrine effects are locally restricted to EHM. Graft identity was confirmed by single-nucleotide variant (SNV) analyses (Extended Data Fig. 8c). The patient demonstrated a stable disease course under EHM treatment (Extended Data Fig. 10). T cells and B cells, as well as macrophage (CD68) and minimal natural killer-cell (CD57) infiltrations, were noted (Extended Data Fig. 11). DSAs (Luminex) were not identified. Collectively, these findings indicate a local immune response against (1) the allograft; (2) the TachoSil support material; or (3) both, despite immune suppression at high target levels (Extended Data Fig. 10). Collectively, the obtained clinical data confirmed the translatability of heart remuscularization by EHM allograft implantation from rhesus macaques to human patients with advanced heart failure. It also established the rationale for continuation of patient treatment in the ongoing clinical trial with the MFD according to the clinical trial protocol (Supplementary Note 2), that is, 20× EHM constructed from 800 million iPS cell-derived CMs and StCs. Immune cell infiltration is commonly observed in heart transplant patients under guideline-directed immunosuppression[26] and will require further attention to improve the outcomes in EHM transplant patients.

To conclude, here we report on tissue-engineered myocardial heart muscle allograft and autograft implantation with long-term follow-up ($n = 7$ for 3 months and $n = 13$ for 6 months) in a clinically relevant homologous large animal model, providing clear evidence for sustained and functionally relevant remuscularization without unacceptable side effects. Importantly, no arrhythmia and tumour formation were observed in any of the EHM-implanted macaques ($n = 20$), which completed follow-up with 66 EHM patches implanted (constructed from 2.64 × 10$^9$ iPS cell-derived CMs and StCs). We assumed that EHM graft vascularization was key for preservation as well as the observed adaptive CM growth, and that the initial survival of EHM must have been supported by relative immaturity-related hypoxia resistance and anaerobic glycolysis until vascularization was established. No evidence for arrhythmia induction despite palpable remuscularization is encouraging because it contrasts findings from direct CM injection studies[4–7]. This fundamental difference may be the result of different modes of electrical integration. Intramurally injected CMs are clearly capable of coupling[12] and ectopic firing, leading to engraftment arrhythmia in large animal models[5–7], which may be attenuated by genetic depletion of depolarizing ion channels[27]. Tissue-engineered patches, by virtue of their epicardial engraftment, cannot readily establish canonical electromechanical connections via intercalated discs, but appear to be mechanically entrained over time to contribute to myocardial performance. This hypothesis is aligned with (1) previous findings of mechanically induced CM contractility[28], (2) observations of mechanically triggered contractions in EHM (Supplementary Video 3) and (3) the finding that chronic mechanical conditioning (1 Hz for 120 h) leads to adaptations of the EHM beating rate and rhythm (Extended Data Fig. 12). Additional studies are required to clarify the time course of, mechanism of and role of mechanical conditioning in integration of EHM grafts.

On the basis of the presented rhesus macaque data, we have obtained approval for the first-in-patient BioVAT-HF-DZHK20 Phase 1/2 clinical trial (ClinicalTrials.gov NCT04396899) to investigate the safety and efficacy of tissue-engineered heart repair by epicardial implantation of EHM allografts. The availability of a human heart explant from an early BioVAT-HF patient (10× EHM dose level) allowed us to present proof of concept for clinical translatability of EHM-based remuscularization of the failing human heart and in the particular patient case supporting evidence for the suitability of EHM as a bridge-to-transplant treatment in advanced heart failure. Moreover, the data were key to the decision to increase the EHM allograft dose in BioVAT-HF to the MFD (20× EHM), as outlined in the study protocol (Supplementary Note 2). Lessons learned from BioVAT and other ongoing clinical trials testing pluripotent stem cell-derived CM implantation[29,30] will improve our understanding of whether and how remuscularization of the failing human heart can

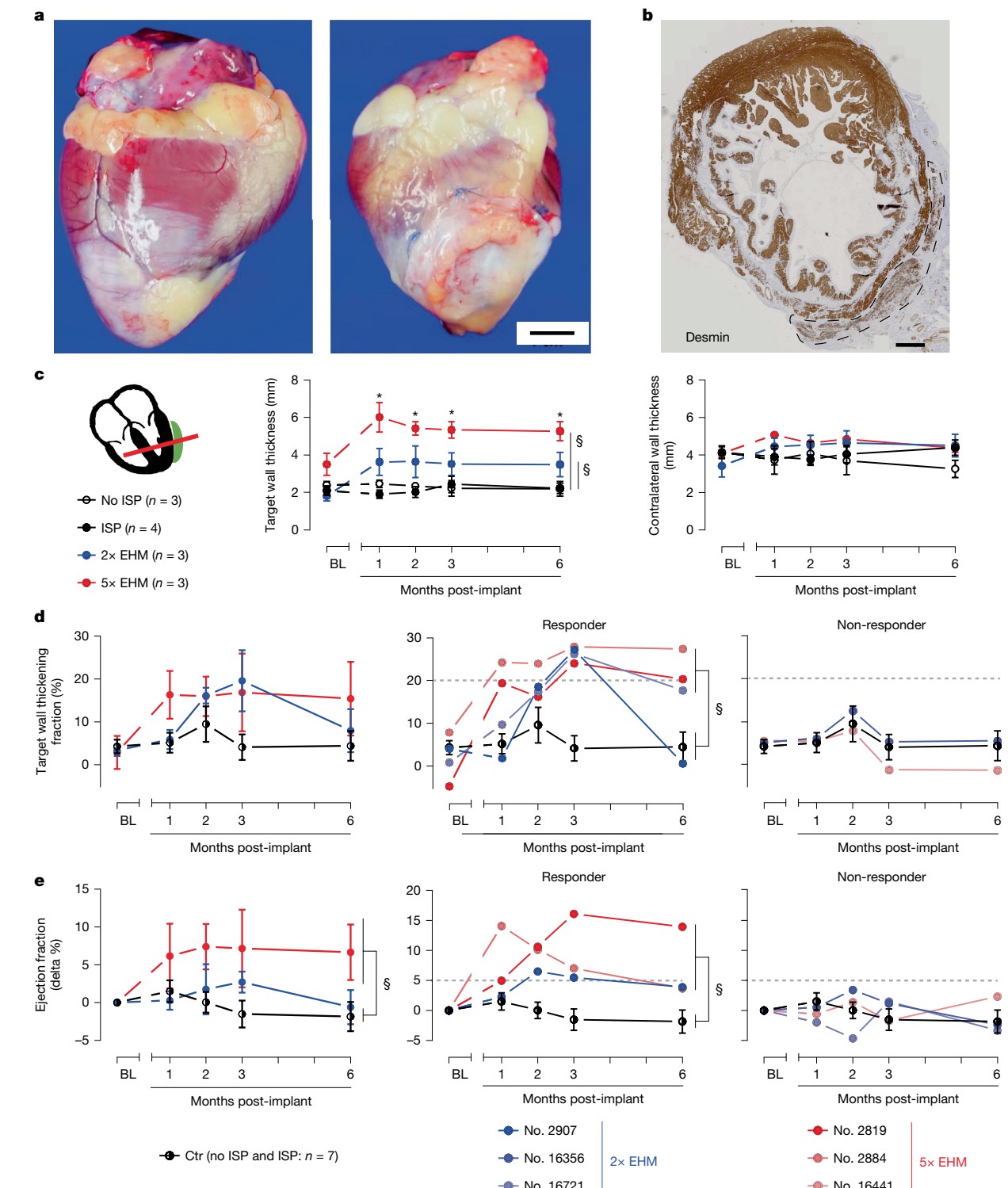

**Fig. 3 | EHM allografts enhance local and global heart function by remuscularization in a chronic heart failure model. a**, Representative images of a rhesus macaque heart without (no. 2750) and with (no. 16356; 2× EHM) EHM graft 6 months after implantation; total times on study were 367 (no. 2750) and 341 (no. 16356) days, respectively; this included follow-up after randomization into the immune suppression (ISP) control (no. 2750) and 2× EHM (no. 16356) groups for 173 and 167 days, respectively. **b**, CM retention 6 months after epicardial implantation of a 5× EHM (no. 2819; refer to Fig. 2 for a summary of the histopathological findings). **c–e**, MRI data: EHM dose-dependent thickening of the target heart wall with no effect on the contralateral heart wall (both parameters recorded in diastole) (**c**); target heart wall thickening fraction (local function; **d**) and ejection fraction (global function; **e**); aggregated values and data separated into responders and non-responders (cutoffs 20% in **d** and +5% in **e** indicated by striped lines; refer to Supplementary Table 5 for a summary of obtained MRI data). All MRI data in cohort 3 were analysed in long-axis two-chamber or four-chamber views to properly identify the mid-anteriorly to apically implanted EHM. Data are presented as mean ± s.e.m. Exact P values were calculated by two-way repeated measures analysis of variance with Greenhouse–Geisser correction and Dunnett's multiple comparison testing. *From left to right: 0.0148, 0.0349, 0.0143 and 0.0150 versus BL; §Ctr versus 2× EHM: 0.0464 and Ctr versus 5× EHM: 0.0002 (**c**); Ctr versus responder: 0.0106 (**d**); Ctr versus 5× EHM: 0.0345 and Ctr versus responder: 0.0065 (**e**). Ctr: no ISP and ISP combined; n = 7. Scale bars, 10 mm (**a**), 2 mm (**b**).

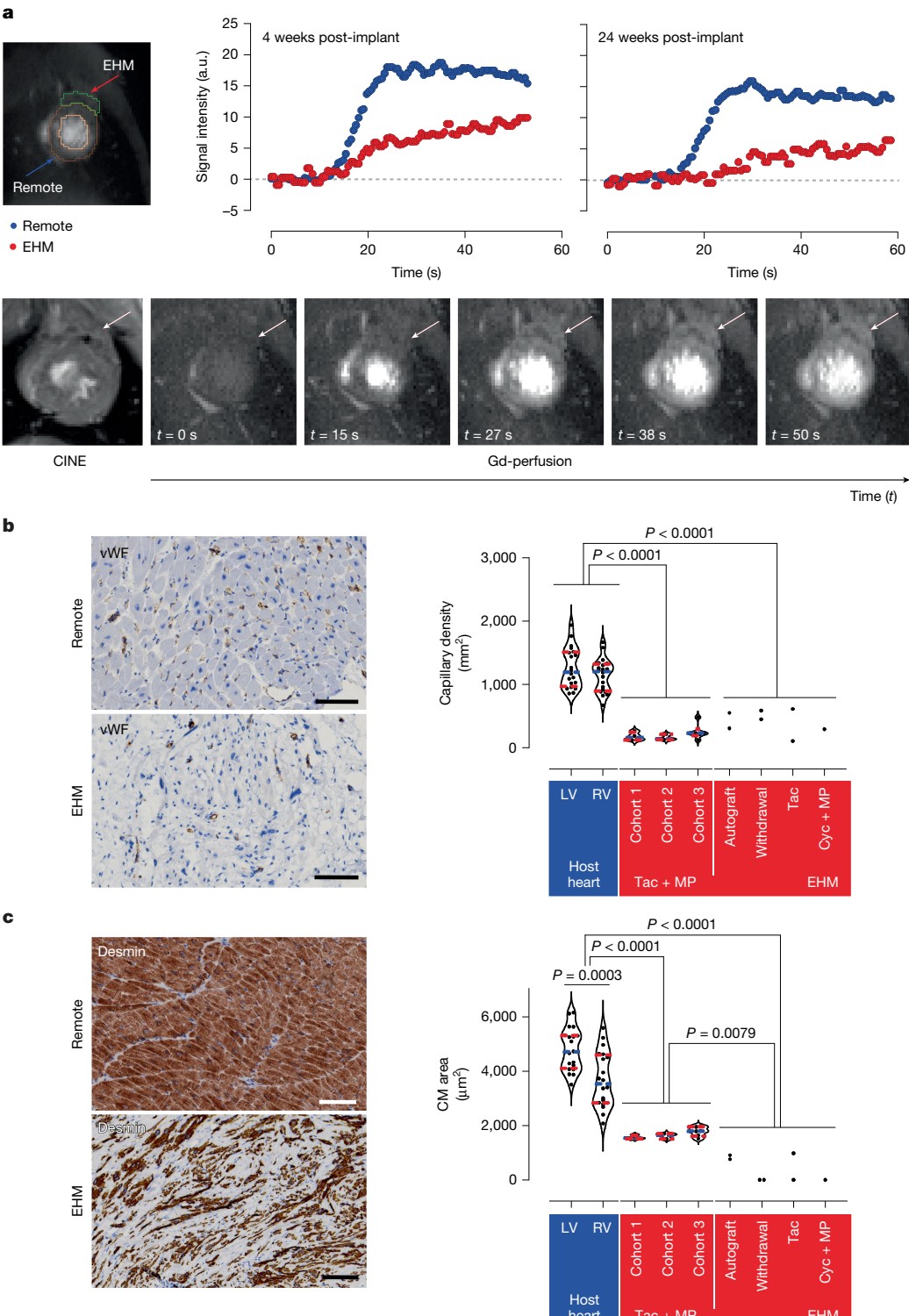

**Fig. 4 | Evidence for EHM allograft vascularization and perfusion.**
**a**, Gadolinium (Gd)-based perfusion MRI data obtained in 5× EHM-implanted rhesus macaque (no. 2819) with evidence for functional vascularization of EHM grafts in a heart failure model at the indicated time points. Left, the regions of interest, from which the Gd signal was reported, are encircled and distinguished as EHM and remote myocardium. The lower magnetic resonance images depict a CINE and the respective Gd-based perfusion images recorded at the indicated time points 4 weeks after EHM (marked by arrows) implantation. **b**, Histopathological analysis of vascularization in EHM and remote myocardium after immunohistochemistry staining for von Willebrand factor (vWF) (brown; experimental animal no. 2819). **c**, Histopathological analysis of CM size in EHM and remote myocardium after immunohistochemistry staining for desmin

(brown; experimental animal no. 2819). Violin plots in **b** and **c** with data points from all EHM-implanted animals (cohorts 1–3) at the respective study end points, that is, 3 and 6 months after EHM implantation ($n = 20$). Medians are indicated by striped blue lines; quartiles (25% and 75%) are indicated by striped red lines. Cohorts 1–3, animals under tacrolimus and methylprednisolone (Tac + MP); Cyc + MP, animal with cyclosporin and methylprednisolone; RV, right ventricle; Tac, animals with tacrolimus only; Withdrawal, animals with withdrawal of tacrolimus and methylprednisolone 3 months after EHM implantation. Exact $P$ values obtained by one-way analysis of variance with Tukey's multiple comparison testing are presented in **b**,**c**; exact $P$ value unpaired two-tailed Student's $t$-test for left ventricle versus right ventricle comparison is presented in **c**. Scale bars, 100 μm.

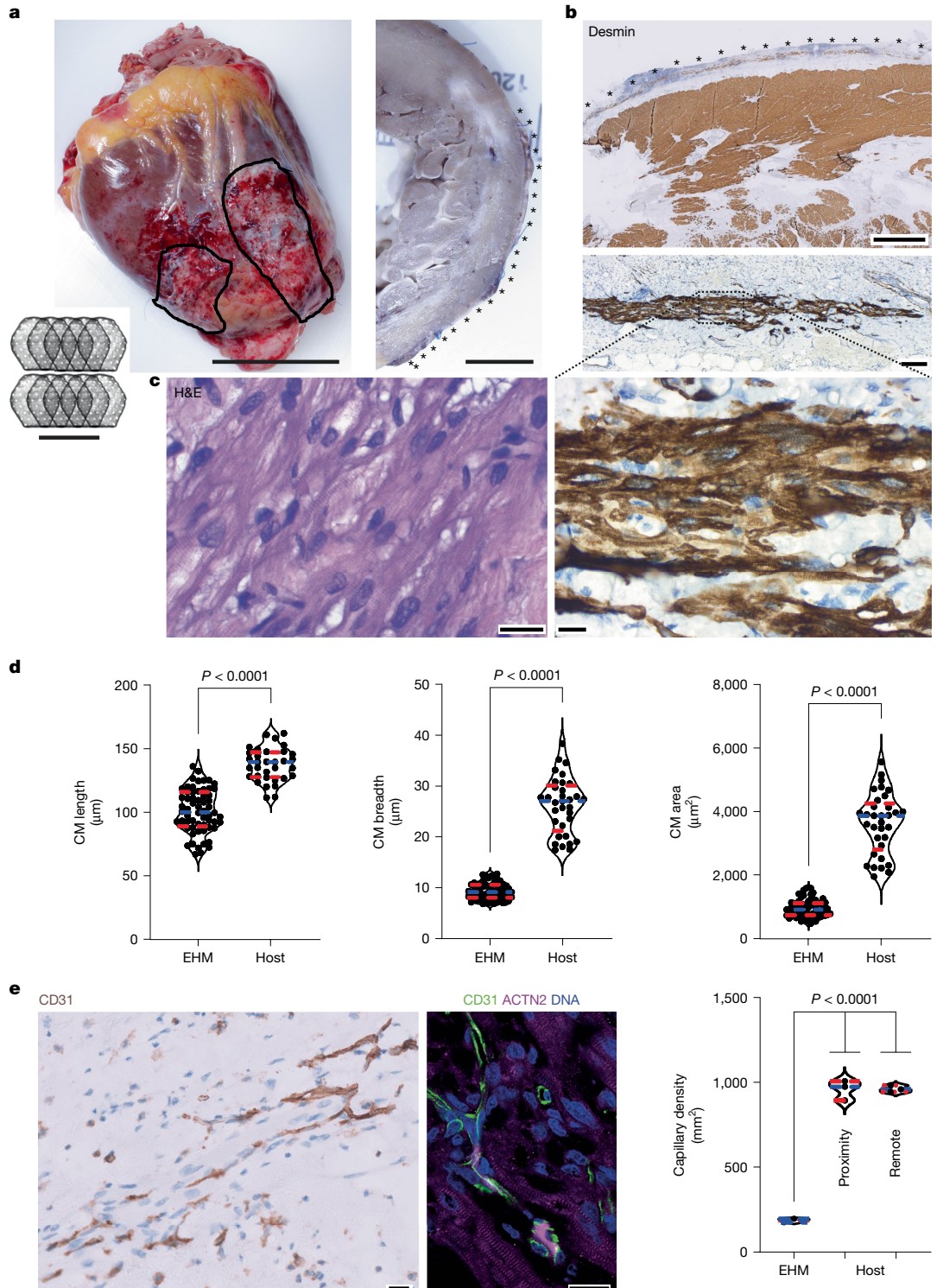

**Fig. 5 | Remuscularization of the human heart. a**, Explanted heart obtained 3 months after EHM implantation from a successfully heart transplanted BioVAT-HF patient (ID: 27016). EHMs were macroscopically visible as epicardial grafts (encircled) and are marked with asterisks in the cross-section (right). Inset, schematic of two arrays of overlapping single-layer EHM grafts applied in the patient. **b,c**, Overview and higher-power magnifications (**b**) of a cross-section with immunohistochemical labelling of desmin (brown; nuclei in blue (haematoxylin)); note the extension of the CM across the epicardial surface (asterisks) and the registered sarcomere patterning along the epicardial surface, which is similarly visible in haematoxylin and eosin (H&E) stains (**c**). **d**, Summary of the evaluation of CM length (end to end), breadth (cross-section at the nucleus level) and CM area (length × breadth). Violin plots with dots representing data from individual CMs in EHM ($n = 62$) and host myocardium ($n = 33$); medians indicated by striped blue lines and quartiles (25% and 75%) indicated by striped red lines. Exact $P$ values obtained by unpaired two-tailed Student's $t$-test are presented. **e**, Immunohistochemical and immunofluorescence labelling of CD31$^+$ endothelial cells with a quantification of capillary density in the EHM graft as well as the adjacent and remote recipient myocardium. Dots represent fields of view analysed for CD31-positive capillaries. Violin plots with dots representing the capillary density ($n = 3$ per group) in EHM graft as well as adjacent (proximity) and remote host myocardium; medians indicated by a striped blue line and quartiles (25% and 75%) indicated by striped red lines. Exact $P$ value (same for EHM versus proximity and EHM versus remote) obtained by one-way analysis of variance with Tukey's multiple comparison testing is presented. Scale bars, 5 cm (**a**, left and inset) and 1 cm (**a**, right), 2 mm (**b**, top), 200 μm (**b**, middle) and 20 μm (**b**, bottom), 20 μm (**c**,**e**).

be achieved with clinically meaningful outcomes. Recent advances in hypo-immune strategies[31–34], augmentation of vascularization from endogenous or exogenous sources[35] and engineering of alternative graft geometries using cast moulding[36] or 3D printing[37] offer exciting opportunities to further advance tissue-engineered heart repair.

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

[1]Department of Cardiothoracic and Vascular Surgery, University Medical Center Göttingen, Göttingen, Germany. [2]German Centre for Cardiovascular Research (DZHK), Partner Site Lower Saxony, Göttingen, Germany. [3]Department of Cardiology and Pneumology, University Medical Center Göttingen, Göttingen, Germany. [4]Department of Cardiology, Campus Kerckhoff of the Justus-Liebig-Universität Gießen, Kerckhoff-Clinic, Bad Nauheim, Germany. [5]Institute of Pharmacology and Toxicology, University Medical Center Göttingen, Göttingen, Germany. [6]Platform Degenerative Diseases, German Primate Center–Leibniz Institute for Primate Research, Göttingen, Germany. [7]Clinic for Cardiac and Thoracic Vascular Surgery, University Medical Center Schleswig Holstein, Campus Lübeck, Lübeck, Germany. [8]German Centre for Cardiovascular Research (DZHK), Partner Site North, Lübeck, Germany. [9]Institute of Pathology, University Medical Center Göttingen, Göttingen, Germany. [10]Functional Imaging Laboratory, German Primate Center, Göttingen, Germany. [11]Stanford Cardiovascular Institute, Stanford University School of Medicine, Stanford, CA, USA. [12]Department of Medicine, Division of Cardiovascular Medicine, Stanford University School of Medicine, Stanford, CA, USA. [13]Laboratory Animal Science Unit, German Primate Center–Leibniz Institute for Primate Research, Göttingen, Germany. [14]Institute of Diagnostic and Interventional Radiology, University Medical Center Göttingen, Göttingen, Germany. [15]Department for Epigenetics and Systems Medicine in Neurodegenerative Diseases, German Center for Neurodegenerative Diseases (DZNE), Göttingen, Germany. [16]Department of Psychiatry and Psychotherapy, University Medical Center Göttingen, Göttingen, Germany. [17]Cluster of Excellence "Multiscale Bioimaging: From Molecular Machines to Networks of Excitable Cells" (MBExC), University of Göttingen, Göttingen, Germany. [18]Institute of Cellular and Molecular Immunology, University Medical Center Göttingen, Göttingen, Germany. [19]Pluripotent Stem Cells Platform, Max Delbrück Center for Molecular Medicine in the Helmholtz Association (MDC), Berlin, Germany. [20]German Centre for Cardiovascular Research (DZHK), Partner Site Berlin, Berlin, Germany. [21]Pathology Unit, German Primate Center–Leibniz Institute for Primate Research, Göttingen, Germany. [22]Primate Genetics Laboratory, German Primate Center–Leibniz Institute for Primate Research, Göttingen, Germany. [23]Institute of Human Genetics, University Medical Center Göttingen, Göttingen, Germany. [24]Unit of Infection Models, German Primate Center–Leibniz Institute for Primate Research, Göttingen, Germany. [25]Department of Transfusion Medicine, University Medical Center Göttingen, Göttingen, Germany. [26]Department of Cardiovascular Sciences, Catholic University of Leuven and Department of Cardiovascular Diseases, University Hospitals Leuven, Leuven, Belgium. [27]German Center for Neurodegenerative Diseases (DZNE), Göttingen, Germany. [28]Fraunhofer Institute for Translational Medicine and Pharmacology (ITMP), Göttingen, Germany. [29]These authors contributed equally: Ahmad-Fawad Jebran, Tim Seidler, Malte Tiburcy. ✉e-mail: w.zimmermann@med.uni-goettingen.de

## Methods

### Approvals

Animal experiments were approved by the Stanford Animal Research Committee (nude rat study at Stanford) and the Niedersächsisches Landesamt für Verbraucherschutz und Lebensmittelsicherheit (LAVES; 33.42502-04-15/1807 and -16/2370; rhesus macaque (*Macaca mulatta*) studies in Göttingen). The BioVAT-HF-DZHK20 Phase 1/2 clinical trial (ClinicalTrials.gov NCT04396899) was approved by the responsible regulatory agency (Paul-Ehrlich-Institut) and the competent ethics committee (Ethics Committee of the University Medical Center Göttingen under file no. 18/7/20). Patients in BioVAT-HF are under guideline-directed medical therapy for advanced heart failure, which also includes listing for heart transplantation if the patients meet the selection criteria (bridge-to-transplant scenario). Informed consent was obtained for clinical trial participation, clinical trial data use and publication and histopathological analysis of the explanted heart. Echocardiography data from the BioVAT-HF trial were evaluated by a core laboratory (J.-U. V.) at the University of Leuven, Belgium.

### Cell lines

We used four rhesus macaque iPS cell lines (Supplementary Table 1) derived from skin fibroblasts using Sendai virus transduction of Oct4, Sox2, Nanog and cMyc (iPS cell 43110-4 and DPZ_iRH25.B1) (refs. 24,38) or episomal plasmid reprogramming (DPZ_iRH34.1 and DPZ_iRH23.1) (ref. 23) and human iPS cell line TC1133, also referred to as LiPSC-GR1.1 (ref. 39), derived by episomal plasmid reprogramming of CD34$^+$ cord blood cells and maintained under Good Manufacturing Practice (GMP) at all times. Cell line authentication was performed by MHC (human) and Mamu (*Macaca mulatta* MHC) typing. All lines were tested to be free from mycoplasma contamination (in-house testing using Lonza MycoAlert Detection Kit as well as external testing of GMP cell line and clinical implants by Minerva Biolabs, Berlin, Germany).

### CM differentiation

CMs were derived from human and rhesus iPS cells according to the previously introduced ABCF+I differentiation protocol[8]. Briefly, the ABCF+I protocol uses activin A (9 ng ml$^{-1}$; R&D Systems), bone morphogenic factor 4 (BMP4; 5 ng ml$^{-1}$; R&D Systems), CHIR99021 (1 µmol l$^{-1}$; Stemgent) and fibroblast growth factor 2 (FGF2; 5 ng ml$^{-1}$; PeproTech) in basal medium (RPMI, 2% B27 and 200 µmol l$^{-1}$ of L-ascorbic acid 2-phosphate sesquimagnesium salt hydrate; Sigma-Aldrich) for 3 days for mesoderm induction (MI) followed by cardiac specification in the presence of IWP4 (5 µmol l$^{-1}$; Stemgent) in basal medium from culture days 4–10. This was followed by recovery in basal medium and metabolic selection in RPMI without glucose (Thermo Fisher Scientific), sodium lactate (2.2 mmol l$^{-1}$; Sigma-Aldrich) and β-mercaptoethanol (100 µmol l$^{-1}$; Sigma-Aldrich). Cultures were performed in laminin (LN521; BioLamina)-coated T flasks (refer to Extended Data Fig. 1a for a summary of the CM differentiation protocols used).

### StC differentiation

StCs were derived from human and rhesus iPS cells using the newly developed ABCF-CRAB-VCF protocol. Briefly, the ABCF-CRAB-VCF protocol uses activin A (9 ng ml$^{-1}$; R&D Systems), BMP4 (5 ng ml$^{-1}$; R&D Systems), CHIR99021 (1 µmol l$^{-1}$; Stemgent) and FGF2 (5 ng ml$^{-1}$; PeproTech) in basal medium (RPMI, 2% B27 and 200 µmol l$^{-1}$ of L-ascorbic acid 2-phosphate sesquimagnesium salt hydrate; Sigma-Aldrich) for 7 days for MI, followed by epicardial differentiation in the presence of CHIR99021 (1 µmol l$^{-1}$; Stemgent), retinoic acid (4 µmol l$^{-1}$; Sigma-Aldrich) and BMP4 (50 ng ml$^{-1}$; R&D Systems) in basal medium from culture days 8–12. Subsequently, epithelial-to-mesenchymal transition was induced in knockout (KO) DMEM supplemented with 10% KO serum replacement (Thermo Fisher Scientific), vascular endothelial growth factor (VEGF) 165 (25 ng ml$^{-1}$; PeproTech), CHIR99021 (1 µmol l$^{-1}$; Stemgent) and FGF2 (50 ng ml$^{-1}$; PeproTech) from culture days 13–16. Finally, expansion was performed in KO DMEM with 10% KO serum replacement, VEGF (25 ng ml$^{-1}$) and FGF2 (50 ng ml$^{-1}$) from culture days 17–29 with repeated passaging of subconfluent cultures. Cultures were performed in laminin-coated (LN521; BioLamina; for MI, epicardial differentiation and epithelial-to-mesenchymal transition) and vitronectin-coated (Thermo Fisher Scientific; for expansion) T flasks (refer to Extended Data Fig. 1b for a summary of the StC differentiation protocol used).

### EHM construction

CMs and StCs were reconstituted as reported previously[8] and indicated in Supplementary Table 3 in a mixture of pH neutralized medical grade bovine collagen (EHM, 0.4 mg per 450 µl; Collagen Solutions LLC) and concentrated serum-free medium (2× RPMI and 8% B27 without insulin). Culture was in Iscove's medium with 4% B27 without insulin (Thermo Fisher Scientific), 1% non-essential amino acids, glutamine (2 mmol l$^{-1}$), ascorbic acid (300 µmol l$^{-1}$), IGF-1 (100 ng ml$^{-1}$), FGF2 (10 ng ml$^{-1}$), VEGF-165 (5 ng ml$^{-1}$) and TGF-β1 (5 ng ml$^{-1}$) (the latter only during culture days 0–3). All growth factors were purchased from PeproTech. Whole-mount immunofluorescence staining for ACTN2 with subsequent confocal analyses (LSM 710, Zen 2.3 SP1; Zeiss), contractility assessments and flow cytometry were performed routinely as reported previously[8] for quality control in EHM batches ($n = 17$ rhesus EHM batches and $n = 1$ human EHM batch) produced for implantation.

### Isometric force measurements

Surrogate EHM loops were suspended under isomeric conditions in thermostatted (37 °C) organ baths filled with Tyrode's solution (T2397; Sigma-Aldrich). Analysis was under electrical field stimulation at 1.5 Hz with 5-ms square pulses of 150-mA electrical current, as described previously[8]. EHMs were mechanically stretched at intervals of 125 µm until the maximum twitch force was observed ($L_{max}$) according to the Frank–Starling law. Contractility data were recorded with Bmon and analysed using the Amon software (VitroDat 3.52; Föhr Medical Instruments).

### Single-nucleus RNA sequencing

Flash frozen iPS cell-derived CMs, StCs or EHMs were used to prepare cell nuclei for cell composition analyses by snRNA-seq. All steps were performed on ice. Cells and EHM were first homogenized in 500 µl lysis buffer (Sigma) using plastic pestles (45 strokes in a microfuge tube). Then, 1.5 ml of lysis buffer was added and further incubated on ice for 7 min. Following centrifugation at 500$g$ and 4 °C for 5 min, the supernatant was discarded and the crude nuclear pellet was resuspended in 2-ml lysis buffer. After incubation for 7 min and centrifugation, the nuclear pellet was resuspended in 1.8 ml nuclei resuspension buffer (NRB containing 0.5% BSA (Serva), 1:200 RNaseIN Plus Inhibitor (Promega) and 1× protease inhibitor (Roche) in PBS (Invitrogen)) and centrifuged for 5 min at 500$g$ and 4 °C. The resulting nuclear pellet was resuspended in 500-µl NRB and filtered through a 35-µm strainer. To remove the remaining debris, nuclei were further purified by fluorescence-activated cell sorting (BD FACSAria III with an 85-µm nozzle and 45-psi sheath pressure) at 4 °C based on their size and collected into NRB-coated tubes. Following sorting, the nuclei were pelleted by centrifugation for 5 min, resuspended in 100 µl PBS and counted in a Neubauer chamber using 10% trypan blue. snRNA-seq libraries were prepared using Chromium Single Cell 3′-RNAseq v.3 library preparation kit (10X Genomics). Following the nucleus counting step, they were immediately processed to capture 2,000 single nuclei in a 10X chromium controller. Pooled complementary DNA libraries, prepared according to the manufacturer's guidelines, were then sequenced in an Illumina NextSeq 550 platform. Gene counts were obtained by aligning reads to the hg38 genome (NCBI:GCA_000001405.22; GRCh38.p7) using Cell Ranger software (v.3.0.2; 10X Genomics). The Cell Ranger count pipeline was used

to generate a gene-count matrix by mapping reads to the pre-messenger RNA as a reference to account for unspliced nuclear transcripts. The Scanpy package was used for pre-filtering, normalization and clustering[40]. Nuclei read counts indicating low-quality cells (that is, too many reads suggesting capture of more than one nucleus, too few reads with a read coverage of less than 200 genes or less than 10% housekeeping gene transcript coverage)[41] were excluded. Next, counts were scaled by the total library size multiplied by 10,000 and transformed into log space. Principal component analysis was performed on the variable genes, and UMAP was run on the top 50 principal components[42]. These were used to build a $k$-nearest-neighbour cell–cell graph with $k = 100$ neighbours. Subsequently, spectral decomposition over the graph was performed with 50 components, and the Louvain graph clustering algorithm was applied to identify cell clusters. We confirmed that the number of principal components captured almost all of the variance in the data. For each cluster, we assigned a cell-type label by manual evaluation of gene expression for sets of known marker genes (Supplementary Note 1).

### Flow cytometry analysis of cell composition

EHMs were washed in PBS and dissociated in Collagenase 1 (2 mg ml⁻¹; Sigma-Aldrich) in PBS with 20% FBS at 37 °C for 1 h followed by Accutase (Millipore), 0.025% trypsin (Thermo Fisher Scientific) and DNase I (20 µg ml⁻¹; Calbiochem) at 20–24 °C for 30 min. After fixation in 70% ice-cold EtOH for more than 10 min, cells were exposed to primary antibodies directed against sarcomeric actinin (ACTN2: 1:4,000; A7811; Sigma) or vimentin (VIM: 1:1,000; ab92547; Abcam) in blocking buffer for 45 min, followed by secondary antibodies in blocking buffer and Hoechst 33342 for 30 min at 4 °C (Supplementary Table 6). Control samples were exposed to undirected IgG1 (MAB002; R&D Systems). Human samples were fixed with 4% formaldehyde and exposed to conjugated antibodies directed against sarcomeric actinin (ACTN2-PE; 1:1,000; 130-106-937; Miltenyi Biotec) and vimentin (VIM-AF647; 1:1,000; 677807; BioLegend) for 15 min at 4 °C. A BD LSRII SORP system (BD Biosciences) or CytoFLEX (Beckman Coulter) was used for the flow cytometry analysis. The gating strategy was as described previously[8]. BD FACSDiva Software (BD Biosciences) or Kaluza (Beckman) was used for analysis.

### I/R injury and EHM implantation in the nude rat study

I/R injury was induced in male 8- to 10-week-old Rowett nude rats (250–350 g; Charles River Laboratories), as described previously[24]. Briefly, rats were treated with buprenorphine (0.05 mg kg⁻¹ by subcutaneous injection) and cefazolin (50 mg kg⁻¹ by intramuscular injection) followed by anaesthesia under 2% isoflurane. Body temperature monitoring and control were performed using a rectal probe and surgery on a 37 °C heating plate. After intubation, ventilation was with a tidal volume of 0.5 ml kg⁻¹ at a rate of 90 breaths per minute. A left thoracotomy was performed at the fourth intercostal space, and myocardial ischaemia was induced by occlusion of the left anterior descending (LAD) coronary artery. After 60 min of no flow, reperfusion was allowed by releasing the ligation. After closure of the chest wall, carprofen (5 mg kg⁻¹ s.c.) was administered for additional analgesia. Four days after I/R injury, animals were randomly grouped into two groups: (1) implantation of viable rhesus macaque EHM ($n = 7$) and (2) implantation of non-viable (60 Gy irradiated, as described previously[18]) rhesus macaque EHM ($n = 8$). For EHM implantation, the chest was reopened, and EHMs were attached to the left ventricular free wall with eight to 12 stitches using a 7-0 PROLENE suture, as described previously[18]. EHM implantations were performed by a study-group-blinded microsurgeon.

### Echocardiography in the nude rat study

Rats were sedated with 1–2% (v/v) isoflurane in oxygen and placed on a warming pad (37 °C) with ECG respiratory recording. M-mode ultrasound images in the short-axis view were acquired for systolic function using a high-frequency (21 MHz) linear transducer (MS250) on a small animal ultrasound system (Vevo 2100; VisualSonics). Tissue Doppler imaging was performed in a four-chamber view for diastolic function using the same ultrasound probe. Cardiac ultrasound data were analysed using the Vevo LAB software. The inner diameter of the left ventricle was measured from M-mode images at the end of diastole and end of systole. The volumes of the end of diastole and end of systole, fractional shortening and ejection fraction were sequentially calculated. Diastolic relaxation of rat hearts was measured from tissue Doppler images; diastolic dysfunction was defined as $E'/A' < 1$.

### Histopathology in the nude rat study

The rats were euthanized 4 weeks after EHM implantation. The hearts were excised and fixed in 4% formaldehyde in PBS overnight at 4 °C. After washing in PBS three times for 10 min, the hearts were cryoprotected in 30% sucrose overnight, embedded in optimum cutting temperature compound (Tissue-Tek) on dry ice and stored at −80 °C. Samples were cryosectioned into 10-µm sections and mounted on Superfrost Plus slides (Thermo Fisher Scientific). Permeabilization was performed using 0.1% Triton-X for 15 min at room temperature, followed by blocking with 10% normal goat serum in PBS for 25 min at room temperature. The primary antibodies (Supplementary Table 6) were applied at 4 °C overnight. Appropriate secondary antibodies were applied for 1 h at room temperature followed by 3 × 5 min washes in wash buffer. Slides were sealed with mounting solution containing DAPI for DNA labelling (ProLong Gold Antifade Mountant with DAPI; Thermo Fisher Scientific).

### EHM implant preparation in the rhesus macaque study

After transfer to the point of care, EHMs were prepared as 1×, 2× or 5× EHM assemblies by folding (1×) and stacking (2× and 5× EHM) in 3D-printed custom-made holders, followed by suturing (5-0 PROLENE; Ethicon) to a TachoSil membrane (4.5 × 4.5 cm; Takeda) used as (1) a security measure to prevent possible epicardial bleeding, (2) to support targeted surgical administration and (3) to reduce pericardial adhesions[43] (Extended Data Fig. 3).

### EHM implantation in healthy rhesus macaques

Fourteen (nine male and five female) rhesus macaques (*Macaca mulatta*) from the breeding colony at the German Primate Center (Deutsches Primatenzentrum) were assigned for the investigation of the feasibility and safety of EHM implantation in healthy animals (cohorts 1 and 2; Supplementary Table 3). The average body weight and age of animals at the time of EHM implantation were 8.4 ± 0.6 kg and 7.5 ± 0.7 years ($n = 14$). Animals were first anaesthetized by intramuscular injection of ketamine (7 mg kg⁻¹) (100 mg ml⁻¹; Wirtschaftsgenossenschaft deutscher Tierärzte (WDT)) and medetomidine (0.04 mg kg⁻¹) (Domitor, 1 mg ml⁻¹; Vetoquinol GmbH). After endotracheal intubation, total intravenous anaesthesia was used by administration of propofol (Propofol-Lipuro 2%, 10–40 mg kg⁻¹ h⁻¹; B. Braun Melsungen AG) and fentanyl (Fentadon, 10 µg kg⁻¹ h⁻¹; Dechra) via an intravenous line introduced into the *vena saphena* or *vena cephalica* using a 22G line (Vasofix Safety, 22 G; B. Braun Melsungen AG) under continuous arterial blood pressure monitoring (via the A. tibialis), ventilation with at least 60% oxygen depending on $O_2$ saturation (Nonin 7500FO) and $CO_2$ concentration in the exhaled air (Siemens Servo Ventilator 900C), and ECG monitoring. Body temperature was monitored using an oesophageal probe. Electrolytes were balanced by continuous Sterofundin infusion (5–10 ml kg⁻¹ h⁻¹; B. Braun Melsungen). Amoxicillin (Duphamox, 15 mg kg⁻¹; Zoetis) and meloxicam (Metacam, 0.2 mg kg⁻¹; Boehringer Ingelheim) or carprofen (Rimadyl, 2–4 mg kg⁻¹; Zoetis) were administered perioperatively. After incision in the fifth intercostal space (according to echocardiographic assessment of the heart position in the operating theatre), the pericardium was opened horizontally, anterior to the phrenic nerve. Epicardial sutures were placed to ensure the proper positioning of the

EHM implant at the epicardial target site. After positioning and suturing (six 5-0 PROLENE single-knot sutures) of the EHM onto the epicardial heart wall and closure of the pericardial sac, the chest wall was closed with surgical sutures. The animals were subsequently weaned from anaesthesia and returned to the animal facility under veterinary care. In cohort 1, 1× EHMs (constructed from $40 \times 10^6$ cells) were implanted (six allografts and one autograft) in an adaptive study design to first identify optimal immune suppression ($n = 2 + 2$; tacrolimus only versus tacrolimus plus methylprednisolone) and then extend the study under optimal immune suppression ($n = 2$) with a parallel autograft without immune suppression as a reference ($n = 1$); follow-up in cohort 1 was 3 months. Cohort 2 tested the MFD (5× EHM constructed from $200 \times 10^6$ cells, approximately 5 g $w/w$) as allografts ($n = 5$) under the cohort 1 identified optimal immune suppression protocol (tacrolimus plus methylprednisolone). An autograft study without concurrent immune suppression ($n = 1$) served as a reference. Cyclosporin with methylprednisolone was tested in one allografted animal ($n = 1$) as an alternative to tacrolimus with methylprednisolone. In two allograft animals, immune suppression was stopped 3 months after implantation to investigate the consequences of EHM rejection. Calcineurin inhibitors were started 5 days and methylprednisolone 2 days before EHM implantation to ensure effective immune suppression already at the time of EHM implantation. Therapeutic drug monitoring was applied with dose adjustment, as needed, to ensure target tacrolimus and cyclosporin trough levels (Supplementary Data 1). To establish baseline information, MRI and clinical chemistry data were obtained at two time points: (1) $44 \pm 7$ days ($n = 14$) before EHM implantation (baseline 1) and (2) on the day of EHM implantation (baseline 2). Additionally, MRI studies were performed 28, 56 and 84 days (cohort 1) and in addition 168 days (cohort 2) after EHM implantation. Blood draws for clinical chemistry studies were taken at 9, 18, 27, 42, 56, 70 and 84 (±2) days (cohort 1) and in addition 112, 140 and 168 (±2) days (cohort 2) after EHM implantation.

## I/R injury in the macaque studies

Twenty rhesus macaques (*Macaca mulatta*) from the breeding colony at the German Primate Center (Deutsches Primatenzentrum) were assigned to the 'EHM implantation in chronic heart failure after I/R injury' cohort (cohort 3; Supplementary Table 3). This study included 14 male and six female macaques. One female macaque (no. 15389) had to be excluded according to the recommendation of the responsible veterinarians before I/R injury induction because of low (6 kg) and no gain in body weight over a time period of more than 2 months after transfer into the experimental animal unit. The average body weight and age of animals at the time of I/R injury were $9.0 \pm 0.3$ kg and $7.3 \pm 0.3$ years ($n = 19$). I/R injury was inflicted under total intravenous anaesthesia, as described above, with repeated control of activated clotting time (ACT; target: 250–350 s; ACT Plus; Medtronic) under heparinization (starting dose of 7,000 U by intravenous injection followed by dose adjustments according to ACT values; Heparin-Natrium-25000-ratiopharm). A 5F Launcher internal mammary artery guiding catheter (Medtronic), bent to fit the aortic root anatomy of the rhesus macaque using heated air, was advanced via the femoral artery into the left main coronary artery under fluoroscopy (Siemens Artis zee multi-purpose). A 0.014 in. coronary guide-wire was advanced into the distal LAD, and a 1.2–2.0 mm over the wire balloon (EMERGE; Boston Scientific) was inflated to 10 bar in the mid-LAD. Occlusion was confirmed via angiography. Balloon pressure (10 bar) was continuously maintained for 3 h and monitored with additional angiographic confirmation of vessel occlusion once per hour (Extended Data Fig. 7). ST-wave elevation myocardial infarction was confirmed by ECG and blood biomarkers (troponin, total creatine kinase/muscle- and brain-type creatine kinase and lactate dehydrogenase; Supplementary Data 3). Five of 19 macaques subjected to I/R injury died because of bleeding complications ($n = 3$) or sudden cardiac death ($n = 2$; Supplementary Table 3).

## EHM implantation in rhesus macaques with chronic heart failure

Six months ($176 \pm 6$ days; $n = 14$) after I/R injury and confirmation of chronic heart failure development (Extended Data Fig. 7), rhesus macaques were randomized to be included in control groups (with ($n = 4$) or without ($n = 3$) immune suppression) or implanted with either 2× ($n = 3$) or 5× ($n = 4$) EHM, as described above. For immune suppression in cohort 3, tacrolimus (0.02 mg kg$^{-1}$ d$^{-1}$ PROGRAF i.m. with dose adjustment to reach target trough levels of 15–25 ng ml$^{-1}$) and methylprednisolone (0.15 mg kg$^{-1}$ d$^{-1}$) were administered. Tacrolimus was started 5 days and methylprednisolone 2 days before EHM implantation. The 6-month follow-up (MRI and blood draws) was the same as in cohort 2.

## Implantation of telemetry ECG event recorders

Reveal LINQ (Medtronic) telemetry monitors were implanted subcutaneously under general anaesthesia at baseline 1 investigations (44 ± 7 days before EHM implantation in cohorts 1 and 2 ($n = 14$); 40 ± 7 days before I/R injury ($n = 19$) and 176 ± 6 days ($n = 14$) before randomization into the four study groups in cohort 3) according to the supplier's instructions for heart rate, activity monitoring and arrhythmia event documentation. The optimal position for the ECG recorder was identified by surface ECG assessments on the right parasternal chest. This location was chosen to not interfere with the surgical route of EHM implantation and the cardiac MRI investigations. The event recorder was set to detect tachycardia (greater than or equal to 222 bpm for 16 or more beats), bradycardia (less than or equal to 30 bpm for four or more beats) and asystole (no beats for more than 3 s).

## Clinical chemistry

Animals were first anaesthetized by i.m. injection of ketamine (7 mg kg$^{-1}$) (100 mg ml$^{-1}$ WDT) and medetomidine (0.04 mg kg$^{-1}$) (Domitor, 1 mg ml$^{-1}$; Vetoquinol GmbH). Blood was collected from the femoral vein using the BD Vacutainer system and 22G BD Vacutainer Eclipse blood collection needles. Standard clinical chemistry and haematology analyses were performed at the University Medical Center Central Laboratory.

## Magnetic resonance imaging

Animals were sedated with 7 mg kg$^{-1}$ of ketamine (100 mg ml$^{-1}$, WDT) and 0.04 mg kg$^{-1}$ medetomidine (Domitor, 1 mg ml$^{-1}$; Vetoquinol GmbH) via intramuscular injection. After endotracheal intubation, anaesthesia was maintained via a continuous infusion of Propofol-Lipuro 2% (10–40 mg kg$^{-1}$ h$^{-1}$; B. Braun Melsungen AG) via an intravenous line introduced into the vena saphena or vena cephalica using a 22G line (Vasofix Safety, 22G; B. Braun Melsungen AG). Pressure-controlled ventilation (Servo Ventilator 900C; Siemens-Elema AB) with at least 60% oxygen in exhaled air was applied to keep the O$_2$ saturation between 98% and 100% (measured by pulse oximetry; Nonin 7500FO). The CO$_2$ concentration in exhaled air (IntelliVue; Philips) was maintained below 30% to avoid spontaneous breathing. Measurements were performed on a 3T MRI system (MAGNETOM Prisma; Siemens Healthineers) using a 16-channel multipurpose coil (VARIETY; Noras MRI Products) for signal detection, as described previously[44]. Gadolinium-based contrast agent was applied intravenously as a contrast agent to identify EHM graft perfusion. Data were analysed independently by four blinded investigators as to condition and timing using Segment v.4.0 R12067 (Medviso, segment.heiberg.se) and Medis-Suite v.3.2 software with QMass module v.8.1 (Medis).

## Donor-specific antibodies

The iPS cell-derived CMs and StCs, unstimulated and after IFNγ (100 ng ml$^{-1}$ for 48 h) were exposed to sera obtained before (pre) and at the indicated time points during the study at different dilutions (1:5 to 1:40). A fluorescein isothiocyanate-labelled anti-rhesus

immunoglobulin G antibody (4700-02; SouthernBiotech) was used to detect antibodies in the sera bound to the CMs and StCs. The cell mean fluorescence intensity (MFI) and the proportion of stained cells were determined by flow cytometry (LSR II SORP; BD Biosciences). Antibodies that display a selective reactivity to IFNγ-stimulated CMs presumably include DSAs to MHC class I molecules. The pan-human leukocyte antigen antibody W6/32 (BioLegend), which reacts with MHC class I molecules of rhesus macaques, was used to demonstrate the expression of these molecules on CMs and StCs.

### Flow cytometry analysis of peripheral immune cells

Whole blood (50 µl) was stained with a mixture of pre-titrated monoclonal antibodies (Supplementary Table 6) for 30 min at room temperature in the dark. Lysis of red blood cells and fixation were performed by incubation with 1-ml red blood cell lysis/fixation solution (BioLegend) for 15 min. Following a washing step with PBS/BSA, cells were analysed using an LSR II cytometer (BD Biosciences) and FlowJo 9.6 software (Tree Star).

### Pathology

Cardiac arrest was introduced under deep ketamine/medetomidin anaesthesia by injection of pentobarbital (90–120 mg kg$^{-1}$) in a potassium chloride solution (30 mmol l$^{-1}$) under online monitoring of heart rate and rhythm via the Reveal LINQ monitor. After confirmed cardiac arrest, the hearts were excised, weighed, photographed (Extended Data Fig. 9) and formaldehyde fixed for histopathological analyses. In addition, the following organs were collected and subjected to macroscopic and microscopic pathology analyses with a particular focus on abnormal cell growth: oesophagus, aorta, lung, trachea, liver, pancreas, spleen, reproductive organs, lymph nodes, thymus, bladder, stomach, small and large intestines, thyroid gland, adrenal gland and parotid gland samples. Histopathological studies were performed on haematoxylin and eosin (H&E)-stained paraffin sections according to standard pathological procedures. The heart explants were sectioned from base to apex in 3- to 5-mm slices before paraffin embedding. Paraffin embedded samples were cut into 2-µm tissue sections and counterstained with Meyer's haematoxylin (Dako; Agilent Technologies) for 8 min and analysed by light microscopy. Immunohistochemistry reactions were performed after antigen retrieval at 97 °C in citrate buffer (pH 6) or EDTA buffer (pH 9). The following antibodies and dilutions were used: anti-vWF (polyclonal rabbit, EDTA buffer (pH 9), ready-to-use (RTU), 20 min of incubation, Dako; Agilent Technologies), anti-desmin (monoclonal mouse, EDTA buffer (pH 9), RTU, 20 min of incubation, clone D33, Dako; Agilent Technologies) and anti-Ki67 (monoclonal mouse, EDTA buffer (pH 9), RTU, 20 min of incubation, MIB-1, Dako; Agilent Technologies). The sections were incubated with an RTU horseradish peroxidase-labelled secondary antibody at room temperature for 25 min (anti-rabbit/mouse, produced in goat; REAL EnVision Detection System, Dako, Agilent Technologies). The substrate DAB+Chromogen system produces a brown end-product and is applied to visualize the site of the target antigen (REAL DAB+Chromogen; Dako) (refer to an overview of applied antibodies in Supplementary Table 6). Individual CM area was calculated by multiplication of CM length and breadth measured in longitudinally sectioned desmin-positive CMs.

### Graft identity assessment

DNA was isolated, as described previously[45], from micro-dissected formaldehyde-fixed paraffin embedded (FFPE) slices using the innuPREP FFPE DNA Kit on the InnuPure C16 System (Analytik Jena) according to the manufacturer's instructions. Samples were obtained from desmin-positive remote and EHM engrafted areas. DNA concentrations were measured on a Qubit 3.0 Fluorometer (Thermo Fisher Scientific). Microsatellite genotyping was performed in macaque samples using a genotyping-by-sequencing approach, as described previously[46]. Allele calling based on sequence data generated on Illumina's MiSeq

platform (251-bp forward and 51-bp reverse) was done with the CHIIMP pipeline[47]. In human samples, deep sequencing of a targeted multigene panel (78 genes) was performed on 50 ng genomic DNA. For library preparation, SureSelectXT HS target enrichment kit (Agilent) with enzymatic fragmentation was used following the manufacturer's protocol. Libraries were sequenced on an Illumina NovaSeq 6000 with 2 × 150-bp read length and with mean coverage of 3,000×. SeqPilot (JSI Medical Systems GmbH) and Varbank 2.0 (Cologne Center for Genomics, University of Cologne) software was used to align sequences to a human reference genome (hg19) and for SNV calling. SNVs were filtered against (1) absence from control area, (2) high coverage and (3) exclusion of sequence artefacts.

### Mechanical conditioning study

EHMs (*n* = 5) were subjected to mechanical stretching in a modified contraction measurement set-up with a programmable actuator[48]. After suspension and equilibration in EHM medium, EHMs were stretched stepwise from $L_0$ (slack length) to $L_{max}$ (preloaded to achieve maximal FOC). Subsequently, preload was adjusted to approximately 80–90% of $L_{max}$, and EHMs were subjected to chronic mechanical stimulation (200-µm extension for 500 ms) at 1 Hz. Contractility was recorded using an automated routine with 2-min data acquisition per hour for the whole study duration of 120 h.

### Statistical analyses

Data are reported as mean ± s.e.m. unless otherwise indicated. Microsoft Excel 2019 MSO (16.0.10415.20025) and GraphPad Prism 10.1.2 were used for statistical analyses. Sample numbers (biological replicates) and statistical tests are indicated in the main body of the text, tables and figure legends.

### Reporting summary

Further information on research design is available in the Nature Portfolio Reporting Summary linked to this article.

## Data availability

Data are available in the Supplementary Information. snRNA-seq data can be retrieved at https://www.ncbi.nlm.nih.gov/geo/query/acc.cgi?acc=GSE276021. Additional requests will be handled by the corresponding author. Source data are provided with this paper.

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

**Acknowledgements** This study was supported by grants from the German Federal Ministry of Science and Education (BMBF FKZ 13GW0007A) and the German Centre for Cardiovascular

Research (DZHK). W.-H.Z. is supported by the German Centre for Cardiovascular Research (DZHK), the German Federal Ministry of Science and Education (BMBF FKZ 161L0250A), the German Research Foundation (DFG SFB 1002 C04/S01, IRTG 1816, EXC 2067-1) and the Leducq Foundation (20CVD04). J.C.W. is supported by the California Institute of Regenerative Medicine (CIRM), RT3-07798 and CLIN2-12735. R.B. is supported by the DZHK and BMBF (FKZ 161L0250A). G.K. is supported by the BMBF-German Network of RASopathy Research (GeNeRARe) subproject 5 (BMBF FKZ 01GM1902D). We thank F. Agdas, A. Berenson, O. Dschun, U. Goedecke, C. Klopper, S. Kögler, K. Kötz, L. Lukat, F. Martinpott, I. Quentin, D. Reher, J. Schnalke, A. Schraut, N. Umland and L. Elsner for excellent technical assistance. We thank C. S. Ferrel (cohorts 1 and 2) and H. Moussavi (cohort 3; German Primate Center) for performing MRI analyses. We thank J. Engelmayer and L. Binder (UMG Laboratory) for performing and supervising the clinical chemistry analyses. We thank T. Friede, M. Placzek, K. Schröder, F. Walker and R. Tostmann for their support of the BioVAT-HF study. We thank P. Menasche (Hôpital Européen Georges-Pompidou) as well as C. W. Don and C. E. Murry (University of Washington) for their advice as to I/R injury induction. Generation of the GMP line LiPSC-GR1.1 (also referred to as TC1133; lot number 50-001-21) was supported by the NIH Common Fund Regenerative Medicine Program and reported in Stem Cell Reports[39]. The NIH Common Fund and the National Center for Advancing Translational Sciences are joint stewards of the LiPSC-GR1.1 resource. A derivative from a GMP working cell bank of the TC1133 line was made available for research use by Repairon GmbH. Illustrations in Extended Data Figs. 1, 2a and 4a were created using BioRender (https://biorender.com).

**Author contributions** W.-H.Z. conceived the study. A.-F.J., I.K., H.B., B.F. and S.E. performed surgical interventions, including EHM implantations. M.T. was responsible for the EHM construction and characterization. W.-H.Z. and M.T. performed the EHM assembly at the point of care for implantation. T.S. and A.-F.J. together with the responsible veterinarians at the German Primate Center (Ma.D., C.D., M.M., S.M. and R.H.) established the heart failure model. A.M. and S.B. performed the MRI studies. C.H. (cohorts 1 and 2) as well as T.N. (cohort 3) analysed the MRI data. J.K., C.O.R. and J.L. supervised the MRI data analysis. Mi.D. performed echocardiographs in rhesus macaques. T.S. and W.-H.Z. interpreted the ECG telemetry data. D.K., M.S.S, L.K. and A.F. performed the snRNA-seq studies. T.M. constructed EHM casting moulds and support instruments by 3D printing. G.K. adapted the force analysis system and performed the mechanical conditioning experiments. R.D., B.P., L.W., B.R., E.L. and C.S.-H. conducted the immunology studies. H.Y., X.Q. and J.C.W. conducted the rat study. K.M.-R., E.G.-D, M.B., F.B. and P.S. performed the pathological analyses. I.R.-P., M.S., S.D., J.C.W. and R.B. provided rhesus macaque iPS cell lines and advice as to experimental design. A.-F.J., C.H., G.H., I.K., K.H., S.E. and T.S. provided expert clinical advice and are involved as study physicians in BioVAT-HF. J.-U.V. was responsible for the echocardiography core laboratory analyses. C.R., L.Z., L.W., S.K., G.Y. and B.W. performed the genetic identity studies. T.L. and J.R. produced the EHM for clinical use. All authors reviewed and approved the submission of the paper.

**Competing interests** W.-H.Z. is the founder, equity holder and advisor of Repairon GmbH. M.T. is advisor of Repairon GmbH. Repairon is working towards market authorization of EHM as advanced therapy medicinal product for applications in heart failure. Repairon had no influence on the design, conduct and interpretation of the study. The other authors declare no competing interests.

## Additional information
**Correspondence and requests for materials** should be addressed to Wolfram-Hubertus Zimmermann.

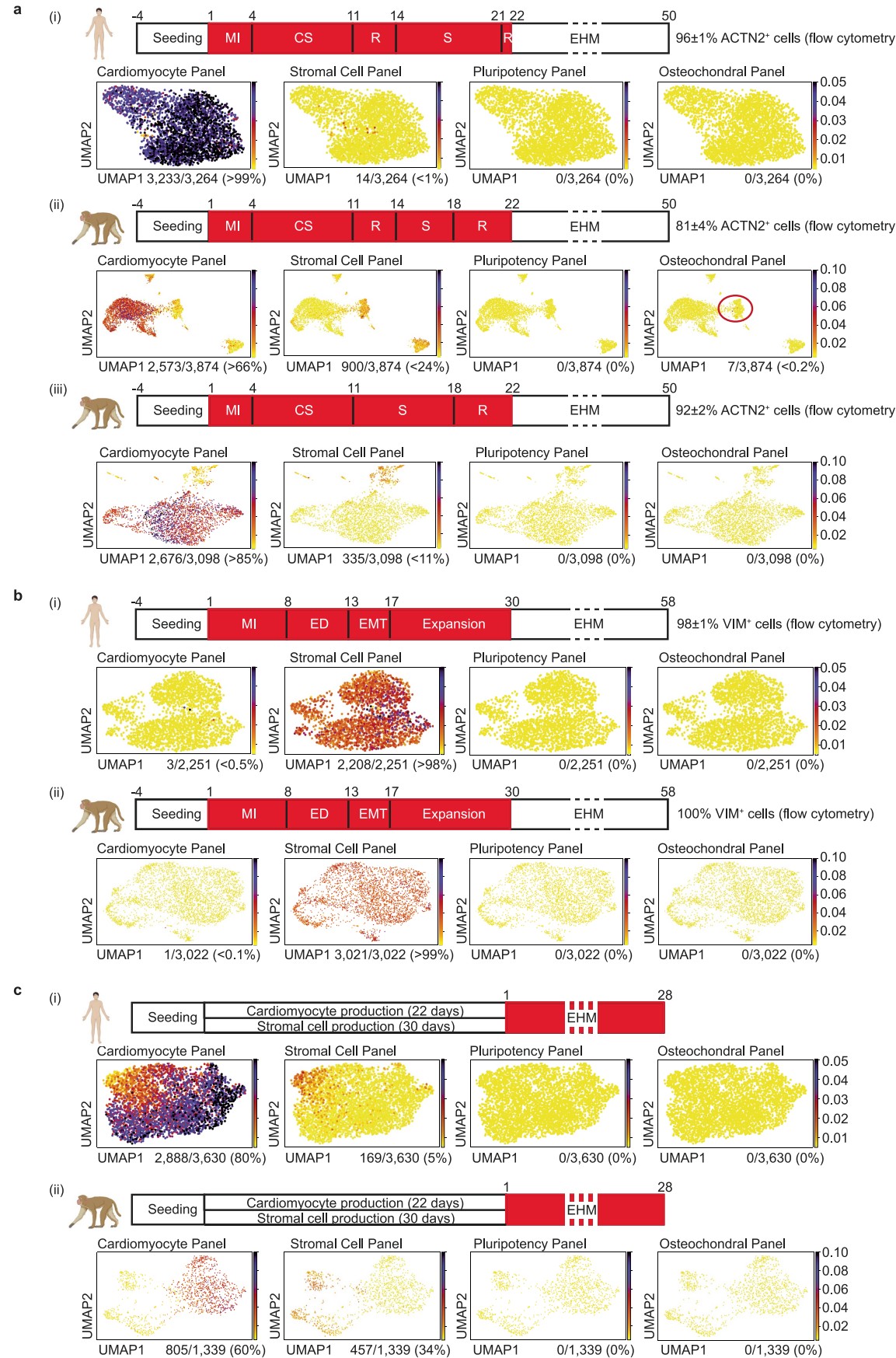

**a**

(i) Seeding | MI | CS | R | S | R | EHM 96±1% ACTN2+ cells (flow cytometry)
-4 1 4 11 14 21 22 50

Cardiomyocyte Panel — UMAP1 3,233/3,264 (>99%)
Stromal Cell Panel — UMAP1 14/3,264 (<1%)
Pluripotency Panel — UMAP1 0/3,264 (0%)
Osteochondral Panel — UMAP1 0/3,264 (0%)

(ii) Seeding | MI | CS | R | S | R | EHM 81±4% ACTN2+ cells (flow cytometry)
-4 1 4 11 14 18 22 50

Cardiomyocyte Panel — UMAP1 2,573/3,874 (>66%)
Stromal Cell Panel — UMAP1 900/3,874 (<24%)
Pluripotency Panel — UMAP1 0/3,874 (0%)
Osteochondral Panel — UMAP1 7/3,874 (<0.2%)

(iii) Seeding | MI | CS | S | R | EHM 92±2% ACTN2+ cells (flow cytometry)
-4 1 4 11 18 22 50

Cardiomyocyte Panel — UMAP1 2,676/3,098 (>85%)
Stromal Cell Panel — UMAP1 335/3,098 (<11%)
Pluripotency Panel — UMAP1 0/3,098 (0%)
Osteochondral Panel — UMAP1 0/3,098 (0%)

**b**

(i) Seeding | MI | ED | EMT | Expansion | EHM 98±1% VIM+ cells (flow cytometry)
-4 1 8 13 17 30 58

Cardiomyocyte Panel — UMAP1 3/2,251 (<0.5%)
Stromal Cell Panel — UMAP1 2,208/2,251 (>98%)
Pluripotency Panel — UMAP1 0/2,251 (0%)
Osteochondral Panel — UMAP1 0/2,251 (0%)

(ii) Seeding | MI | ED | EMT | Expansion | EHM 100% VIM+ cells (flow cytometry)
-4 1 8 13 17 30 58

Cardiomyocyte Panel — UMAP1 1/3,022 (<0.1%)
Stromal Cell Panel — UMAP1 3,021/3,022 (>99%)
Pluripotency Panel — UMAP1 0/3,022 (0%)
Osteochondral Panel — UMAP1 0/3,022 (0%)

**c**

(i) Seeding | Cardiomyocyte production (22 days) / Stromal cell production (30 days) | EHM
1 28

Cardiomyocyte Panel — UMAP1 2,888/3,630 (80%)
Stromal Cell Panel — UMAP1 169/3,630 (5%)
Pluripotency Panel — UMAP1 0/3,630 (0%)
Osteochondral Panel — UMAP1 0/3,630 (0%)

(ii) Seeding | Cardiomyocyte production (22 days) / Stromal cell production (30 days) | EHM
1 28

Cardiomyocyte Panel — UMAP1 805/1,339 (60%)
Stromal Cell Panel — UMAP1 457/1,339 (34%)
Pluripotency Panel — UMAP1 0/1,339 (0%)
Osteochondral Panel — UMAP1 0/1,339 (0%)

**Extended Data Fig. 1** | See next page for caption.

**Extended Data Fig. 1 | Overview of differentiation protocols with single nucleus RNA-sequencing and flow cytometry data confirming cell purity in cardiomyocyte and stromal cell differentiations from human and rhesus iPSC as well as EHM thereof. a**, Cardiomyocyte differentiations: **(i)** human cGMP-protocol used in BioVAT-HF clinical trial (ClinTrial.gov registration: NCT04396899) for comparison, **(ii)** rhesus macaque protocol used in Cohorts 1 and 2; circle highlights identified osteochondral cells (7 of 3,874), and **(iii)** rhesus macaque protocol used in Cohort 3. Additional flow cytometry data (mean ± s.e.m.) from **(i)** n = 23, **(ii)** n = 10, and **(iii)** n = 7 differentiations.

**b**, Stromal cell differentiations: **(i)** human cGMP-protocol used in BioVAT-HF for comparison and **(ii)** rhesus macaque protocol used in Cohorts 1–3. Additional flow cytometry data (mean ± s.e.m.) from **(i)** n = 3 and **(ii)** n = 2 differentiations. **c**, Cell composition of EHM used for implantation in the BioVAT-HF clinical trial **(i)** and Cohort 3 of the rhesus macaque study **(ii)**. Flow cytometry data are displayed in Fig. 1b. MI: mesoderm induction; CS: cardiac specification; R: recovery; S: selection. Illustrations created using BioRender (https://biorender.com).

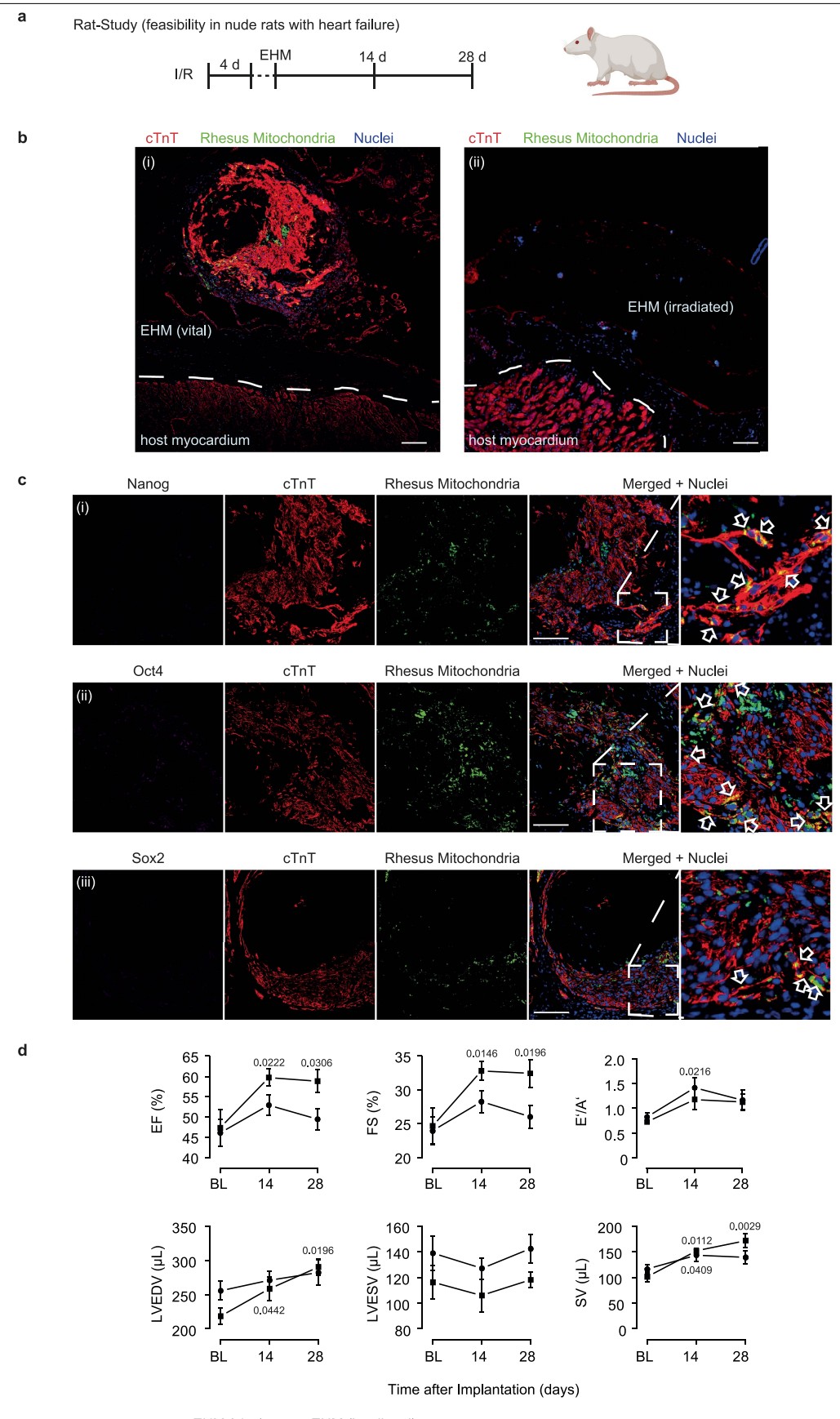

**Extended Data Fig. 2** | See next page for caption.

**Extended Data Fig. 2 | Rat feasibility study. a**, Schematic of the rat feasibility study protocol. **b**, Overview of graft-host interface 28 days after implantation of viable **(i)** and lethally irradiated **(ii)** rhesus EHM 4 days after ischemia/reperfusion (I/R) injury with immunofluorescent labelling of cardiac troponin T (cTnT), primate mitochondria to distinguish rhesus (graft) from rat (host) cells, and nuclei. **c**, Immunofluorescence staining for pluripotency markers **(i)** NANOG, **(ii)** OCT4, and **(iii)** SOX2 with co-labelling of cardiomyocytes (cTnT), rhesus mitochondria (graft), and nuclei did not identify residual stem cells in the EHM grafts. Arrows highlight engrafted cardiomyocytes (co-labelling for cTnT and rhesus mitochondria). Micrographs in **b**, **c** display representative data from 7 animals implanted with vital and 8 animals implanted with irradiated EHM. Bars: 100 µm. **d**, Summary of echocardiography data obtained 14 and 28 days after EHM implantation compared to baseline (BL) data obtained before I/R. Display of mean ± s.e.m. from n = 7 and n = 8 rats implanted with vital and irradiated EHM, respectively. EF: ejection fraction, FS: fractional shortening, E'/A': diastolic function, LVEDV: left ventricular end-diastolic volume, LVESV: left ventricular end-systolic volume, SV: stroke volume. Exact $P$ values versus BL calculated by 2-way ANOVA (repeated measures) with Greenhouse-Geisser correction and Dunnett's multiple comparison testing are presented. Illustration in **a** created using BioRender (https://biorender.com).

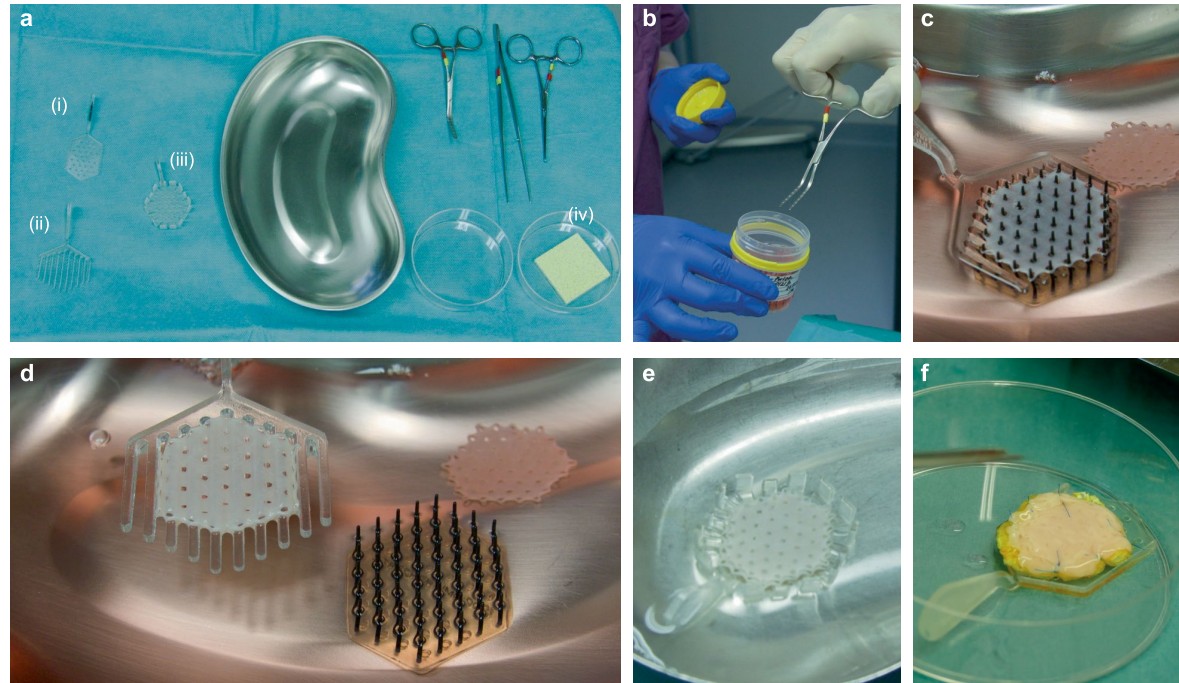

**Extended Data Fig. 3 | EHM assembly. a**, EHM assembly tools at the point-of-care included standard surgical instruments as well as custom-made 3D-printed (i) shovel, (ii) fork and (iii) assembly device for EHM handling and positioning on a (iv) TachoSil™ membrane. **b**, Removal of EHM with holder from a transport container. **c**, Lift-off of EHM from holder with 3D printed fork. **d**, EHM on fork (left) and freely floating (right) after removal from an EHM holder (middle). **e**, 2x EHM stack in assembly device. **f**, 2x EHM sutured to a TachoSil™ membrane before hand-over to the cardiothoracic surgeon for implantation (Extended Data Fig. 4b).

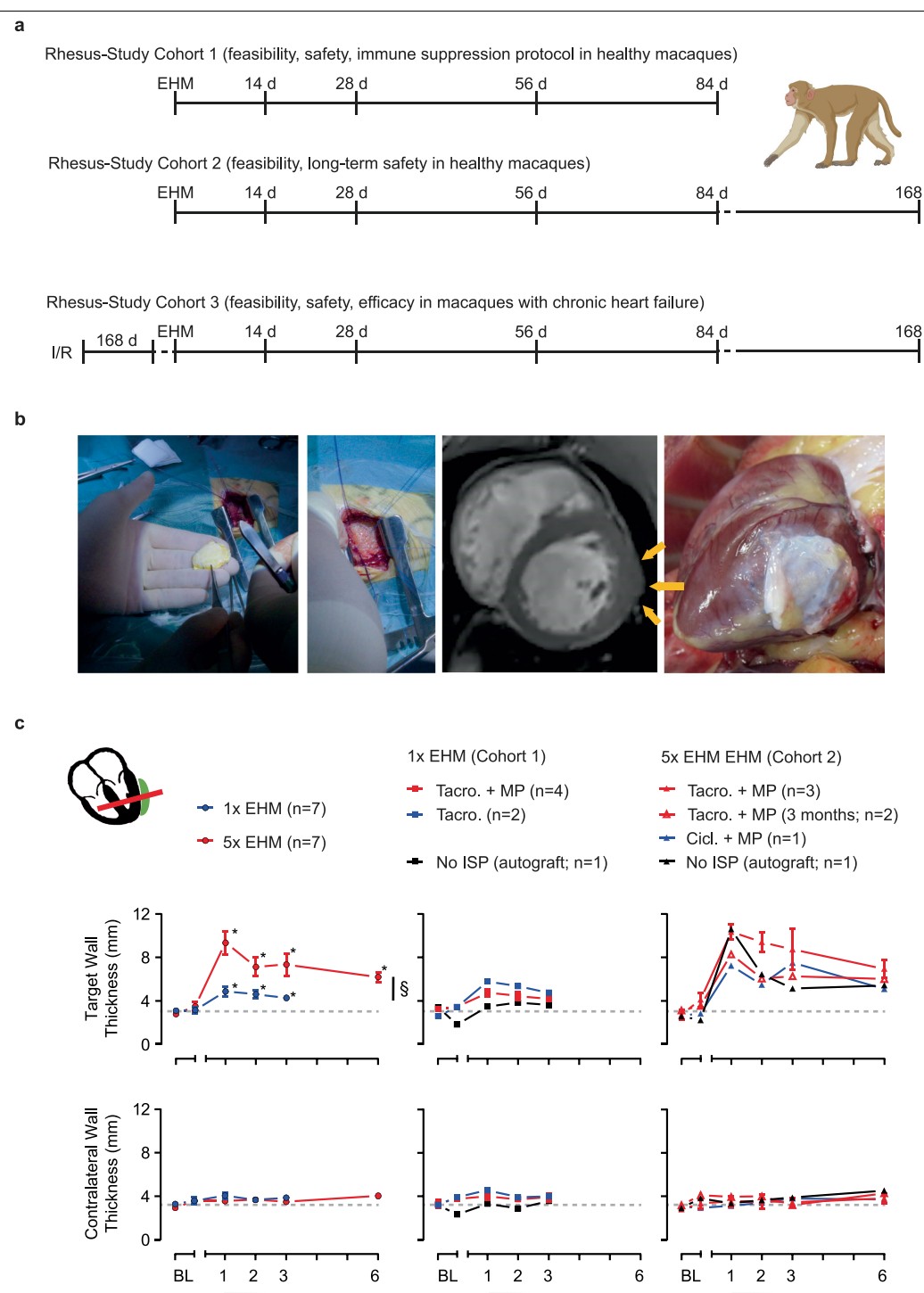

**Extended Data Fig. 4 | EHM allografts augment target heart wall thickness in a dose-dependent manner. a**, Schematic of the rhesus macaque study protocols for Cohorts 1–3. **b**, From left to right: photographs of the EHM implantation procedure through a left lateral thoracotomy onto the beating heart of a healthy rhesus macaque (#2444); magnetic resonance image with 1x EHM graft highlighted by arrows 2 month after implantation (refer to Supplementary Video 2); heart explant with 1x EHM graft 3 months after implantation. **c**, Summary of MRI data (mean ± s.e.m. of target heart and contralateral heart wall thickness in diastole) to assess target heart wall augmentation by 1x and 5x EHM. After 2 baseline [BL] recordings, experimental animals were subjected to additional MRI studies 1, 2, 3, and 6 months after implantation. Refer to Supplementary Table 4 for a summary of the obtained MRI data. All MRI data in Cohorts 1 and 2 were analysed in short axis views with an optimal view of the basal to mid-anterolaterally implanted EHM. Left panels: summary of all obtained data; middle and right panels: individual group data from Cohorts 1 (1x EHM – 3-months follow-up) and 2 (5x EHM – 6-months follow-up), respectively. Exact *P* values were calculated using a mixed-effects model with Greenhouse-Geisser correction and Dunnett's multiple comparison testing: * from left to right 1x EHM vs BL: 0.0172, 0.0405, 0.0154 and 5x EHM vs BL: 0.0041, 0.0066, 0.0106; § 1x vs 5x EHM: 0.0054. ISP: immunosuppression, Tacro: tacrolimus, MP: methylprednisolone, Cicl: cyclosporin. Illustration in **a** created using BioRender (https://biorender.com).

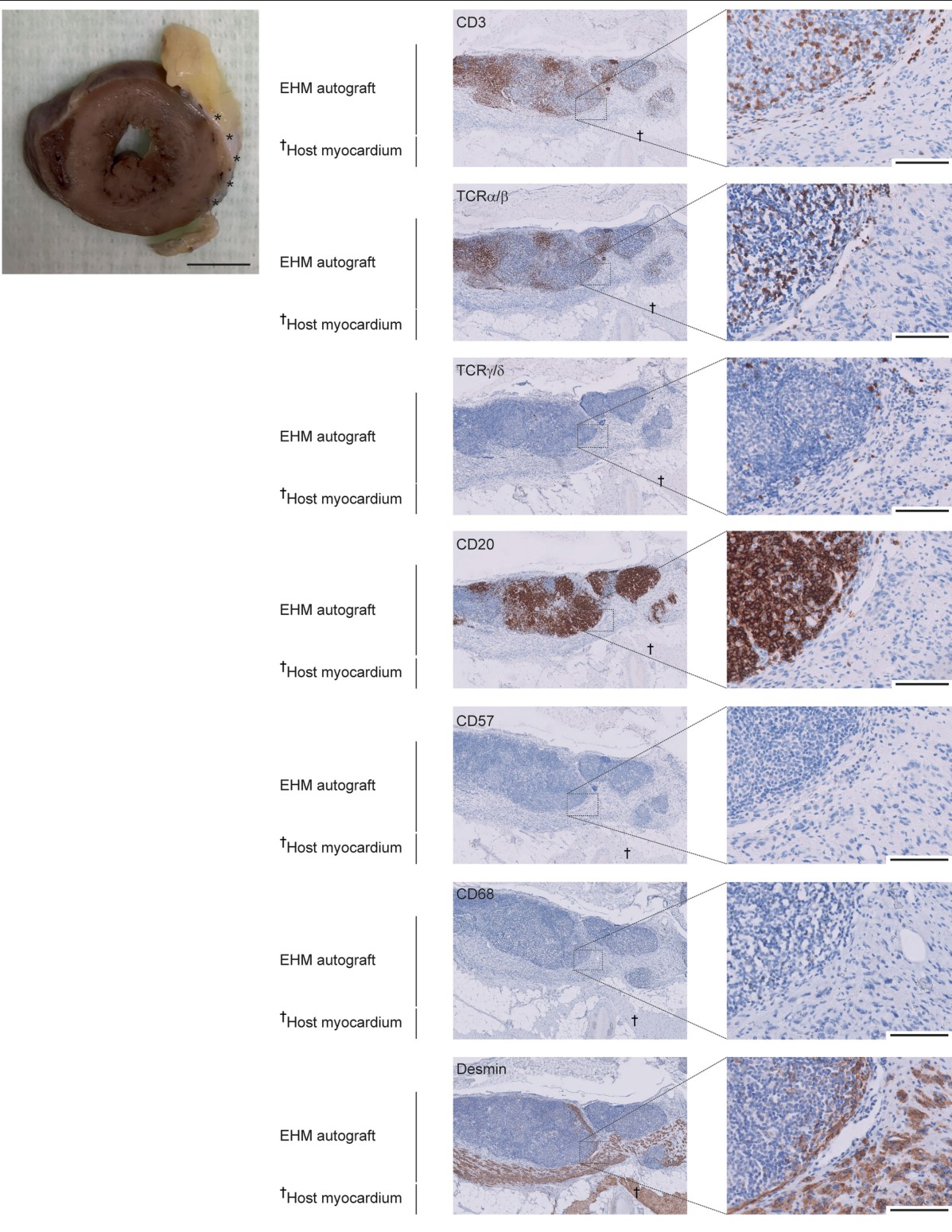

**Extended Data Fig. 5 | T-cell mediated autograft rejection.**
Immunohistochemical staining for T cells (CD3, TCRα/β, TCRγ/δ), B cells (CD20), NK cells (CD57), and macrophages (CD68) with additional staining for cardiomyocytes (desmin) to determine the mode of EHM autograft rejection in experimental animal #2483 (Cohort 1). Macroscopic overview: cross section with rhesus autograft EHM marked with asterisks (from experimental animal #2483; refer to Supplementary Table 3 for details). Scale bars: 10 mm (macroscopic overviews); 100 μm (right panels).

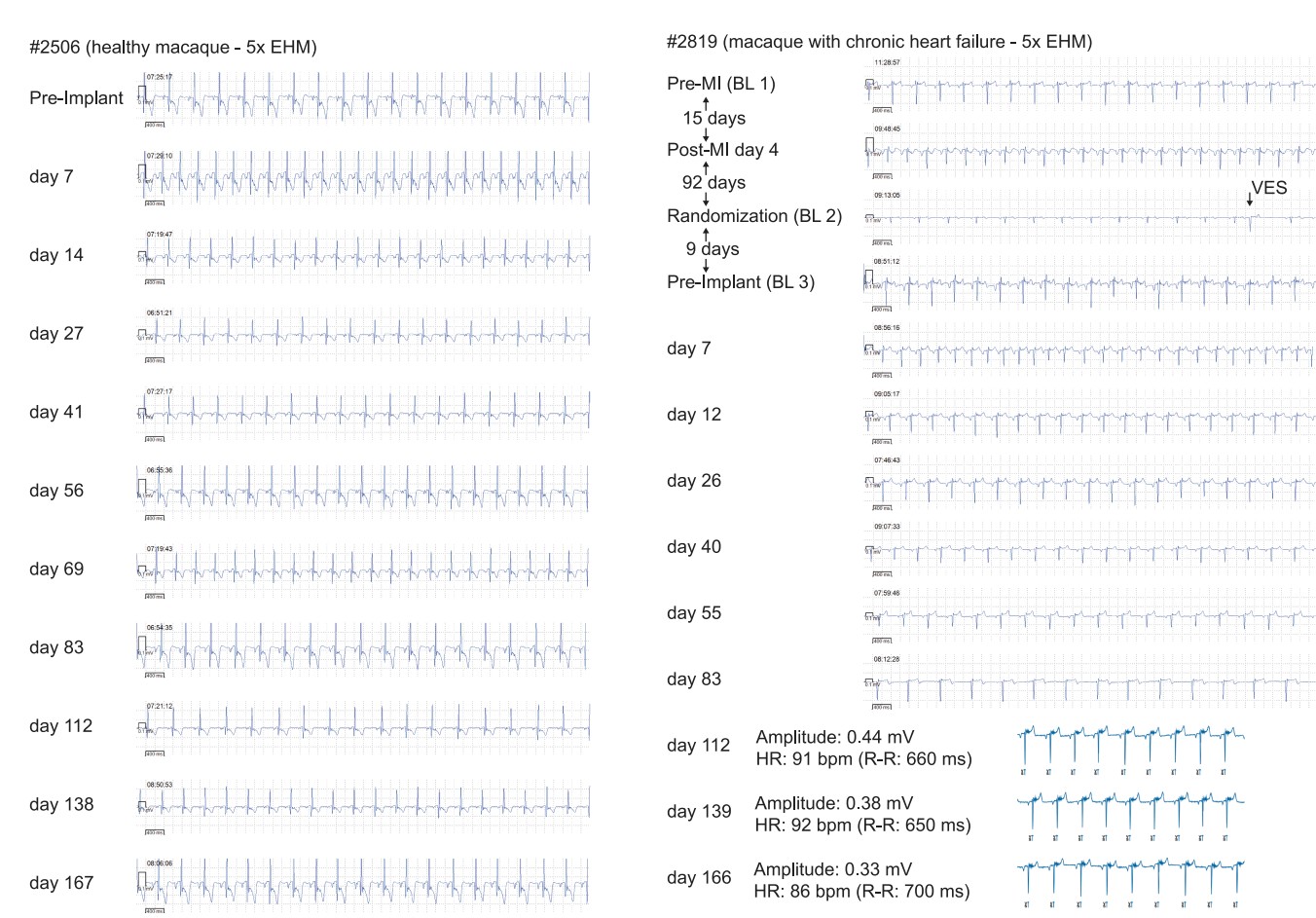

**a**

#2506 (healthy macaque - 5x EHM)

Pre-Implant

day 7

day 14

day 27

day 41

day 56

day 69

day 83

day 112

day 138

day 167

**b**

#2819 (macaque with chronic heart failure - 5x EHM)

Pre-MI (BL 1)

↑ 15 days

Post-MI day 4

↑ 92 days

Randomization (BL 2)

↑ 9 days

Pre-Implant (BL 3)

day 7

day 12

day 26

day 40

day 55

day 83

VES

day 112 — Amplitude: 0.44 mV / HR: 91 bpm (R-R: 660 ms)

day 139 — Amplitude: 0.38 mV / HR: 92 bpm (R-R: 650 ms)

day 166 — Amplitude: 0.33 mV / HR: 86 bpm (R-R: 700 ms)

**Extended Data Fig. 6 | Representative telemetry recordings before and after EHM implantation.** Data from experimental animals #2506 (**a** – Cohort 2) and #2819 (**b** – Cohort 3) implanted with 5x EHM under immune suppression with tacrolimus and methylprednisolone. A ventricular extrasystole (VES) at the day of randomization, i.e., 96 days after myocardial infarction (MI) and 9 days before EHM implantation is highlighted. Note that the traces recorded on days 112, 139, and 166 post EHM implantation (in #2819) are displayed in a different format due to a change in the telemetry software package.

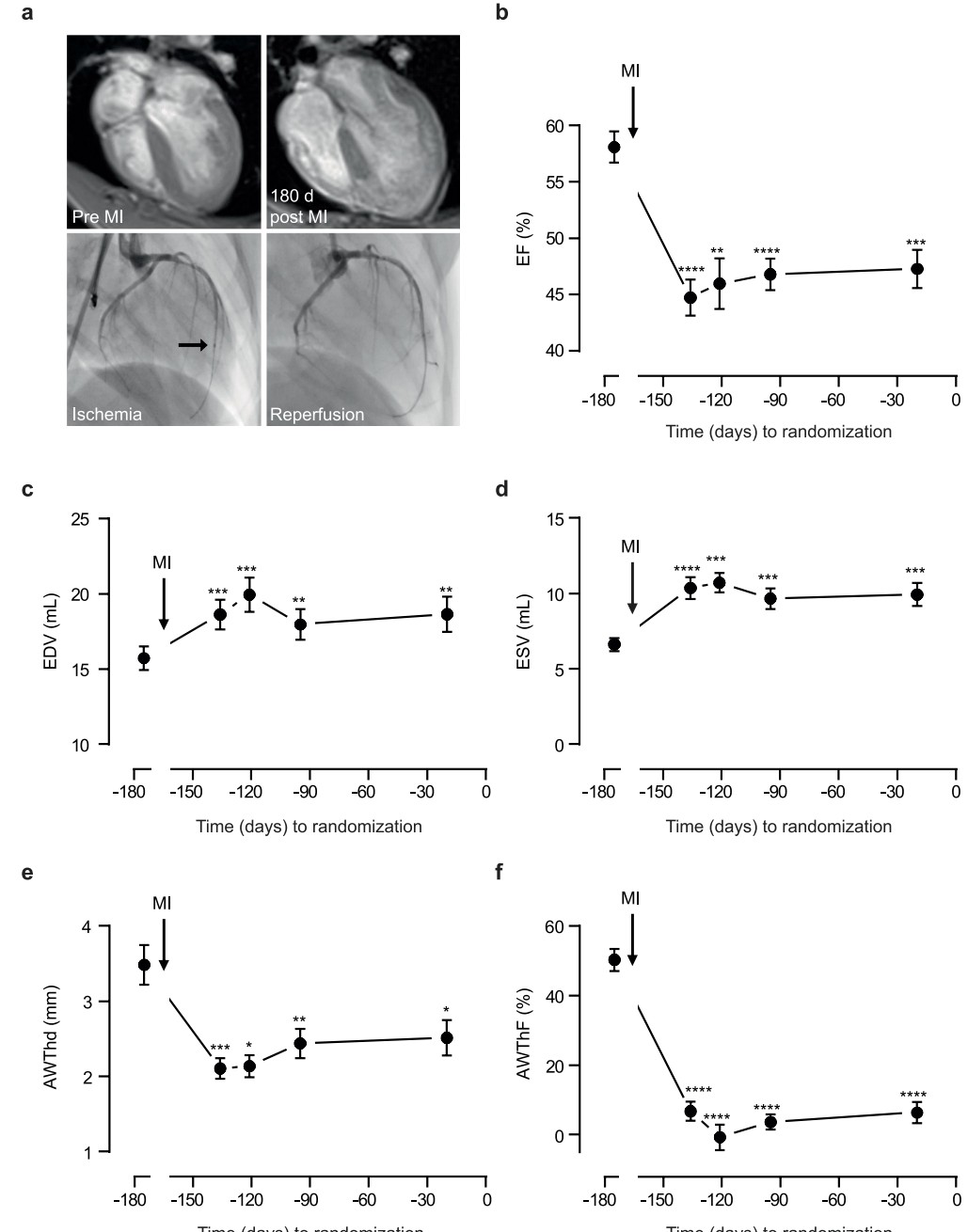

**Extended Data Fig. 7 | Chronic heart failure induced by ischemia/reperfusion injury. a**, MRIs (top panels) pre- and 180 days post-myocardial infarction (MI) inflicted by ischaemia/reperfusion (I/R) injury as well as angiographies (bottom panels) demonstrating balloon occlusion (ischaemia; the guide-wire can be detected in occluded artery; arrow points at balloon marker) and reperfusion (after deflation of occluding balloon). **b-f**, Summary of MRI data (mean ± s.e.m.) obtained in n = 13 macaques at the indicated timepoints post-I/R injury to investigate: **b**, global heart function (ejection fraction [EF]); **c**, left ventricular dimensions in diastole (end-diastolic volume [EDV]); **d**, left ventricular dimensions in systole (end-systolic volume [ESV]); **e**, EHM target heart wall structure (anterior wall thickness in diastole [AWThd]); and **f**, EHM target heart wall function (anterior wall thickening fraction [AWThF]). Animals were subsequently randomized for inclusion in the Cohort 3 study groups (refer to Supplementary Table 5 for details). Exact *P* values versus baseline (pre-I/R injury) were calculated by a mixed-effects model with Greenhouse-Geisser correction and Dunnett's multiple comparison testing: from left to right (**b**) <0.0001, 0.0016, <0.0001, 0.0001; (**c**) 0.0003, 0.0005, 0.0017, 0.0030; (**d**) <0.0001, 0.0003, 0.0001, 0.0001; (**e**) 0.0009, 0.0145, 0.0037, 0.0135; (**f**) <0.0001, <0.0001, <0.0001, <0.0001.

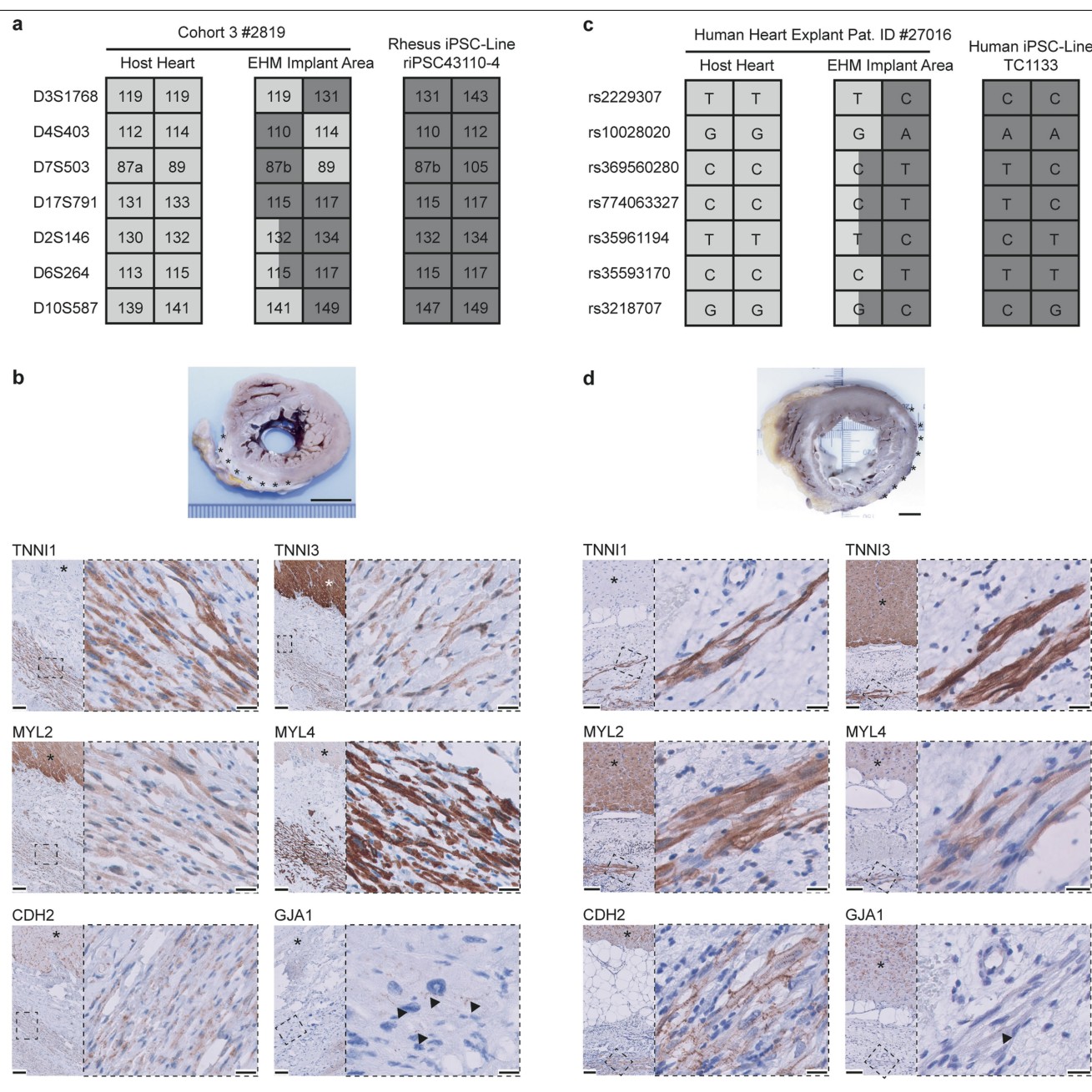

**Extended Data Fig. 8 | EHM graft identity and engrafted cardiomyocyte phenotype. a,b**, Rhesus macaque (#2819); **c,d**, BioVAT-HF patient (#27016) data. **a**, Microsatellite analyses performed in microdissected FFPE-samples obtained from remote host myocardium and desmin-positive EHM implant area (#2819 with riPSC43110-4 EHM allograft). Lengths of microsatellite markers are indicated in base pairs (bp). Light grey indicates host heart alleles; dark grey indicates iPSC/EHM implant alleles. **c**, Deep sequencing of a multigene panel containing 78 genes was performed in microdissected FFPE-samples obtained from remote host myocardium and desmin-positive EHM implant area of the patient's heart explant (displayed in Fig. 5). Single nucleotide variants (SNVs) were detected compared to SNVs present in exome sequencing data of the iPSC master cell line (TC1133). SNVs that were detected in a homozygous wild-type state in the host heart samples were further analysed.

Allele distribution of 7 distinct SNVs showed the presence of iPSC-derived alleles in biopsies of the patched area. Note that host cell infiltration (for example vascular cells) contribute to the expected mixed microsatellite and SNV patterns in the EHM implant area. **b**, Rhesus (#2819) and **d**, human allograft cardiomyocyte phenotype characterized by immunohistochemistry for slow skeletal (fetal) troponin I (TNNI1) and cardiac (adult) troponin I (TNNI3), ventricular (MYL2) and immature/atrial (MYL4) myosin light chain as well as N-cadherin (CHD2) and connexin 43 (GJA1) as markers for intercalated disc and gap junction formation. Asterisks label host myocardium. Arrow point at putative connexin 43 positive gap junction. EHM allografts are highlighted with asterisks in the respective macroscopic overviews. Scale bars: 10 mm (macroscopic overviews); 100 µm (low power magnifications); 20 µm (high power magnifications of boxed regions).

**a**  Control: no immunosuppression / no EHM

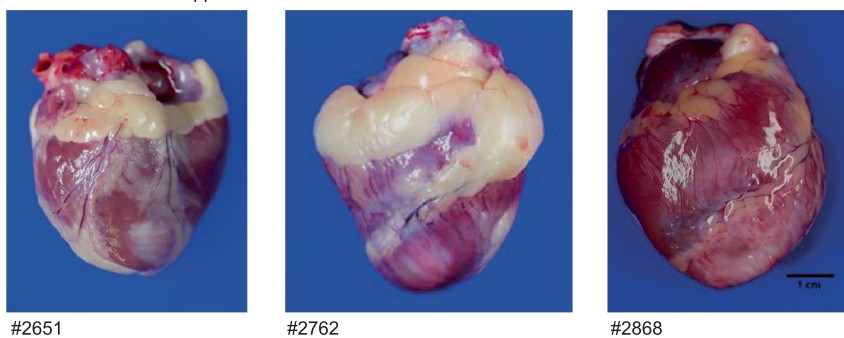

#2651          #2762          #2868

**b**  Control: immunosuppression (Tacrolimus + Methylprednisolone) / no EHM

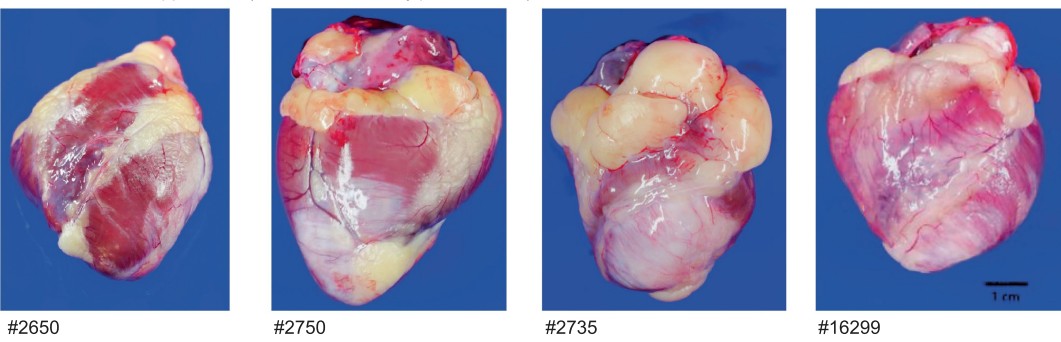

#2650          #2750          #2735          #16299

**c**  2x EHM / immunosuppression (Tacrolimus + Methylprednisolone)

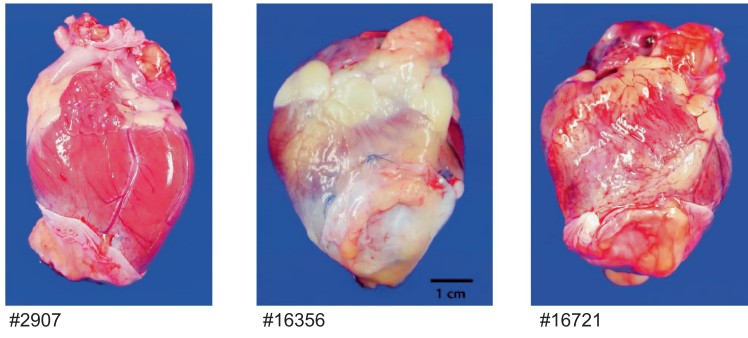

#2907          #16356          #16721

**d**  5x EHM / immunosuppression (Tacrolimus + Methylprednisolone)

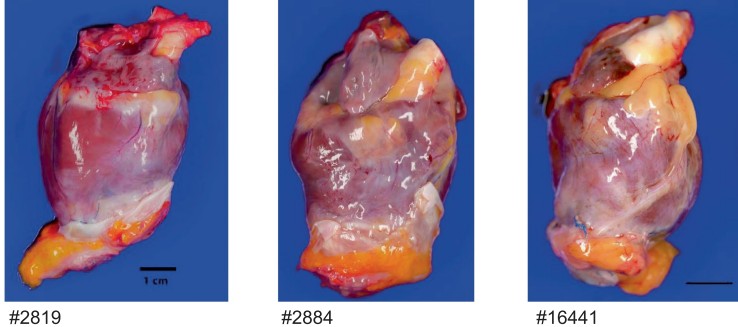

#2819          #2884          #16441

**Extended Data Fig. 9 | Cohort 3 heart explants.** Hearts were explanted and photographed 6 months after randomization into the study groups: **a**, controls without immune suppression (no ISP); **b**, controls with immune suppression (ISP); **c**, NHP with 2x EHM implants (2x EHM); and **d**, NHP with 5x EHM implants (5x EHM).

**46 y/o female**

**Medical Diagnoses:**
Coronary Heart Disease
Myocardial infarction in 2016
Heart Failure (NYHA III)
Type 2 Diabetes
Hypertension
Dyslipidemia
Kidney failure (eGFR 30-59 mL/min)

**Guideline-directed Heart Failure Therapy:**
Bisoprolol            (Beta-Adrenoceptor-Blocker)
Eplerenone            (Mineralcotricoid Receptor Antagonist)
Valsartan / Sacubitril   (Angiotensin-Receptor-Neprilysin-Inhibitor)
Dapagliflozin         (Sodium-Glucose-Transporter 2 Inhibitor)

Torasemid
ASS
Atorvastatin

ICD (Internal Cardiac Defibrillator)

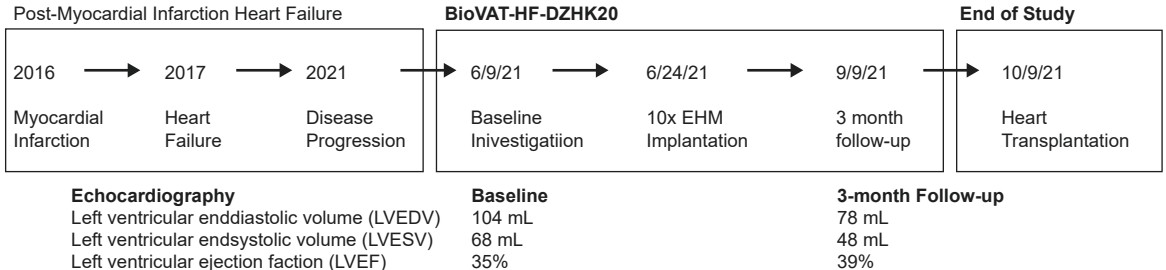

| Post-Myocardial Infarction Heart Failure | | | BioVAT-HF-DZHK20 | | | End of Study |
|---|---|---|---|---|---|---|
| 2016 → | 2017 → | 2021 | 6/9/21 → | 6/24/21 → | 9/9/21 | 10/9/21 |
| Myocardial Infarction | Heart Failure | Disease Progression | Baseline Inivestigatiion | 10x EHM Implantation | 3 month follow-up | Heart Transplantation |

| Echocardiography | Baseline | 3-month Follow-up |
|---|---|---|
| Left ventricular enddiastolic volume (LVEDV) | 104 mL | 78 mL |
| Left ventricular endsystolic volume (LVESV) | 68 mL | 48 mL |
| Left ventricular ejection faction (LVEF) | 35% | 39% |

| HLA-Typing (iPSC-line): | | HLA-Typing (Patient): | |
|---|---|---|---|
| HLA-A | *02, *03 | HLA-A | *02, *24 |
| HLA-B | *07, *39 | HLA-B | *15:01, *35 |
| HLA-C | *07, *15 | HLA-C | *03:03:01G, *04 |
| HLA-DQB1 | *04, *08 | HLA-DQB1 | *05, *06 |
| HLA-DRB1 | *03, *04 | HLA-DRB1 | *10, *13 |

**Immunosuppression:**
| Tacrolimus: | 2-3 mg, b.i.d |
|---|---|
| Average trough levels | 15±2 ng/mL |
| | |
| Methylprednisolone | 12 mg q.d. |
| Target concentration | 0.15 mg/kg |

**Extended Data Fig. 10 | BioVAT-HF patient information.** Summary of the presented BioVAT-HF patient diagnosis and co-morbidities, guideline-directed therapy, disease trajectory and BioVAT study participation until heart transplantation designated as end of study (EOS), echocardiography findings, human leukocyte antigen (HLA) information from patient and allograft as well as immune suppression regimen.

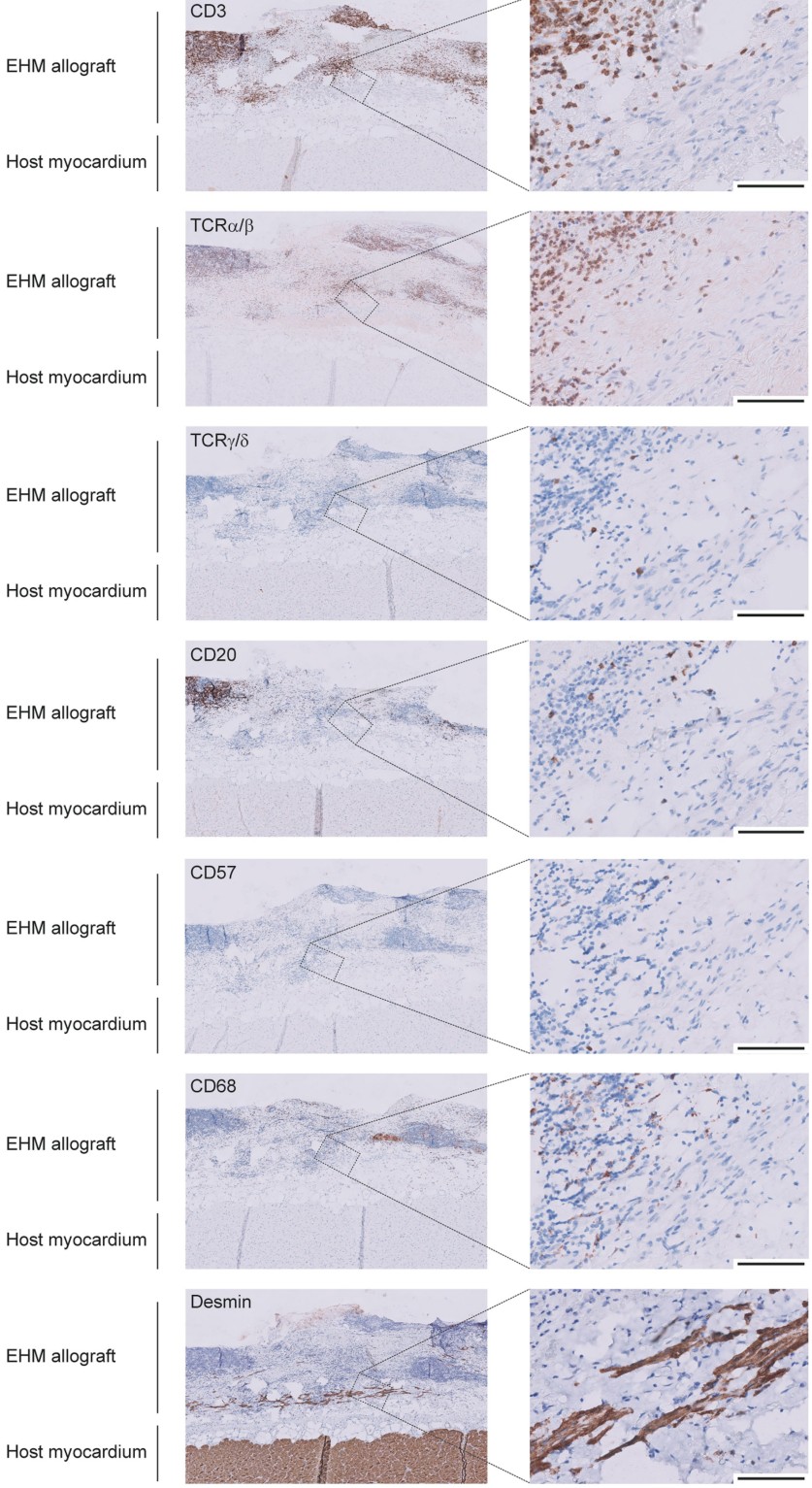

**Extended Data Fig. 11 | Immune cell infiltration in human allograft.**
Immunohistochemical staining for T cells (CD3, TCRα/β, TCRγ/δ), B cells (CD20), NK cells (CD57), and macrophages (CD68) with additional staining for cardiomyocytes (desmin) to investigate immune cell infiltration in the human allograft at the timepoint of heart transplantation. Note that the heart is from a patient implanted with a mid-dose level EHM allograft (10x EHM; refer also to Fig. 5a, Extended Data Fig. 8d). Scale bars: 100 µm.

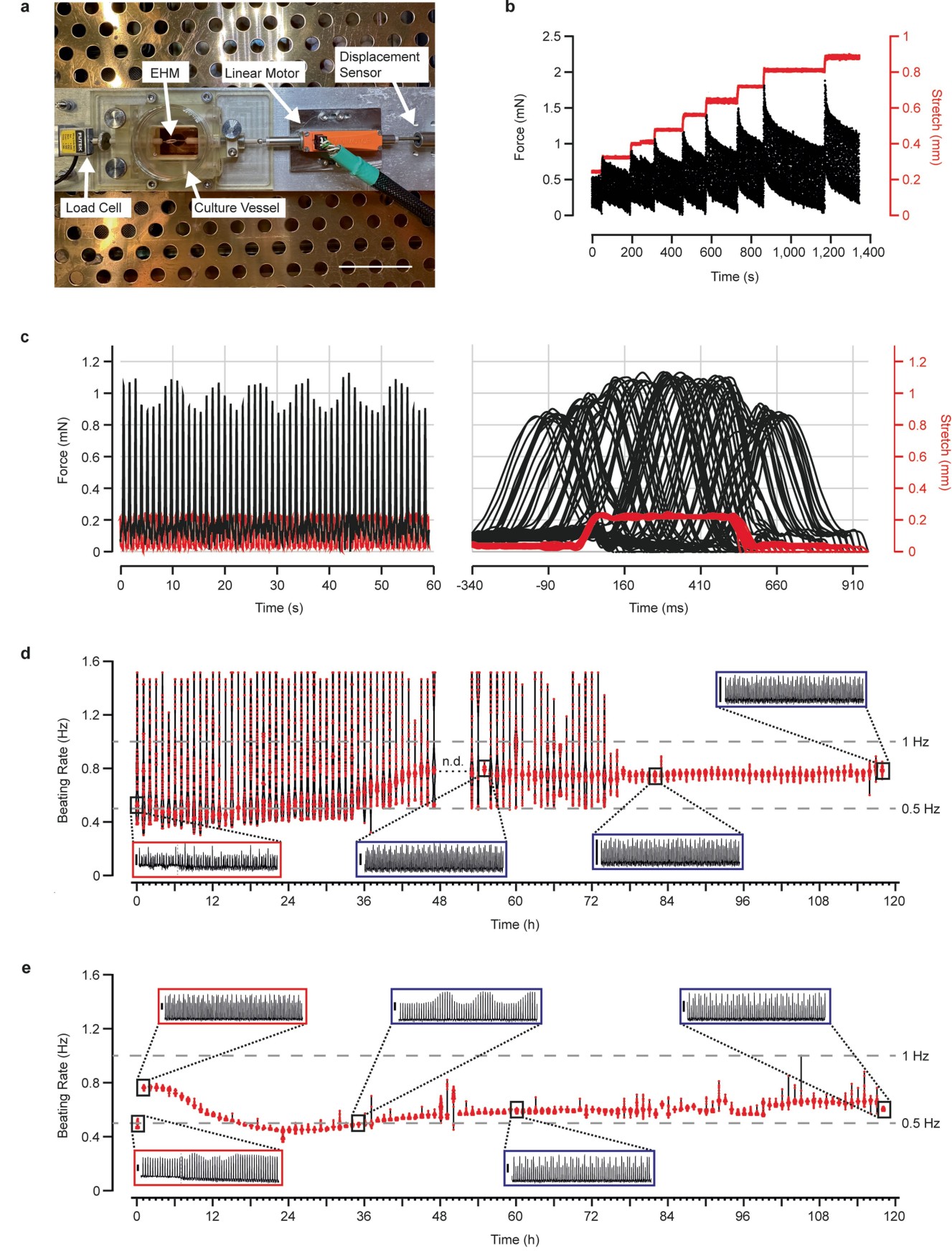

**Extended Data Fig. 12** | See next page for caption.

**Extended Data Fig. 12 | Sensing of mechanical stimuli. a**, Set-up for EHM culture under chronic (120 h) mechanical stimulation. Scale bar: 5 cm. **b**, Stepwise preloading (red traces) resulted in enhancement of contractile force (black traces) according to the Frank-Starling mechanism (positive force-length relationship). **c**, EHM respond to increasing preload on a beat-to-beat basis in spontaneously contracting EHM: (left) continuous recordings (2 min) of EHM contractions under cyclic mechanical stretch (200 µm stretch for 500 ms at a 1 Hz cycle), (right) superimposition of contractions (black) in relation to the 500 ms mechanical stimulus (red). **d**, **e**, Violin plots with individual data points showing the spontaneous beating frequency of two independent EHM recorded over 2 min intervals of every hour during the time course of 120 h in culture. Beating frequency was calculated based on the beat-to-beat time interval of individual contractions (58–117 per timepoint in **d**) and (48–90 per timepoint - **e**). Red and blue boxes depict the respectively indicated contraction traces recorded over the 2 min: in **d** mechanical conditioning resulted in conversion of arrhythmic contractility (red box) into a stable rhythm at ~0.8 Hz over time (blue boxes); in **e** mechanical conditioning resulted in a fast (after 1 h) adaptation of spontaneous beating from $0.49 \pm 0.02$ to $0.76 \pm 0.01$ Hz with subsequent reduction to $0.49 \pm 0.01$ Hz followed by an adaption to the mechanical stimulation by slowly increasing beating rate ($0.61 \pm 0.01$ Hz at the end of the study). Grey striped bars in 0 h boxes indicate the starting points of the mechanical stimulation protocols. n.d.: not determined (data not recorded).

# Reporting Summary

## Statistics

For all statistical analyses, confirm that the following items are present in the figure legend, table legend, main text, or Methods section.

| n/a | Confirmed | |
|---|---|---|
| ☐ | ☒ | The exact sample size (*n*) for each experimental group/condition, given as a discrete number and unit of measurement |
| ☐ | ☒ | A statement on whether measurements were taken from distinct samples or whether the same sample was measured repeatedly |
| ☐ | ☒ | The statistical test(s) used AND whether they are one- or two-sided *Only common tests should be described solely by name; describe more complex techniques in the Methods section.* |
| ☒ | ☐ | A description of all covariates tested |
| ☐ | ☒ | A description of any assumptions or corrections, such as tests of normality and adjustment for multiple comparisons |
| ☐ | ☒ | A full description of the statistical parameters including central tendency (e.g. means) or other basic estimates (e.g. regression coefficient) AND variation (e.g. standard deviation) or associated estimates of uncertainty (e.g. confidence intervals) |
| ☐ | ☒ | For null hypothesis testing, the test statistic (e.g. *F*, *t*, *r*) with confidence intervals, effect sizes, degrees of freedom and *P* value noted *Give P values as exact values whenever suitable.* |
| ☒ | ☐ | For Bayesian analysis, information on the choice of priors and Markov chain Monte Carlo settings |
| ☒ | ☐ | For hierarchical and complex designs, identification of the appropriate level for tests and full reporting of outcomes |
| ☒ | ☐ | Estimates of effect sizes (e.g. Cohen's *d*, Pearson's *r*), indicating how they were calculated |

*Our web collection on statistics for biologists contains articles on many of the points above.*

## Software and code

Policy information about availability of computer code

| Data collection | BD FACSDiva Software (Version 6.1.3), Kaluza 2.2.1 (Beckman), VitroDat 3.52 (Föhr Medical Instruments), Zen 2.3 SP1 (Zeiss) |
|---|---|
| Data analysis | Microsoft Excel 2019 MSO (16.0.10415.20025) 32-Bit, Graph Pad Prism 10.1.2 (64-bit), FlowJo 9.6 software, BD FACSDiva Software (Version 6.1.3), Kaluza 2.2.1 (Beckman), CellRanger (v.3.1.0; 10XGenomics), SeqPilot (JSI medical systems GmbH, Version 5.2.0 Build 505), Varbank 2.0 (Cologne Center for Genomics (CCG); University of Cologne), VitroDat 3.52 (Föhr Medical Instruments), Zen 2.3 SP1 (Zeiss), Segment v4.0 R12067 (Medviso, segment.heiberg.se), Medis-Suite Version 3.2 with QMass module Version 8.1 (Medis) |

For manuscripts utilizing custom algorithms or software that are central to the research but not yet described in published literature, software must be made available to editors and reviewers. We strongly encourage code deposition in a community repository (e.g. GitHub). See the Nature Portfolio guidelines for submitting code & software for further information.

## Data

Policy information about availability of data

All manuscripts must include a data availability statement. This statement should provide the following information, where applicable:
- Accession codes, unique identifiers, or web links for publicly available datasets
- A description of any restrictions on data availability
- For clinical datasets or third party data, please ensure that the statement adheres to our policy

The data sets generated during and/or analysed during the current study are available from the corresponding author. Please refer also to the Source Data

document. The snRNAseq data is publicly accessible under GSE276021 (GEO data base). Gene counts were obtained by aligning reads to the hg38 genome (NCBI:GCA_000001405.22; GRCh38.p7) using CellRanger software (v.3.0.2; 10XGenomics).

## Human research participants

Policy information about studies involving human research participants and Sex and Gender in Research.

| | |
|---|---|
| Reporting on sex and gender | gender is reported |
| Population characteristics | not appliacable |
| Recruitment | Case report from BioVAT-HF-DZHK20 - EudraCT No. 2019-000885-39 [EU CT No. 2024-515708-38-01] and ClinicalTrials.gov ID NCT04396899 |
| Ethics oversight | The BioVAT-HF-DZHK20 Phase I/II clinical trial (ClinicalTrial.gov NCT04396899) was approved by the responsible regulatory agency (Paul-Ehrlich-Institute) and the competent ethics committee (ethics committee of the University Medical Center Göttingen under the file #18/7/20). |

Note that full information on the approval of the study protocol must also be provided in the manuscript.

# Field-specific reporting

Please select the one below that is the best fit for your research. If you are not sure, read the appropriate sections before making your selection.

☒ Life sciences ☐ Behavioural & social sciences ☐ Ecological, evolutionary & environmental sciences

For a reference copy of the document with all sections, see nature.com/documents/nr-reporting-summary-flat.pdf

# Life sciences study design

All studies must disclose on these points even when the disclosure is negative.

| | |
|---|---|
| Sample size | Sample size (n=7 Cohort 1 ; n=7 in Cohort 2; n=20 in Cohort 3) was chosen based on previous experience and taking in account 3R considerations. The adaptive study design in Cohorts 1 and 2 informed choice of immune suppression and dosing in Cohort 3. Refer to Supplementary Table 3 for an overview of all Rhesus macaques included in the study. |
| Data exclusions | No data was excluded |
| Replication | Data was replicated in Cohorts 1 to 3 with adequate groups sizes of 7 (Cohort 1), 7 (Cohort 2) and 20 (Cohort 3 - 1 animal was not allowed to be included in the implantation study due to low body weight, 5 animals died post myocardial infarction, 1 animal died upon weaning from anesthesia after implantation of a 5x EHM). |
| Randomization | Rat study: animals were assigned radomly to the experiemtal groups (with vital or irradiated EHM implant). Rhesus macaque study: allograft animals were assigned to the different study groups by coin flip. |
| Blinding | With the exception of the surgeons (blinding is not possible), investigators were blinded to the study protocol. Recording of MRI data were performed by investigators blinded to the treatment condition. MRI image analysis of Cohorts 1 and 2 were performed by 2 independent observers. Investigations of MRI data of Cohorts 3 were performed by 2 additional independent observers.  Pathological analyses were performed by as to the treatment condition blinded pathologists. |

# Behavioural & social sciences study design

All studies must disclose on these points even when the disclosure is negative.

| | |
|---|---|
| Study description | *Briefly describe the study type including whether data are quantitative, qualitative, or mixed-methods (e.g. qualitative cross-sectional, quantitative experimental, mixed-methods case study).* |
| Research sample | *State the research sample (e.g. Harvard university undergraduates, villagers in rural India) and provide relevant demographic information (e.g. age, sex) and indicate whether the sample is representative. Provide a rationale for the study sample chosen. For studies involving existing datasets, please describe the dataset and source.* |
| Sampling strategy | *Describe the sampling procedure (e.g. random, snowball, stratified, convenience). Describe the statistical methods that were used to predetermine sample size OR if no sample-size calculation was performed, describe how sample sizes were chosen and provide a rationale for why these sample sizes are sufficient. For qualitative data, please indicate whether data saturation was considered, and what criteria were used to decide that no further sampling was needed.* |

| Data collection | Provide details about the data collection procedure, including the instruments or devices used to record the data (e.g. pen and paper, computer, eye tracker, video or audio equipment) whether anyone was present besides the participant(s) and the researcher, and whether the researcher was blind to experimental condition and/or the study hypothesis during data collection. |
|---|---|
| Timing | Indicate the start and stop dates of data collection. If there is a gap between collection periods, state the dates for each sample cohort. |
| Data exclusions | If no data were excluded from the analyses, state so OR if data were excluded, provide the exact number of exclusions and the rationale behind them, indicating whether exclusion criteria were pre-established. |
| Non-participation | State how many participants dropped out/declined participation and the reason(s) given OR provide response rate OR state that no participants dropped out/declined participation. |
| Randomization | If participants were not allocated into experimental groups, state so OR describe how participants were allocated to groups, and if allocation was not random, describe how covariates were controlled. |

# Ecological, evolutionary & environmental sciences study design

All studies must disclose on these points even when the disclosure is negative.

| Study description | Briefly describe the study. For quantitative data include treatment factors and interactions, design structure (e.g. factorial, nested, hierarchical), nature and number of experimental units and replicates. |
|---|---|
| Research sample | Describe the research sample (e.g. a group of tagged Passer domesticus, all Stenocereus thurberi within Organ Pipe Cactus National Monument), and provide a rationale for the sample choice. When relevant, describe the organism taxa, source, sex, age range and any manipulations. State what population the sample is meant to represent when applicable. For studies involving existing datasets, describe the data and its source. |
| Sampling strategy | Note the sampling procedure. Describe the statistical methods that were used to predetermine sample size OR if no sample-size calculation was performed, describe how sample sizes were chosen and provide a rationale for why these sample sizes are sufficient. |
| Data collection | Describe the data collection procedure, including who recorded the data and how. |
| Timing and spatial scale | Indicate the start and stop dates of data collection, noting the frequency and periodicity of sampling and providing a rationale for these choices. If there is a gap between collection periods, state the dates for each sample cohort. Specify the spatial scale from which the data are taken |
| Data exclusions | If no data were excluded from the analyses, state so OR if data were excluded, describe the exclusions and the rationale behind them, indicating whether exclusion criteria were pre-established. |
| Reproducibility | Describe the measures taken to verify the reproducibility of experimental findings. For each experiment, note whether any attempts to repeat the experiment failed OR state that all attempts to repeat the experiment were successful. |
| Randomization | Describe how samples/organisms/participants were allocated into groups. If allocation was not random, describe how covariates were controlled. If this is not relevant to your study, explain why. |
| Blinding | Describe the extent of blinding used during data acquisition and analysis. If blinding was not possible, describe why OR explain why blinding was not relevant to your study. |

Did the study involve field work? ☐ Yes ☐ No

# Field work, collection and transport

| Field conditions | Describe the study conditions for field work, providing relevant parameters (e.g. temperature, rainfall). |
|---|---|
| Location | State the location of the sampling or experiment, providing relevant parameters (e.g. latitude and longitude, elevation, water depth). |
| Access & import/export | Describe the efforts you have made to access habitats and to collect and import/export your samples in a responsible manner and in compliance with local, national and international laws, noting any permits that were obtained (give the name of the issuing authority, the date of issue, and any identifying information). |
| Disturbance | Describe any disturbance caused by the study and how it was minimized. |

# Reporting for specific materials, systems and methods

We require information from authors about some types of materials, experimental systems and methods used in many studies. Here, indicate whether each material, system or method listed is relevant to your study. If you are not sure if a list item applies to your research, read the appropriate section before selecting a response.

nature portfolio | reporting summary

March 2021

## Materials & experimental systems

| n/a | Involved in the study |
|-----|------------------------|
| ☐ | ☒ Antibodies |
| ☐ | ☒ Eukaryotic cell lines |
| ☒ | ☐ Palaeontology and archaeology |
| ☐ | ☒ Animals and other organisms |
| ☐ | ☒ Clinical data |
| ☒ | ☐ Dual use research of concern |

## Methods

| n/a | Involved in the study |
|-----|------------------------|
| ☒ | ☐ ChIP-seq |
| ☐ | ☒ Flow cytometry |
| ☒ | ☐ MRI-based neuroimaging |

## Antibodies

**Antibodies used**
Refer to Supplementary Table 6 for details

**Validation**
Antibodies used in this study were validated in previous studies (e.g., Riegler et al. 2015 Circ Res, Tiburcy et al. 2017 Circulation). Antibodies used in the pathology studies are all validated for veterinarian and clinical use and tested for cross-reactivity to Rhesus macaque samples.

## Eukaryotic cell lines

Policy information about cell lines and Sex and Gender in Research

**Cell line source(s)**
Rhesus iPSC-lines:
iPSC 43110-4: fibroblast obtained from the California National Primate Research Center in Davis were reprogrammed using Sendai Virus mediated deliver of Oct4, Sox2, Nanog, and cMyc as reported in Zhao et al 2018 (https://doi.org/10.1016/j.stemcr.2018.01.002) and Yang et al. 2021 (https://doi.org/10.1093/cvr/cvaa281)

DPZ_iRH34.1 and DPZ_iRH23.1 (from animal #2500 - autograft recipient): fibroblast obtained from the Deutsches Primatenzentrum (German Primate Center) in Göttingen were reprogrammed by nucleofection of three episomal vectors pCXLE-hOCT3/4-shp53-F (encoding for human OCT3/4 and shRNA against p53; Addgene #27077), pCXLE-hSK (encoding for human SOX2 and KLF4; Addgene #27078) and pCXLE-hUL (encoding for human L-MYC and LIN28; Addgene #27080) as reported in Stauske et al. 2020 (doi:10.3390/cells9061349).

DPZ_iRH25.B1 (from animal #2483 - autograft recipient): fibroblast obtained from the Deutsches Primatenzentrum (German Primate Center)  were reprogrammed using Sendai Virus mediated delivery of Oct4, Sox2, Nanog, and cMyc similar as described in Zhao et al 2018 (https://doi.org/10.1016/j.stemcr.2018.01.002) and Yang et al. 2021 (https://doi.org/10.1093/cvr/cvaa281).

Human iPSC-line:
Generation of the GMP line LiPSC-GR1.1 (also referred to as TC1133; lot number 50-001-21) was supported by the NIH Common Fund Regenerative Medicine Program, and reported in Baghbaderani et al. 2015 (doi:10.1016/j.stemcr.2015.08.015).  Further information is available at https://hpscreg.eu/cell-line/RUCDRi002-A. The NIH Common Fund and the National Center for Advancing Translational Sciences (NCATS) are joint stewards of the LiPSC-GR1.1 resource.  A derivative from a GMP master cell bank of the TC1133-line was obtained by Repairon GmbH and was used as starting material for EHM production for the BioVAT-HF-DZHK20 Phase I/II clinical trial.

**Authentication**
By MHC (human) or Mamu (macaque) typing

**Mycoplasma contamination**
All lines were tested to be free from mycoplasma contamination (in-house testing using Lonza MycoAlert® Detection Kit, external testing of GMP cell lines and products by Minerva Biolabs, Berlin, Germany).

**Commonly misidentified lines**
(See ICLAC register)
no commonly misidentified cell lines were used in the study

## Palaeontology and Archaeology

**Specimen provenance**
*Provide provenance information for specimens and describe permits that were obtained for the work (including the name of the issuing authority, the date of issue, and any identifying information). Permits should encompass collection and, where applicable, export.*

| Specimen deposition | *Indicate where the specimens have been deposited to permit free access by other researchers.* |
|---|---|
| Dating methods | *If new dates are provided, describe how they were obtained (e.g. collection, storage, sample pretreatment and measurement), where they were obtained (i.e. lab name), the calibration program and the protocol for quality assurance OR state that no new dates are provided.* |

☐ Tick this box to confirm that the raw and calibrated dates are available in the paper or in Supplementary Information.

| Ethics oversight | *Identify the organization(s) that approved or provided guidance on the study protocol, OR state that no ethical approval or guidance was required and explain why not.* |
|---|---|

Note that full information on the approval of the study protocol must also be provided in the manuscript.

## Animals and other research organisms

Policy information about studies involving animals; ARRIVE guidelines recommended for reporting animal research, and Sex and Gender in Research

| Laboratory animals | Rats: 8-10 weeks old male RNU rats (250-350g) obtained from Charles River Laboratories, Wilmington, MA<br>Rhesus macaques: refer to Supplementary Table 3 for a detailed overview of the experimental animals used in this study. |
|---|---|
| Wild animals | No wild animals were used in the study. |
| Reporting on sex | refer to Supplementary Table 3 for a detailed overview of the experimental animals (Macaca mulatta) used in this study |
| Field-collected samples | No field collected samples were used in the study. |
| Ethics oversight | Animal experiments were approved by the by the Stanford Animal Research Committee (nude rat study at Stanford) and Niedersächsisches Landesamt für Verbraucherschutz und Lebensmittelsicherheit (LAVES; 33.42502-04-15/1807 and -16/2370; Rhesus macaque [Macaca mulatta] studies in Göttingen). |

Note that full information on the approval of the study protocol must also be provided in the manuscript.

## Clinical data

Policy information about clinical studies

All manuscripts should comply with the ICMJE guidelines for publication of clinical research and a completed CONSORT checklist must be included with all submissions.

| Clinical trial registration | EU CT No. 2024-515708-38-01 (previously EudraCT: 2019-000885-39) and ClinicalTrials.gov: NCT04396899 |
|---|---|
| Study protocol | We are reporting a case from the BioVAT-HF-DZHK20 Phase I/II clinical trial. Refer to Supplementary Note 5 for Clinical Reserach Information and the Synopsis of the BioVAT-HF-DZHK20 Clinical Trial Protocol. |
| Data collection | Here we are reporting immunohistochemistry and clinical data from a heart explant patient included in the BioVAT-HF-DZHK20 Phase I/II clinical trial (case report). The release of the data was approved by the Sponsor (University Medical Center Göttingen) and the repsonsible clinical trial statistician (Prof. T. Friede, Institut of Medical Statistics). The clinical trial is ongoing and the full data set will be reported seperately after completion of the clinical trial. |
| Outcomes | iPSC-derived cardiomyocyte allograft retention was determined by histopathology (H&E staining and immunohistochemistry) and deep sequencing of microdissected FFPE-samples followed by sequence alignment to determine host and graft specific single nucleotide variants (SNVs). |

## Dual use research of concern

Policy information about dual use research of concern

### Hazards

Could the accidental, deliberate or reckless misuse of agents or technologies generated in the work, or the application of information presented in the manuscript, pose a threat to:

| No | Yes | |
|---|---|---|
| ☒ | ☐ | Public health |
| ☒ | ☐ | National security |
| ☒ | ☐ | Crops and/or livestock |
| ☒ | ☐ | Ecosystems |
| ☒ | ☐ | Any other significant area |

## Experiments of concern

Does the work involve any of these experiments of concern:

| No | Yes | |
|----|-----|---|
| ☒ | ☐ | Demonstrate how to render a vaccine ineffective |
| ☒ | ☐ | Confer resistance to therapeutically useful antibiotics or antiviral agents |
| ☒ | ☐ | Enhance the virulence of a pathogen or render a nonpathogen virulent |
| ☒ | ☐ | Increase transmissibility of a pathogen |
| ☒ | ☐ | Alter the host range of a pathogen |
| ☒ | ☐ | Enable evasion of diagnostic/detection modalities |
| ☒ | ☐ | Enable the weaponization of a biological agent or toxin |
| ☒ | ☐ | Any other potentially harmful combination of experiments and agents |

# ChIP-seq

## Data deposition

☐ Confirm that both raw and final processed data have been deposited in a public database such as GEO.

☐ Confirm that you have deposited or provided access to graph files (e.g. BED files) for the called peaks.

| | |
|---|---|
| Data access links<br>*May remain private before publication.* | *For "Initial submission" or "Revised version" documents, provide reviewer access links. For your "Final submission" document, provide a link to the deposited data.* |
| Files in database submission | *Provide a list of all files available in the database submission.* |
| Genome browser session<br>(e.g. UCSC) | *Provide a link to an anonymized genome browser session for "Initial submission" and "Revised version" documents only, to enable peer review. Write "no longer applicable" for "Final submission" documents.* |

## Methodology

| | |
|---|---|
| Replicates | *Describe the experimental replicates, specifying number, type and replicate agreement.* |
| Sequencing depth | *Describe the sequencing depth for each experiment, providing the total number of reads, uniquely mapped reads, length of reads and whether they were paired- or single-end.* |
| Antibodies | *Describe the antibodies used for the ChIP-seq experiments; as applicable, provide supplier name, catalog number, clone name, and lot number.* |
| Peak calling parameters | *Specify the command line program and parameters used for read mapping and peak calling, including the ChIP, control and index files used.* |
| Data quality | *Describe the methods used to ensure data quality in full detail, including how many peaks are at FDR 5% and above 5-fold enrichment.* |
| Software | *Describe the software used to collect and analyze the ChIP-seq data. For custom code that has been deposited into a community repository, provide accession details.* |

# Flow Cytometry

## Plots

Confirm that:

☒ The axis labels state the marker and fluorochrome used (e.g. CD4-FITC).

☒ The axis scales are clearly visible. Include numbers along axes only for bottom left plot of group (a 'group' is an analysis of identical markers).

☒ All plots are contour plots with outliers or pseudocolor plots.

☒ A numerical value for number of cells or percentage (with statistics) is provided.

## Methodology

| | |
|---|---|
| Sample preparation | Flow cytomry analysis of EHM cell composition. EHM were washed in PBS and dissociated in 2 mg/ml Collagenase 1 (Sigma-Aldrich) in PBS with 20% FBS at 37°C for 1 h followed by Accutase (Millipore), 0.025% Trypsin (ThermoScientific) and 20 µg/ml DNAse I (Calbiochem) at 20-24°C for 30 min. After fixation in 70% ice cold EtOH for >10 min, cells were either exposed to primary antibody directed against sarcomeric actinin (ACTN2: 1:4,000; A7811, Sigma) or vimentin (VIM; 1:1,000; ab92547, abcam) in blocking buffer for 45 min followed by secondary antibodies in blocking buffer and Hoechst 33342 for 30 min at 4° |

C (Supplementary Table 6). Control samples were exposed to undirected IgG1 (MAB002; R&D Systems). Human samples were fixed with 4% formalin and exposed to conjugated antibodies directed against sarcomeric actinin (ACTN2-PE, 1:1000, 130-106-937, Miltenyi Biotec) and vimentin (VIM-AF647, 1:1000, Biolegend, 677807) for 15 min at 4°C. A BD LSRII SORP system (BD Biosciences) or CytoFLEX (Beckman/Coulter) was used for flow cytometry analysis.

Donor-specific antibody analysis. iPSC-derived cardiomyocytes (CMs) and stromal cells (StCs), unstimulated and after IFNγ (100 ng/mL for 48 h) were exposed to sera obtained before (pre) and at the indicated timepoints during the study at different dilutions (1:5 to 1:40). A FITC-labeled anti-Rhesus IgG antibody (4700-02, Southern Biotech; Birmingham, AL, USA) was used to detect antibodies in the sera bound to the CMs and StCs. The cell mean fluorescence intensity (MFI) and the proportion of stained cells were determined by flow cytometry (LSR II SORP, BD Biosciences). Antibodies that display a selective reactivity to IFNγ stimulated CMs presumably include DSAs to MHC class I molecules. The pan-HLA antibody W6/32 (Biolegend, San Diego, CA, USA), which reacts with MHC class I molecules of rhesus macaques was used to demonstrate the expression of these molecules on CMs and StCs.

Flow Cytometry analysis of peripheral immune cells. 50 µl of whole blood were stained with a mixture of pre-titrated monoclonal antibodies (refer to antibody information in Supplementary Table 6) for 30 min at room temperature in the dark. Lysis of red blood cells and fixation was performed by incubation with 1 ml RBC lysis/fixation solution (BioLegend, San Diego, CA) for 15 minutes. Following a washing step with PBS/BSA cells were analyzed using a LSRII cytometer (BD Biosciences) and FlowJo 9.6 software (Treestar, Ashland, OR).

| | |
|---|---|
| Instrument | BD LSRII SORP system (BD Biosciences) |
| Software | FlowJo 9.6 software, BD FACSDiva Software, Kaluza 2.2 (Beckman) |
| Cell population abundance | Cell sorting was not applied |
| Gating strategy | Gating strategy for cardiomyocyte and stromal cell quantification: Living cells were gated based on nuclear DNA signal after labeling with Hoechst-33342 (Pacific Blue-channel). Single cells were separated from cell aggregates. Cardiomyocytes and stromal cells were either labeled with antibodies directed against ACTN2 or VIM, respectively, and detected with an Alexa Flour-488 (FITC-channel) conjugated secondary antibody or exposed to fluorochrome-conjugated antibodies.

Gating strategy for donor specific antibody (DSA) detection: Cardiomyocytes (CM) not stimulated or stimulated with IFN-γ for 48 h were gated based on FSC-A and SSC-A parameters to exclude debris and not incubated or incubated with 1:20 diluted sera obtained from #2887 16 weeks after EHM implantation and after withdrawal of immunosuppression. 20.000 events were measured. A FITClabeled anti-rhesus IgG antibody detected antibodies in the sera bound to the CMs. In addition to the mean fluorescence intensity, the proportion of stained CMs has been determined using the second marker (FITC-A subset 2). The expression of MHC class I molecules on the CMs used in this experiment was determined in parallel using the W6/32 antibody and a FITC-labeled secondary antibody against mouse IgG. Antibodies that display a selective reactivity to IFN-γ-stimulated CMs presumably include DSAs to MHC class I molecules.

Gating strategy peripheral blood mononuclear cells: Leukocytes were gated based on CD45 expression versus SSC-A. Following exclusion of douplets either CD11c+ cells or CD45+ lymphocytes were further gated. T and B cells were distinguished based on CD3 versus CD20 expression. T cells were further divided into CD4+ and CD8+ T cells. NK cells were identified as CD3- CD20-/CD8+ CD159a+ cells. Activation of immune cells was assessed by analyzing CD80 expression on B cells and CD11c+ cells as well CD69 and HLA-DR expression on total CD3+, CD4+, CD8+ T cells and NK cells.

Refer for further details to Supplementary Information: Supplementary Method 1_Flow Cytometry Gating Strategy |

☒ Tick this box to confirm that a figure exemplifying the gating strategy is provided in the Supplementary Information.

# Magnetic resonance imaging

## Experimental design

| | |
|---|---|
| Design type | *Indicate task or resting state; event-related or block design.* |
| Design specifications | *Specify the number of blocks, trials or experimental units per session and/or subject, and specify the length of each trial or block (if trials are blocked) and interval between trials.* |
| Behavioral performance measures | *State number and/or type of variables recorded (e.g. correct button press, response time) and what statistics were used to establish that the subjects were performing the task as expected (e.g. mean, range, and/or standard deviation across subjects).* |

## Acquisition

**Imaging type(s)**
*Specify: functional, structural, diffusion, perfusion.*

**Field strength**
*Specify in Tesla*

**Sequence & imaging parameters**
*Specify the pulse sequence type (gradient echo, spin echo, etc.), imaging type (EPI, spiral, etc.), field of view, matrix size, slice thickness, orientation and TE/TR/flip angle.*

**Area of acquisition**
*State whether a whole brain scan was used OR define the area of acquisition, describing how the region was determined.*

**Diffusion MRI**  ☐ Used  ☐ Not used

## Preprocessing

**Preprocessing software**
*Provide detail on software version and revision number and on specific parameters (model/functions, brain extraction, segmentation, smoothing kernel size, etc.).*

**Normalization**
*If data were normalized/standardized, describe the approach(es): specify linear or non-linear and define image types used for transformation OR indicate that data were not normalized and explain rationale for lack of normalization.*

**Normalization template**
*Describe the template used for normalization/transformation, specifying subject space or group standardized space (e.g. original Talairach, MNI305, ICBM152) OR indicate that the data were not normalized.*

**Noise and artifact removal**
*Describe your procedure(s) for artifact and structured noise removal, specifying motion parameters, tissue signals and physiological signals (heart rate, respiration).*

**Volume censoring**
*Define your software and/or method and criteria for volume censoring, and state the extent of such censoring.*

## Statistical modeling & inference

**Model type and settings**
*Specify type (mass univariate, multivariate, RSA, predictive, etc.) and describe essential details of the model at the first and second levels (e.g. fixed, random or mixed effects; drift or auto-correlation).*

**Effect(s) tested**
*Define precise effect in terms of the task or stimulus conditions instead of psychological concepts and indicate whether ANOVA or factorial designs were used.*

**Specify type of analysis:**  ☐ Whole brain  ☐ ROI-based  ☐ Both

**Statistic type for inference**
(See Eklund et al. 2016)
*Specify voxel-wise or cluster-wise and report all relevant parameters for cluster-wise methods.*

**Correction**
*Describe the type of correction and how it is obtained for multiple comparisons (e.g. FWE, FDR, permutation or Monte Carlo).*

## Models & analysis

| n/a | Involved in the study |
|-----|----------------------|
| ☐ | ☐ Functional and/or effective connectivity |
| ☐ | ☐ Graph analysis |
| ☐ | ☐ Multivariate modeling or predictive analysis |

**Functional and/or effective connectivity**
*Report the measures of dependence used and the model details (e.g. Pearson correlation, partial correlation, mutual information).*

**Graph analysis**
*Report the dependent variable and connectivity measure, specifying weighted graph or binarized graph, subject- or group-level, and the global and/or node summaries used (e.g. clustering coefficient, efficiency, etc.).*

**Multivariate modeling and predictive analysis**
*Specify independent variables, features extraction and dimension reduction, model, training and evaluation metrics.*

