## [Peer Review File · Nature]

Engineered heart muscle allografts for heart repair in primates and humans

Corresponding Author: Professor Wolfram Zimmermann

Parts of this Peer Review File have been redacted as indicated to maintain confidential clinical trial data.

Version 0:

Reviewer comments:

Referee #1

(Remarks to the Author)

Jebran, et. al. present an interesting manuscript exploring whether allogeneic or autologous induced pluripotent stem cell-derived cardiomyocyte (iPSC-CM) transplantation remuscularizes non-human primate hearts. The biggest advantage of this study is that non-human primate is the most adequate model to evaluate immune response following allogeneic iPSC-CM transplantation, as the authors mentioned. The results are not surprising but providing important information to the scientific community. Comments are below.

In the rat model, the authors identified grafted CMs with rhesus mitochondria. How did they identify grafted CMs transplanted into rhesus hearts? If grafted CMs were easily identified because they were transplanted in the epicardial area (i.e., outside of the heart) and separated by the fibrous tissue, how could grafted tissue support cardiac contraction?

In the same context, do the grafted EHM contract in synchrony with the heart? It would be surprising if the EHM, located outside of the heart, electrically integrated with host cardiomyocytes.

What is the rationale for the use of irradiated EHM as a control graft. Is that just an object containing dead cardiomyocytes or functional material releasing something?

Extended data figure 6 contains important information, essential part of this manuscript in my view, but authors should provide more detail regarding immune response. What kind of immune cells infiltrated?

They mentioned that animal #2915 who received ciclosporin, different calcineurin inhibitor from other animals, showed considerable leukocyte infiltration (line 158); however, Extended data figure 6F indicates essentially no inflammation in the same animal. How do they explain? It would be of interest if the recipient treated with ciclosporin fails to have grafted CMs with or without immune response, but multiple number of recipients with same combination of immunosuppressants would be required to conclude this.

The authors claimed that the immune response following iPSC-CM transplantation is stronger in Rhesus macaques than that in human, but, in my view, no one knows that in human.

Two recipients that received autologous EHT showed different outcome; one animal showed surviving graft CMs at 3 months post-transplantation but no graft CM at 6 months. They attributed this graft loss to immune response, indicating autoimmune reaction. Detailed histological analysis will be required but again it would be hard to conclude this with just one recipient.

The authors should provide more detailed histological data from cohort 3 experiments, such as magnified image of graft CMs, maturation markers, gap junction proteins, cadherins, inflammatory responses, etc.

Some animals showed osteochondral tissue in the graft area and I was wondering if it is possible that the tissue was derived from the stromal cell preparation.

They presented EF data of Responder and Non-responder separately in figure 3E and mentioned "EHM allograft-enhanced

target heart wall contractility and ejection fraction, measures for local and global heart support, was obtained" in the abstract. They should show aggregated EF data.

How did they calculate "cardiomyocyte volume"?

They mentioned "clear separation of host and graft (5x EHM) heart muscle was possible" (line 246), but I am not sure about this when I see figure 4A without green and orange lines.

Referee #2

(Remarks to the Author)

This study by Drs. Jebran et al seek to examine whether the engineered heart muscle could repair myocardial infarction using a resus macaques model. The manuscript is well prepared. Using the NHP model for a long term follow up is certainly very valuable. The concern is that there is no convincing data demonstrating the EHM graft 6 months after transplantation. For example, Figure 3B (and Figure 4A), the dotted line circulating an area. However, we do not see scientific data showing that are muscle cells from the EHM. In extended Fig 3, the Rhesus mitochondria (green) are so few in number, which makes one concern that long term graft size could be very small if any. Similarly, in extended figure 6, because there is no EHM specific staining, the claimed EHM staining is speculative, and are not supported by the scientific data.

Specifics:

- 1) Abstract, please include numbers in each groups (n=?)
- 2) Heart failure is used in the entire manuscript. However, there is no data indicate that this is a heart failure NHP model. Maybe you should use the term of postinfarction LV remodeling.
- 3) From lines 63-66, for conveying the scientific clarity, please include mean \pm SD, and p= ? for each of the assessments of contractility and ejection fraction, histological analysis, MRI graft size (I am not sure your MRI method applied can measure graft size), and fractional vascularization
- 4) Discussion, maybe you can include a few sentences on Dr Y Sawa's human clinical trial (Japan) using hiPSC-cardiomyocytes, which has been going on for a few years.

Version 1:

Reviewer comments:

Referee #1

(Remarks to the Author)

The authors presented substantially new data, including the histological outcomes of human heart transplanted EHM, in the revised manuscript, making it even more valuable. They have also addressed some issues raised in the initial round of review, which were summarized in the point-by-point rebuttal; however, it brought forth additional concerns, and part of the issues remains unresolved.

1. The histological outcome from the human recipient of EHM is intriguing. The authors should provide detailed information.
 - A. Firstly, why did the patient receive a heart transplant? Didn't the transplantation of EHM provide sufficient effects?
 - B. Are there any immune responses to the grafted EHM? Please provide detailed immuno-histological data.
 - C. Is the picture showing CD31 staining in the graft? Please provide pictures double-stained with a cardiac marker and CD31 that clearly show location of host and grafted CMs. The authors repeatedly claimed, "EHM grafts, very much in contrast to injected cardiomyocytes, can be clearly identified and distinguished from the recipient myocardium".
 - D. In the same context, please provide pictures, perhaps with cardiac markers, showing the location of graft and host CMs. Also, it would be helpful to compare characteristics of CMs between host and graft CMs following human allogeneic transplantation.
 - E. Please provide the lengths of the three scale bars in figure 4A.
2. The authors claimed that electrical coupling of EHM has been established in their previous publication as follows: "Electrical coupling and synchronicity of EHM allografts⁹, as well as injected cardiomyocyte allografts¹⁰ and xenografts¹, is a consistent observation." However, ref^{#9} did not provide direct evidence of electrical integration of EHM; instead, it presented indirect observations. The current manuscript does not demonstrate evidence of "mechano-electrical integration." I was surprised that they did not assess any paracrine effects.
3. The response, "Irradiated EHM represent an upon implantation dying tissue graft, which we had also used in a previous rodent study¹³ as non-contractile/dead tissue controls," does not directly address the comment. In reference #13, the transplantation of irradiated EHM yielded functional benefits comparable to intact EHM, in contrast to the findings in the current study. This suggests distinct functional characteristics of irradiated EHM. Please address this discrepancy.
4. The statement "We have also scored immune cell infiltration using clinical pathology standards" lacks specific information about the actual scores related to immune cell infiltration. It would be helpful if the provided details or scores on immune cell

infiltration could be included for clarity.

5. Please provide more reader-friendly figures. I will provide some of the examples:

- A. Include information regarding the experimental group and/or antibodies for pictures in Extended Data Figure 7A, B, D, G.
- B. In Extended Data Figure 7, along with the raw data, consider creating a figure or table that summarizes the results for better clarity.
- C. Present the aggregated ejection fraction (EF) figure shown in the rebuttal within the manuscript for better integration and understanding.

6. Extended Data Figure 9 was confusing. Please provide clarification by specifying which cells are graft cardiomyocytes and which are host cardiomyocytes. Additionally, despite the numerous CD56 positive cells observed, the claim of "no evidence for innate immune cell (NK-cells, macrophages) involvement" appears contradictory. Addressing this discrepancy would enhance the clarity of the presented data.

7. The authors utilized cyclosporin in only one animal (#2915) and did not observe either graft survival or inflammation. The conclusion drawn, suggesting graft rejection due to donor-specific antibodies (DSA), may not accurately reflect the actual graft rejection, as this determination would require the examination of a larger number of animals under cyclosporin treatment. It is recommended to conduct the analysis on multiple animals to establish a more robust conclusion regarding "graft rejection" under the influence of cyclosporin.

8. I am not certain if the equation for "cardiomyocyte volume" presented by the authors accurately represents the actual volume. Using the term "cardiomyocyte area" might be more accurate, considering that the parameters were obtained from 2-dimensional sections. This adjustment would better align with the nature of the data obtained.

9. To enhance clarity, it would be beneficial to measure and present graft size in all recipients. This additional information would provide a more comprehensive understanding of the study's results.

Referee #2

(Remarks to the Author)

The authors are responsive. And manuscript has been significantly improved. The findings are significant and important for the field. I have the following suggestion/comments for further improve the manuscript :

1. Although authors showed that engraftment of EHM is receipt animal hearts, almost all images were shown in relative high magnification. This makes it difficult to have an overall impression of how the EHM survived in the receipt hearts, representative images like right panel of Figure 4 A shall be shown for extended data Figure 9 and extended data figure 16A&16B)
2. Authors heavily used immunohistochemistry to show the engraft and immune cell infiltration in engraft. Immunofluorescence staining shall be performed to show simultaneously the engraft and immune cell. This also applies to Figure 4B, immunofluorescence staining shall be performed to show the vessels in engraft.
3. It was reported by authors that RiPSC-CMs used for manufacturing EHM were more than 95% pure for ACTN2. ACTN2 is not a CM specific protein, and is expressed in skeletal muscle as well. Authors shall perform cTnT or cTnI staining to determine the purity of RiPSC-CM used for EHM.
4. It is unclear the 95% purity of RiPSC-CMs was freshly differentiated RiPSC-CMs or after purification.
5. It was mentioned by authors that "we tested 4 different Rhesus macaque iPSC-lines, including 2 newly generated lines to also obtain insight as to in vivo autograft responses (Extended Data Table)." However, it is unclear which cell line was used in which animal and immunosuppression drugs. Did EHMs manufacture from 3 RiPSC-CM have the similar structure and contractile performance?
6. Authors heavily used desmin as an evidence of engraft in receipt animals hearts which is unusual (Figures 2B, 3C, 4B, extended data figure 7). Desmin is also expressed in smooth muscle cells and skeletal muscles, authors relied on desmin to show the host and donor CM, which is unconvincing. CM specific protein markers, such as cTnI or cTnI, shall be performed.
7. Extended data Figure 3, merged pic of cTnT and Rhesus Mitochondria shall be provided. Currently it is difficult to assess whether cTnT and Rhesus Mitochondria were colocalized.
8. Can individual data be presented, such as scatter dot plot?
9. Can authors discuss/speculate what benefit can be expected from such a small and thin engraftment (μm thickness) on a heart with cm thick left ventricular wall in human clinical trials?

Referee #4

(Remarks to the Author)

Jebran et al. conducted allogeneic transplantation experiments using Rhesus macaque-derived EHM under various conditions including transplant cell volume and immunosuppressive agent usage. In an optimized cohort, they demonstrated tissue regeneration with blood perfusion and cardiac functional recovery effects six months post-transplantation. Importantly, no significant adverse effects such as arrhythmias or tumor formation were observed, marking a significant advancement in this research area. However, reviewer believes that further detailed examination of the treatment mechanism and

presentation of data is necessary for authors to assert that this therapeutic effect is directly due to remuscularization. Furthermore, authors obtained a rare opportunity in this paper to analyze heart samples from recipients of the BioVAT-HF trial, proving the potential of EHM to engraft as regenerated myocardium in human hearts. While this finding holds great significance in the study, additional evaluation of these samples is desirable.

Specific comments:

1. As the authors also comment in the text, in previous similar studies, the therapeutic effects have been suggested to be "at least partially mediated by immune responses or paracrine mechanisms" (line 86-87). In contrast, the authors suggest based on the results of experiments using irradiated EHM as a control group that the therapeutic effects in this study are mediated by "muscularization-independent mechanisms" (line 89). However, the reviewer believes that further detailed examination of this therapeutic mechanism is warranted. This is because, as noted by the authors themselves (citing #20-22), in previous studies on pluripotent stem cell-derived cardiomyocyte transplantation, at least some of the therapeutic effects have been attributed to paracrine indirect effects. Therefore, even if there were therapeutic effects due to remuscularization in the EHM transplantation in this study, there should have also been paracrine effects. It is unclear from the data presented what proportion of the therapeutic effects is attributed to each mechanism. At the very least, the authors should demonstrate what indirect effects, if any, were present in this EHM transplantation. For instance, if there were angiogenic effects in the ischemic heart, it is plausible that the degree of angiogenesis would primarily occur in the border zone of ischemia rather than within the EHM itself, but data regarding this aspect are not provided. How were cardiomyocyte apoptosis, likely induced by cardiac ischemia in the border zone, affected by EHM transplantation? If most of the therapeutic effects were due to indirect paracrine effects, the therapeutic effects of remuscularization would be limited, thus evaluating this aspect is crucial.

2. Understanding how EHM provides mechanical contractile force to the host heart post-transplantation is a crucial point. While the authors suggest the presence of a mechano-electrically coordinated manner (line 278-279), data supporting this assertion seem to be lacking. It remains unclear if there are any mechano-sensors present in the EHM and the host heart tissue at the transplantation site. If so, as indicated in Fig. 1D, since this EHM exhibits automaticity (self-beating without external stimuli), if mechano-electrically coordinated manner is indeed the mechanical coupling pattern between EHM and the host heart, wouldn't it potentially induce ectopic contractions in the host heart immediately after transplantation, leading to arrhythmias? Why then, in actuality, did arrhythmias not occur even in responders where EHM engraftment was successful? The authors should address this point to provide clarity.

3. The patient samples transitioning to heart transplantation in the BioVAT-HF trial could provide invaluable data. However, it is conceivable that these samples were targeted for heart transplantation precisely because they were non-responders in the clinical trial itself. In other words, this heart might be considered a sample that didn't achieve the desired therapeutic effects adequately. Was the engraftment, particularly in this human heart sample, insufficient compared to responders in the Rhesus macaque allograft transplantation performed in this study? If adequate engraftment was indeed achieved, why couldn't sufficient therapeutic effects be attained? Could it be attributed to factors such as the extent of preoperative cardiac impairment? Furthermore, did arrhythmias not occur post-transplantation in this patient? While acknowledging that data obtained from this sample should include information that ought to be confidential within the BioVAT-HF trial, it is crucial for the research to demonstrate and discuss the evidence of therapeutic efficacy in humans regarding EHM allograft transplantation. This is important not only for advancing this treatment approach clinically but also for the research itself. At least, it is deemed necessary to ensure that the results in Rhesus macaque allograft transplantation are not contradictory to the data regarding therapeutic efficacy and safety in this human case.

Version 2:

Reviewer comments:

Referee #1

(Remarks to the Author)

The authors have addressed most of the comments. However, I have an additional comment regarding the new Supplementary Video 3, which is quite intriguing. After physical stimulation of one side of the engineered heart muscle (EHM), the other side contracts. However, the interval between stimulation and contraction is approximately 2 seconds, which is too slow to synchronize with host beating hearts. Additionally, the propagation of contraction within the EHM does not appear sequential, which is unexpected. Please provide an explanation for this.

Referee #2

(Remarks to the Author)

The authors are very responsive. The responses are satisfactory. The manuscript is significantly improved. The findings are significant and novel. The data are solid.

Referee #4

(Remarks to the Author)

The reviewer believes that the authors have sufficiently addressed the reviewers' concerns. The reviewer also understands that a detailed evaluation of the effects of paracrine factors is challenging within the study design using Rhesus macaques.

The additional data provided by the authors can serve as a valuable contribution for ongoing discussions within the scientific community on this point. Furthermore, the reviewer agrees that continued investigation into electrical and mechanical coupling is necessary. The reviewer also appreciates the effort to present valuable human sample data derived from the BioVAT-HF trial to the extent possible.

[Redacted text and figure]

[Redacted text and figure]

Point-by-Point Response (Jebran et al. 2023-03-04361):

We like to thank the referee for the careful review of our manuscript. Please, find a point-by-point response to the critiques (**verbatim in bold**) below:

Referee #1 (Remarks to the Author):

Jebran, et. al. present an interesting manuscript exploring whether allogeneic or autologous induced pluripotent stem cell-derived cardiomyocyte (iPSC-CM) transplantation remuscularizes non-human primate hearts. The biggest advantage of this study is that non-human primate is the most adequate model to evaluate immune response following allogeneic iPSC-CM transplantation, as the authors mentioned. The results are not surprising but providing important information to the scientific community. Comments are below.

We thank the reviewer for pointing out the importance of non-human primate models in the preclinical assessment of cardiac remuscularization. As to the statement that the results are not surprising, we respectfully disagree. To date there is no other study demonstrating extensive long-term remuscularization with concomitant functional improvement of heart function without arrhythmia in a from a translational point of view relevant allograft large animal model of heart failure. In fact, previous excellent studies on cardiomyocyte injection in relevant large animal models (macaques and pigs) demonstrated ventricular arrhythmia^{1-3,6,7} or no graft retention 6 months after implantation.⁸ In addition, a previous report found that “hESC-CM-treated animals also received epicardial application of 1–3 tissue-engineering constructs where hESC-CMs were seeded in a collagen scaffold. (These tissue engineered constructs did not adhere to the epicardial surface and were not recovered at the end of the experiment).” (cited from Chong et al.¹).

In our study, we clearly demonstrate that EHM allografts are retained for at least 6 months in Rhesus macaques and that EHM implantation is safe and efficacious. In the revision, we include a new **ED Fig. 17**, showing how EHM are assembled at the point-of-care before implantation (**ED Fig. 5A**). We now also include data from a patient from the BioVAT-HF Phase I/II clinical trial, who was successfully heart transplanted 3 months after EHM implantation, confirming human allograft retention for at least 3 months. Please note that we are including this data after discussion with the editor, but cannot provide additional clinical details from this ongoing clinical trial.

In the rat model, the authors identified grafted CMs with rhesus mitochondria. How did they identify grafted CMs transplanted into rhesus hearts? If grafted CMs were easily identified because they were transplanted in the epicardial area (i.e., outside of the heart) and separated by the fibrous tissue, how could grafted tissue support cardiac contraction?

Pericardial EHM implants can be easily identified by their anatomical location and sheer size. It is indeed a recurrent observation that the graft-host interface comprises of a 100-500 μm regenerative fibrosis. In addition, EHM grafts are implanted to “bridge” scarred myocardium. While the mechanisms of graft-host coupling (regardless whether cardiomyocytes are injected into scarred myocardium or patches are implanted on scarred myocardium) remains to be clarified, electrical coupling and synchronicity of EHM allografts⁹ as well as injected cardiomyocyte allografts¹⁰ and xenografts¹ is a consistent observation. It is quite obvious that the classical view of homogeneous gap junction mediated coupling cannot explain the observed host-graft synchrony in any of the reported models. Accordingly, we hypothesize that cardiomyocyte grafts are mechanically conditioned to electrically synchronize with the recipient myocardium; the underlying mechanism may be termed mechano-electrical integration. We understand that this is thought provoking and

that further investigations will have to confirm or reject this hypothesis. Accordingly, we have added the following statement into the revised manuscript:

Lines 272-280:

This fundamental difference may be the result of different modes of electrical integration. Intramurally injected cardiomyocytes are clearly capable of coupling¹⁰ and ectopic firing, leading to engraftment arrhythmia in large animal models^{2,3,6}, which may be attenuated by genetic depletion of depolarizing ion channels.¹¹ Tissue engineered patches, by virtue of their epicardial location, cannot readily establish electromechanical connections, but appear to be mechanically entrained to contribute to myocardial performance in a mechano-electrically coordinated manner. Further studies are needed to clarify the time course and mode of integration as well as beyond 6-month safety and efficacy outcomes.

In the same context, do the grafted EHM contract in synchrony with the heart? It would be surprising if the EHM, located outside of the heart, electrically integrated with host cardiomyocytes.

Synchrony of contraction, as evidenced by systolic thickening of the target heart wall, has been observed in rat (Fig 4G in Zimmermann et al. 2006¹²), mouse (Fig 9D in Didié et al. 2013¹⁰) and now consistently also in our NHP allograft study (**Figure 2D**). In contrast to the classical view of electro-mechanical coupling via gap junctions, we put forward a mechanism, which we term mechano-electrical synchronization. It considers that the mechanical impulses from the beating recipient heart entrain and mature the engrafted cardiomyocytes to contract in synchrony with the recipient myocardium. In the revised manuscript, we include a statement on cardiomyocyte integration and how it may differ in case of cardiomyocyte injections and epicardial patch implantations (please, refer to highlighted text above and in the revised main manuscript).

What is the rationale for the use of irradiated EHM as a control graft. Is that just an object containing dead cardiomyocytes or functional material releasing something?

Irradiated EHM represent an upon implantation dying tissue graft, which we had also used in a previous rodent study¹³ as non-contractile/dead tissue controls.

Extended data figure 6 contains important information, essential part of this manuscript in my view, but authors should provide more detail regarding immune response. What kind of immune cells infiltrated?

We have done extensive immune phenotyping by comprehensive DSA analyses in all implanted animals (**ED Fig ED8**), we have also scored immune cell infiltration using clinical pathology standards (**ED Fig 7F**). In addition, we have performed comprehensive flow cytometry analyses of circulating immune cells (referred to in **lines 212-215**). In the PoC Cohort 3 (which was informed by the immune protocol investigations in Cohorts 1 and 2), no immune rejection with concurrently strong remuscularization was observed under increased Tacrolimus (~20 ng/mL trough levels) and concomitant Methylprednisolone (refer to **ED Fig 7F** copied blow with Cohort 3 data highlighted by red boxes).

To improve our understand of the immune response, we performed additional immunohistochemistry investigations in the Cohort 1 autograft model (#2483; refer to blue bars in Cohort 1 in **ED Fig 7F**), in which cardiomyocyte retention and graft directed immune responses were clearly notable (**ED Fig 7B**); note that this was not the case in #2500 with 6-month follow-up with notable immune cell infiltration, but only very few cardiomyocytes retained. After testing of several

antibodies for their specificity in Rhesus macaque FFPE samples, we identified CD3/TCR-positive T-cell infiltrates with concomitant CD20-positive B-cell infiltrates, suggestive of a classical T-cell mediated graft response (**ED Fig 9**). This finding was further substantiated by the absence of an innate immune response, i.e., no evidence for CD56/CD57-positive NK-cells and CD68-positive macrophages.

This finding is included in lines 154-157:

A detailed analysis of the leukocyte infiltrate in the corresponding 3-months autograft model (cohort 1 #2483; **Extended Data Fig. 7B**) identified T-cell mediated rejection with concomitant B-cell accumulation and no evidence for an innate immune response (**Extended Data Fig. 9**).

They mentioned that animal #2915 who received ciclosporin, different calcineurin inhibitor from other animals, showed considerable leukocyte infiltration (line 158); however, Extended data figure 6F indicates essentially no inflammation in the same animal. How do they explain? It would be of interest if the recipient treated with ciclosporin fails to have grafted CMs with or without immune response, but multiple number of recipients with same combination of immunosuppressants would be required to conclude this.

We apologize for having misstated our observation in the Cyclosporin animal. Indeed, there was no obvious immune cell infiltration at the 6 months endpoint of the study in animal #2915, which had fully rejected the cardiomyocyte allograft (refer to **ED Fig 7D and F** below – Cyclosporin group highlighted with red arrows):

DSA analyses (**ED Fig 8 animal #2915; right panel below**) revealed a strong allograft immunization after reduction of Cyclosporin to target trough levels (140-250 ng/mL; **ED Fig 6B: left panel below**). These observations point to an acute rejection episode during the 6-month follow-up as a consequence of Cyclosporin dose adjustment:

We have revised the main text (**lines 151-162**) to include more information on the unanticipated immune responses in autografts (#2483 [3-month Cohort 1]; #2500 [6-month Cohort 2]) and the with Cyclosporin and Methylprednisolone immune suppressed allograft animal (#2915 [6-month Cohort 2]):

Unexpectedly, we observed rejection of the autograft (#2500) and the allograft (#2915) under Cyclosporin with Methylprednisolone (**Extended Data Fig. 7D**). Donor-specific antibody (DSA) analyses revealed no evidence for autograft immunization in #2500 (**Extended Data Figs 8**). A detailed analysis of the leukocyte infiltrate in the corresponding 3-months autograft model (cohort 1 #2483; **Extended Data Fig. 7B**) identified T-cell mediated rejection with concomitant B-cell accumulation and no evidence for an innate immune response (**Extended Data Fig. 9**). In contrast, strong allograft immunization was observed in #2915 (6-months allograft) after reduction of Cyclosporin to target trough levels (140-250 ng/mL), suggesting rejection upon dose adjustment (**Extended Data Fig. 6**). These findings are in agreement with a previous report,³ demonstrating that even under Mamu[MHC]-matched allograft conditions higher than clinically accepted doses of calcineurin inhibitors are required to ensure macaque cardiomyocyte allograft retention.

The authors claimed that the immune response following iPSC-CM transplantation is stronger in Rhesus macaques than that in human, but, in my view, no one knows that in human.

This statement was stimulated by the surprising observation of the need for beyond clinically approved levels of tacrolimus and the rejection under cyclosporine at clinically established doses. We agree that more work is needed to substantiate our claim and have accordingly deleted the original statement.

Two recipients that received autologous EHT showed different outcome; one animal showed surviving graft CMs at 3 months post-transplantation but no graft CM at 6 months. They attributed this graft loss to immune response, indicating autoimmune reaction. Detailed histological analysis will be required but again it would be hard to conclude this with just one recipient.

The observation of complete rejection 6 months after implantation of an autograft (animal #2500) was indeed surprising (see response above), but aligned with the observation of immune cell infiltration in the autograft animal in Cohort 1 (#2483; 3-month follow-up). We performed additional immunotyping of the leukocyte infiltration in the autograft animal #2483 and identified a classical T-cell mediate response, with the commonly observed secondary B-cell infiltration and no evidence for innate immune cell (NK-cells, macrophages) involvement. This finding was surprising to

us and is to our knowledge the first time that iPSC-autograft rejection is documented in a clinically relevant non-human primate model. A similar observation had been made in an earlier mouse study.⁵ Collectively, the data from Zhao et al.⁵ and us raise an important caveat as to the common assumption that iPSC autografts would be accepted without immune suppression. The observed T-cell mediated rejection points to MHC-presentation of autoantigens as trigger of the immune response. We cannot claim that autografts will always be rejected, but find it important to raise awareness of the caveat that iPSC-autografts may have to be handled similarly as iPSC-allografts, i.e., with concomitant immune suppression.

The authors should provide more detailed histological data from cohort 3 experiments, such as magnified image of graft CMs, maturation markers, gap junction proteins, cadherins, inflammatory responses, etc.

Please note that we have provided low-, mid-, and high-power magnifications of cardiomyocyte grafts as well as a detailed analysis of cardiomyocyte volumes (panels below from Cohort 3 samples):

Fig 2B (left panel) shows a cross-section of an entire heart with EHM graft (encircled); **Fig 3C** (middle panel) shows representative regions within an engrafted EHM; **ED Fig 7D** (right panel) highlights a single implanted cardiomyocyte. In addition to providing images, we have carefully analyzed cardiomyocyte volume (**Fig 3C**) and demonstrate phenotypic maturation and organotypic growth, which we term adaptive hypertrophic growth, after implantation.

In response to the reviewer's recommendation, we now include additional immunohistochemistry stainings for slow skeletal (fetal) troponin I (TNNT1) and cardiac (adult) troponin I (TNNT3), ventricular (MYL2) and embryonic/atrial (MYL4) myosin light chains, n-cadherin (intercalated disk protein) and connexin 43 (gap junction protein) performed in Rhesus Cohort 3 (**ED Fig 16A**) and the proof-of-concept human heart (**ED Fig 16B**). The data collectively confirms a maturing, but in comparison to adult myocardium immature cardiomyocyte phenotype.

This information has been added to lines 246-252 (NHP findings) and 258-259 (findings from human heart):

The relative immaturity of the implanted cardiomyocytes was further supported by the identification of TNNI1 (troponin I isoform in immature myocardium) and TNNI3 (troponin I isoform in adult ventricular myocardium) as well as stronger staining for MYL4 (myosin light chain isoform in atrial and immature myocardium) compared to MYL2 (myosin light chain isoform in adult ventricular myocardium). Engrafted cardiomyocytes showed evidence of intercalated disk formation (CDH2) with sparse expression of the gap junction protein connexin 43 (GJA1; **Extended Data Fig. 16A**).

An immature cardiomyocyte phenotype, similar to our observations in the NHP model, was observed (**Extended Data Fig. 16B**).

Some animals showed osteochondral tissue in the graft area and I was wondering if it is possible that the tissue was derived from the stromal cell preparation.

We agree that this is a plausible assumption. However, our snRNA-seq data argues against this as it shows evidence for osteochondral cells in the NHP-iPSC derived cardiomyocyte population (**ED Fig 1A (ii)**). Another finding that argues against the stromal cell preparation being the origin of the osteochondral cells is that we did observe osteochondral cells in #2520 and #2506, which, in contrast to #2887, #2909, #2913, #2907, and #16721, were not prepared with separately from the cardiomyocyte population prepared stromal cells (refer to **ED Table 3** for a summary of the EHM cell compositions implanted in Cohorts 1 and 2).

They presented EF data of Responder and Non-responder separately in figure3E and mentioned “EHM allograft-enhanced target heart wall contractility and ejection fraction, measures for local and global heart support, was obtained” in the abstract. They should show aggregated EF data.

The data is aggregated in **ED Table 6A** and now also depicted in the graph below. Please note that we prefer to show the individual animal trajectories in the figures for maximal transparency. A particular strength of such a display is that the trajectory (based on sequential MR-imaging) can be clearly appreciated.

How did they calculate “cardiomyocyte volume”?

We have added the following information to the methods section (**lines 846-848**):

Cardiomyocyte volume was calculated from desmin-stained samples using planimetry to determine cardiomyocyte length and breadth: $CM\ volume = \pi * (CM\ length/2) * (CM\ breadth/2) * CM\ breadth$

They mentioned “clear separation of host and graft (5x EHM) heart muscle was possible” (line 246), but I am not sure about this when I see figure4A without green and orange lines.

The lines encircle the regions of interest, which can be separated as EHM implant and recipient (remote) myocardium. To improve clarity, we have included a CINE and Gd-perfusion images from different time points after Gd-injection. The white arrows point at the EHM graft which can be clearly distinguished from the recipient myocardium.

Figure 4: Evidence for EHM allograft vascularization and perfusion. (A) Gadolinium (Gd)-perfusion MRI data obtained in a 5x EHM implanted Rhesus macaque (#2819) with evidence for functional vascularization of EHM grafts in a heart failure model at the indicated timepoints. **Left panel: the regions of interest (ROIs) from which the Gd-signal was reported are encircled and distinguished as EHM and remote myocardium.** The lower MRI images depict a CINE and the respective Gd-perfusion images recorded at the indicated time points 4 weeks after EHM (marked by arrows) implantation.

We like to thank the referee for the careful review of our manuscript. Please, find a point-by-point response to the critiques (verbatim in bold) below:

Referee #2 (Remarks to the Author):

This study by Drs. Jebran al et al seek to examine whether the engineered heart muscle could repair myocardial infarction using a resus macaques model. The manuscript is well prepared. Using the NHP model for a long term follow up is certainly very valuable. The concern is that there is no convincing data demonstrating the EHM graft 6 months after transplantation. For example, Figure 3B (and Figure 4A), the dotted line circulating an area. However, we do not see scientific data showing that are muscle cells from the EHM. In extended Fig 3, the Rhesus mitochondria (green) are so few in number, which makes one concern that long term graft size could be very small if any. Similarly, in extended figure 6, because there is no EHM specific staining, the claimed EHM staining is speculative, and are not supported by the scientific data.

- 1) No convincing data demonstrating the EHM graft 6 months after transplantation. For example, Figure 3B (and Figure 4A), the dotted line circulating an area. However, we do not see scientific data showing that are muscle cells from the EHM.**

Note that the analyses were performed by a clinical pathologist, who was blinded to the study procedure (Ctr vs EHM-implanted hearts; EHM dose; immune suppression protocol). The epicardial location as well as the coherent structure and size of the graft make it very easy to distinguish graft from host myocardium. Please also note that this is IND-enabling study, genetic labels (such as GFP) are not compatible with clinical use of cellular grafts as they may cause unwanted immune responses. As another proof of graft and host identity, we have performed additional microsatellite analyses in Cohort 3 macaques (**ED Figure 15A**) and deep sequencing for the identification of genomic variants in the human heart (**ED Figure 15B**).

Lines 217-218:

EHM grafts graft identity was further confirmed by genomic microsatellite analysis (Extended Data Fig. 15A).

Lines 258-259:

EHM graft identity was confirmed by single nucleotide variant (SNV) analyses (Extended Data Fig. 15B).

The new method description is inserted in **lines 850-865**.

Graft identity assessment. DNA was isolated as described previously⁴¹ from micro-dissected formaldehyde-fixed paraffin embedded (FFPE) slices using the InnuPREP FFPE DNA Kit on the InnuPure C16 System (Jena Analytika, Jena, Germany) according to manufacturer instructions. Samples were obtained from desmin positive remote and EHM engrafted areas. DNA concentrations were measured on a Qubit 3 Fluorometer (ThermoScientific, Paisley, UK). Microsatellite genotyping was performed in macaque samples using a genotyping-by-sequencing approach as described previously.¹⁴ Allele calling based on sequence data generated on Illumina's MiSeq platform (251 bp forward, 51 bp reverse) was done with the CHIIMP pipeline.¹⁵ In human samples, deep sequencing of a targeted multigene panel (78 genes) was performed on 50 ng genomic DNA. For library preparation, SureSelectTM XTHS target

enrichment Kit (Agilent) with enzymatic fragmentation was used following the manufacturer's protocol (Agilent). Libraries were sequenced on an Illumina NovaSeq6000 with 2x 150 bp read length and with mean coverage of 3,000x. Sequence Pilot (jsi medical systems GmbH) software was used to align sequences to a human reference genome (hg19) and for single nucleotide variant (SNV) calling. SNVs were filtered against (1) absence from control area, (2) high coverage, and (3) exclusion of sequence artefacts.

2) In extended Fig 3, the Rhesus mitochondria (green) are so few in number, which makes one concern that long term graft size could be very small if any.

We agree and believe that this may have to do with the lower sensitivity of the human mitochondria specific antibody in Rhesus mitochondria. The rat study served as a first screen whether NHP-EHM grafts would survive after implantation similarly as observed for human EHM.¹³ In this study, we demonstrated >200 days retention of xenografts in the rat. Xenograft studies have limited value as to iPSC-CM integration and long-term survival, but provide a first important hint as to in vivo feasibility. EHM graft sizes in our monkey study are large, with no evidence for cell loss after implantation. In fact, there is evidence for physiological hypertrophic growth (**Fig 3C**) resulting in a Desmin/Patch Ratio (i.e., an estimate of cardiomyocyte volume per patch) of ~50% under optimized conditions (Cohort 3; **Fig 3C** and **ED Fig 7E**). This is an at least 2-fold higher volume fraction compared to the cardiomyocyte volume fraction at the time point of implantation, indicating an effective heart muscle increase. Note that in previous cardiomyocyte injection studies, cell retention at 3 months was estimated to be ~15% (n=2 at 3 months in Liu et al. 2018²), indicating that a substantial loss of muscle occurred. These findings are aligned with the general observation of higher cell retention if applied via a tissue engineered format.

3) Similarly, in extended figure 6, because there is no EHM specific staining, the claimed EHM staining is speculative, and are not supported by the scientific data.

We respectfully disagree, EHM grafts, very much in contrast to injected cardiomyocytes, can be clearly identified and distinguished from the recipient myocardium by their location and appearance. The now in response to the referee's critique added unbiased genetic analyses (**ED Fig 15**) are in agreement with our original interpretation.

Specifics:

1) Abstract, please include numbers in each groups (n=?)

We can do this, but like to point out that the abstract/summary paragraph word count is very much limited and that the addition of n-numbers, means+-SD/SEM would disturb the flow of the summary paragraph. In the main body of the manuscript and figures as well as with the extended data figures and tables detailed information as to n-numbers, means+-SD/SEM, and performed statistical tests is provided.

2) Heart failure is used in the entire manuscript. However, there is no data indicate that this is a heart failure NHP model. Maybe you should use the term of postinfarction LV remodeling.

ED Fig 14 clearly demonstrates a stably reduced EF (more than 10%; **ED Fig 14B**), a dramatic loss of infarcted heart wall contractility; **ED Fig 14F**), and LV dilation (**ED Fig 14C**). We agree that we have not created a model of severe heart failure, such as observed in patients with advanced heart

failure, but emphasize that such a model does not exist to our knowledge and would very likely not find ethical approval (animals suffering from heart failure symptoms such as edema, shortness of breath, and arrhythmia would have to be euthanized to prevent suffering). We would like to point out that all studies we are aware of in the field implant in a much less chronic disease scenario (typically 2 weeks post-MI) and that the by us introduced model of chronic heart failure with implantation 6 months after myocardial infarction is unique by itself. Post-MI remodeling is of course occurring, but does in our view not describe well the phenotype with stably and significantly compromised heart function.

3) From lines 63-66, for conveying the scientific clarity, please include mean \pm SD, and p= ? for each of the assessments of contractility and ejection fraction, histological analysis, MRI graft size (I am not sure your MRI method applied can measure graft size), and fractional vascularization

Please refer to the response above. The summary paragraph is very much limited in length and prepared according to the journal's instructions.

4) Discussion, maybe you can include a few sentences on Dr Y Sawa's human clinical trial (Japan) using hiPSC- cardiomyocytes, which has been going on for a few years.

We would be happy to do this, but could not find a specific reference reporting data from the cardiomyocyte cell sheet trial (registered as NCT04696328) pioneered by Dr. Sawa. We are aware of two reports describing the outcome of skeletal muscle cell sheet implantations.^{16,17} To recognize the ongoing efforts in clinical translation of cardiomyocyte therapies, we include a review by Kim et al. 2022, which provides a nice overview of the ongoing iPSC-clinical trials (summary of clinical trials, including the Sawa trial, in Table 1 in Kim et al.¹⁸). In addition to the studies listed in Kim et al., we are aware of another recruiting iPSC-cardiomyocyte study spearheaded by Prof. Fukuda/HeartSeed (LAPiS - NCT04945018, Japan) and an ESC-cardiomyocyte study led by Prof. Wu (HECTOR - NCT05068674, Stanford). The limited word count makes it unfortunately difficult to recognize all groups in the field. To highlight the ongoing translation from late preclinical to clinical studies we have added the following text:

Lines 286-289:

Lessons learned from BioVAT and other ongoing clinical trials testing pluripotent stem cell-derived cardiomyocytes implantation¹⁸ will improve our understanding of whether and how remuscularization of the failing human heart can be achieved with clinically meaningful outcomes.

We are happy to add the NCT registration numbers or a table of the to us known clinical trials on PSC-derived cardiomyocyte implantation as Supplementary Information:

Clintrials.gov	Akronym	Cell Therapy Medicinal Product	Start	Patients (n)	Country
NCT04696328	CellSheet	iPSC-cardiomyocyte cell sheet	2019	10	Japan
NCT04396899	BioVAT-HF	iPSC-cardiomyocyte patch	2020	53	Germany
NCT03763136	HEAL-CHF	iPSC-cardiomyocytes	2021	20	China
NCT04982081			2021	20	
NCT05566600			2022	32	
NCT05068674	HECTOR	ESC-cardiomyocytes	2022	18	USA
NCT04945018	LAPiS	iPSC-cardiomyocyte spheroids	2022	10	Japan

References:

- 1 Chong, J. J. *et al.* Human embryonic-stem-cell-derived cardiomyocytes regenerate non-human primate hearts. *Nature* **510**, 273-277, doi:10.1038/nature13233 (2014).
- 2 Liu, Y. W. *et al.* Human embryonic stem cell-derived cardiomyocytes restore function in infarcted hearts of non-human primates. *Nature biotechnology* **36**, 597-605, doi:10.1038/nbt.4162 (2018).
- 3 Shiba, Y. *et al.* Allogeneic transplantation of iPS cell-derived cardiomyocytes regenerates primate hearts. *Nature* **538**, 388-391, doi:10.1038/nature19815 (2016).
- 4 Kashiwama, N. *et al.* MHC-mismatched Allotransplantation of Induced Pluripotent Stem Cell-derived Cardiomyocyte Sheets to Improve Cardiac Function in a Primate Ischemic Cardiomyopathy Model. *Transplantation* **103**, 1582-1590, doi:10.1097/TP.0000000000002765 (2019).
- 5 Zhao, T., Zhang, Z. N., Rong, Z. & Xu, Y. Immunogenicity of induced pluripotent stem cells. *Nature* **474**, 212-215, doi:10.1038/nature10135 (2011).
- 6 Romagnuolo, R. *et al.* Human Embryonic Stem Cell-Derived Cardiomyocytes Regenerate the Infarcted Pig Heart but Induce Ventricular Tachyarrhythmias. *Stem cell reports* **12**, 967-981, doi:10.1016/j.stemcr.2019.04.005 (2019).
- 7 Kawaguchi, S. *et al.* Intramyocardial Transplantation of Human iPS Cell-Derived Cardiac Spheroids Improves Cardiac Function in Heart Failure Animals. *JACC Basic Transl Sci* **6**, 239-254, doi:10.1016/j.jacbts.2020.11.017 (2021).
- 8 Kawamura, T. *et al.* Cardiomyocytes Derived from MHC-Homozygous Induced Pluripotent Stem Cells Exhibit Reduced Allogeneic Immunogenicity in MHC-Matched Non-human Primates. *Stem cell reports* **6**, 312-320, doi:10.1016/j.stemcr.2016.01.012 (2016).
- 9 Zimmermann, W. H. *et al.* Engineered heart tissue grafts improve systolic and diastolic function in infarcted rat hearts. *Nature medicine* **12**, 452-458, doi:10.1038/nm1394 (2006).
- 10 Didie, M. *et al.* Parthenogenetic stem cells for tissue-engineered heart repair. *The Journal of clinical investigation* **123**, 1285-1298, doi:10.1172/JCI66854 (2013).
- 11 Marchiano, S. *et al.* Gene editing to prevent ventricular arrhythmias associated with cardiomyocyte cell therapy. *Cell Stem Cell* **30**, 396-414 e399, doi:10.1016/j.stem.2023.03.010 (2023).
- 12 Zimmermann, W. H. *et al.* Cardiac grafting of engineered heart tissue in syngenic rats. *Circulation* **106**, 1151-1157 (2002).
- 13 Riegler, J. *et al.* Human Engineered Heart Muscles Engraft and Survive Long Term in a Rodent Myocardial Infarction Model. *Circulation research* **117**, 720-730, doi:10.1161/CIRCRESAHA.115.306985 (2015).
- 14 Trede, F. *et al.* A refined panel of 42 microsatellite loci to universally genotype catarrhine primates. *Ecol Evol* **11**, 498-505, doi:10.1002/ece3.7069 (2021).
- 15 Barbian, H. J. *et al.* CHIIMP: An automated high-throughput microsatellite genotyping platform reveals greater allelic diversity in wild chimpanzees. *Ecol Evol* **8**, 7946-7963, doi:10.1002/ece3.4302 (2018).
- 16 Araki, K. *et al.* Autologous skeletal myoblast sheet implantation for pediatric dilated cardiomyopathy: A case report. *Gen Thorac Cardiovasc Surg* **69**, 859-861, doi:10.1007/s11748-020-01540-x (2021).
- 17 Kainuma, S. *et al.* Long-term outcomes of autologous skeletal myoblast cell-sheet transplantation for end-stage ischemic cardiomyopathy. *Molecular therapy : the journal of the American Society of Gene Therapy* **29**, 1425-1438, doi:10.1016/j.ymthe.2021.01.004 (2021).
- 18 Kim, J. Y., Nam, Y., Rim, Y. A. & Ju, J. H. Review of the Current Trends in Clinical Trials Involving Induced Pluripotent Stem Cells. *Stem Cell Rev Rep* **18**, 142-154, doi:10.1007/s12015-021-10262-3 (2022).

Point-by-Point Response (Jebran et al. 2023-03-04361B):

We thank referee #1 for the careful review of our revised manuscript and the kind acknowledgement that the first revision made our manuscript even more valuable. Please, find a point-by-point response to the critiques (**verbatim in bold**) below:

Referee #1 (Remarks to the Author):

The authors presented substantially new data, including the histological outcomes of human heart transplanted EHM, in the revised manuscript, making it even more valuable. They have also addressed some issues raised in the initial round of review, which were summarized in the point-by-point rebuttal; however, it brought forth additional concerns, and part of the issues remains unresolved.

We thank the reviewer for acknowledging that the addition of the human heart data from the BioVAT-HF-DHK20 (NCT04396899) Phase I/II clinical trial adds value to our study. Please note that this data only became available because a patient from the dose finding cohort of BioVAT-HF was subjected to heart transplantation. This provided us with the unique opportunity to study the patient heart and obtain first proof for remuscularization by cardiomyocyte implantation without safety issues in patients with advanced heart failure. Collectively, the procedure and findings support the use of BioVAT as bridge-to-transplantation.

1. The histological outcome from the human recipient of EHM is intriguing. The authors should provide detailed information.

A. Firstly, why did the patient receive a heart transplant? Didn't the transplantation of EHM provide sufficient effects?

Patients in the BioVAT-HF-DZHK20 Phase I/II Clinical Trial (NCT04396899) are in advanced heart failure despite guideline-directed palliative medical care. If they qualify, patients are listed for heart transplantation and are in case of successful donor heart allocation transplanted.

[Redacted text]

The reported patient with advanced ischemic heart failure had a stable disease trajectory after EHM implantation with attenuated left ventricular dilation and mildly enhanced left ventricular ejection fraction 3 months after EHM implantation. The patient was listed for heart transplantation and was transplanted according the study protocol.

We now include this information in **lines 256-257** and add a new **Extended Data Figure 9** with further information:

The patient demonstrated a stable disease course under EHM treatment (Extended Data Fig. 9).

Please note that BioVAT-HF is an ongoing registered clinical trial, which adheres strictly to an approved Clinical Trial Protocol (Synopsis includes as **Supplementary Note 5**). The full clinical trial results will be released and published after completion of the trial. Please note that the inclusion of the clinical case was approved by the sponsor and clinical trial statistician and that this particular

Point-by-Point Response (Jebran et al. 2023-03-04361B):

case was invaluable for the decision to increase the dose from 10x EHM (400 million iPSC-derived cardiomyocytes and stromal cells) to the maximal feasible dose according to the study protocol (20x EHM – 800 million iPSC-derived cardiomyocytes and stromal cells). The dose increase was endorsed by the independent Data Safety Monitoring Board. We now add this important information to our manuscript in **lines 261-268**:

Collectively, the obtained clinical data confirmed translatability of heart remuscularization by EHM allograft implantation from Rhesus macaque to human patients with advanced heart failure. It also established the rationale for continuation of patient treatment in the ongoing clinical trial with the maximal feasible dose according to the clinical trial protocol, i.e., 20x EHM constructed from 800 million iPSC-derived cardiomyocytes and stromal cells. Immune cell infiltration is commonly observed in heart transplant patients under guideline-directed immunosuppression¹ and will require further attention to improve outcome also in EHM-transplant patients.

[Redacted text]

B. Are there any immune responses to the grafted EHM? Please provide detailed immunohistological data.

Additional immunohistochemical analyses revealed T-cell, B-cell and macrophage infiltration in the human allograft (new **Extended Data Figure 10**), despite immune suppression at target tacrolimus concentrations (new **Extended Data Figure 9**). NK-cell infiltration was negligible. In contrast to the observations in Rhesus macaques, we did not find donor specific antibodies. Immune cell infiltration is commonly observed in heart transplant patients.¹ We cannot be certain whether the immune response was primarily directed against the allograft or the biodegradable TachoSil™ membrane, which we use as (1) a security measure to prevent possible epicardial bleeding, (2) to support targeted surgical administration, and (3) to reduce pericardial adhesions (refer to Kuschel et al. 2012).² We now add this new information on immune cell infiltration in the revised manuscript in **lines 256-263**:

T- and B-cells as well as macrophage (CD68) and minimal NK-cell (CD57) infiltrations were noted (**Extended Data Fig. 10**). Donor-specific antibodies (Luminex) were not identified. Collectively, these findings point to a local immune response against (1) the allograft, (2) the TachoSil™ support material or (3) both, despite immune suppression at high target levels (**Extended Data Fig. 9**).

C. Is the picture showing CD31 staining in the graft? Please provide pictures double-stained with a cardiac marker and CD31 that clearly show location of host and grafted CMs.

Yes, the picture is showing CD31 stained capillaries inside the EHM graft. The capillaries are in close proximity to the implanted cardiomyocytes, which can be identified by their distinct morphology (encircled).

Point-by-Point Response (Jebran et al. 2023-03-04361B):

We understand that this may not come across well in the provided pdf document. In response to the reviewer's critique, we have (1) exchanged the original CD31 IHC and (2) added an immunofluorescence co-staining for CD31, ACTN2 (marking cardiomyocyte in the EHM graft) and DNA; in addition, we have quantified capillary density in EHM graft and host heart see blow (**Figure 4E**):

The authors repeatedly claimed, “EHM grafts, very much in contrast to injected cardiomyocytes, can be clearly identified and distinguished from the recipient myocardium”.

Yes, this is the case even without genetic labels because of the clear epicardial localization of EHM grafts (refer to an example in **Figure 2b** or new **Figure 4b**). This was further confirmed by microsatellite analyses and deep-sequencing (**Extended Data Figures 8a** and **8c**).

D. In the same context, please provide pictures, perhaps with cardiac markers, showing the location of graft and host CMs. Also, it would be helpful to compare characteristics of CMs between host and graft CMs following human allogeneic transplantation.

Please refer to **Extended Data Figure 8b** (Rhesus macaque) and **8d** (human) in which we provide immunohistochemistries for TNNI1 (labels “immature” cardiomyocytes) and TNNI3 (labels “mature” cardiomyocytes), the ventricular and embryonic myosin light chain isoforms MYL2 and MYL4, as well as the intercalated disk protein cadherin 2 (CDH2) and gap junction protein connexin 43 (GJA1). Asterisks mark the host myocardium in the low power overviews (left panels).

Point-by-Point Response (Jebran et al. 2023-03-04361B):

In response to the reviewer's request, we now also provide a new **Figure 4d** with bar graphs summarizing graft vs host cardiomyocyte dimension.

E. Please provide the lengths of the three scale bars in figure 4A.

Lengths of scale bars are provided in the caption to revised Figure 4 (**line 533**)

2. The authors claimed that electrical coupling of EHM has been established in their previous publication as follows: "Electrical coupling and synchronicity of EHM allografts⁹, as well as injected cardiomyocyte allografts¹⁰ and xenografts¹, is a consistent observation." However, ref#9 did not provide direct evidence of electrical integration of EHM; instead, it presented indirect observations. The current manuscript does not demonstrate evidence of "mechano-electrical integration." I was surprised that they did not assess any paracrine effects.

We agree that the therapeutically relevant mechanism of integration remains elusive; this is in fact irrespective of the mode of cardiomyocyte implantation. In the discussion, we point to the possibility of mechanical entrainment, which we hypothesize to be an important component of the integration process. The phenomenon of mechanical communication without direct cell-cell contact has been demonstrated previously in an elegant study by Nitsan et al. 2016.³ In **lines 283-291**, we point out that a similar mechanisms may underlie EHM-host heart coupling and provide new data on mechanically triggered (new **Supplementary Video 3**) and conditioned (new **Extended Data Figure 11**) contractility. We also acknowledge that further studies are needed to resolve the mechanisms of cardiac patch-host heart synchronization:

Lines 283-291:

Tissue engineered patches, by virtue of their epicardial **engraftment**, cannot readily establish **canonical electromechanical connections via intercalated disks**, but appear to be mechanically entrained **over time** to contribute to myocardial performance. **This hypothesis is aligned with previous findings of mechanically induced cardiomyocyte contractility,³ observations of mechanically triggered contractions in EHM (Supplementary Video 3), and the finding that chronic mechanical conditioning (1 Hz for 120 h) leads to adaptations of EHM beating rate and rhythm (Extended Data Figure 11).** Extensions of these studies are required to clarify the time course, mechanism, and role of mechanical conditioning integration of EHM grafts.

The reviewer is referring to our previous publication,⁴ in which we conducted a high density epicardial mapping study in (1) spontaneous beating Langendorff perfused heart explants, (2) under electrical point stimulation and mapping of impulse propagation from graft to host and vice versa, and (3) by pH-shift uncoupled EHT grafts and remote myocardium. The observations from these studies were suggestive of electrical host-graft coupling 4 weeks after implantation of engineered heart tissue (EHT) in a rat allograft model. Unfortunately, we cannot perform similar studies in Rhesus macaques or human. More recently, iPSC-lines expressing genetically-encoded calcium-sensors^{5,6} and voltage tracer (RH237) infusion⁷ have been used to confirm graft-host coupling in intramural cardiomyocyte and epicardial EHT grafts. Rhesus iPSC-lines with genetically-encoded calcium-sensors are not available to us. In addition, genetically-encoded sensors may be immunogenic⁸ and thus were deemed not well-suited for pivotal (IND-enabling) preclinical studies.

Point-by-Point Response (Jebran et al. 2023-03-04361B):

We cannot exclude paracrine mechanisms and had studied such effects previously (implantation of tissue patches comprised of non-myocytes);⁴ in this study, “paracrine/non-contractile ECT patches” were inferior to contractile allografts.

[Redacted text]

In **lines 83-91** we are referring to alternative (paracrine, environmental modulation) mechanisms, but emphasize that our own previous data suggested better outcome in cardiomyocyte-based remuscularization. In the present non-human primate study, we focused fully on contractile EHM allo- and autografts. Testing of a “paracrine” support hypothesis would require an independent and differently designed monkey study, for which we cannot obtain approval under the strict animal protection regulations in Germany, because of the anticipated inferior outcome.

Finally, investigation of capillary density (bar graph in **Figure 4e**) and activated caspase 3 in close proximity to the EHM graft and in remote recipient myocardium (see response to reviewer #3 below) did not provide evidence for angiogenesis-inducing or anti-apoptotic paracrine effects at the 3- and 6-month study endpoints.

3. The response, "Irradiated EHM represent an upon implantation dying tissue graft, which we had also used in a previous rodent study¹³ as non-contractile/dead tissue controls," does not directly address the comment. In reference #13, the transplantation of irradiated EHM yielded functional benefits comparable to intact EHM, in contrast to the findings in the current study. This suggests distinct functional characteristics of irradiated EHM. Please address this discrepancy.

We appreciate this insightful comment and would like to point out that therapeutic outcomes in xenograft studies with human (such as in Riegler et al. 2015¹⁰) or Rhesus macaque (**Extended Data Figures 2**) EHM in nude rats are similar and best explained by mechanical stabilization, paracrine effects or modulation of inflammatory responses. Accordingly, we concluded in Riegler et al. 2015¹⁰ that the observed effects of viable and lethally irradiated EHM are “...consistent with the anticipated lack of electric integration of human xenografts in rat hearts, but highlights the possibility that cell-

Point-by-Point Response (Jebran et al. 2023-03-04361B):

independent effects (eg, activation of immune cells, mechanical stabilization) could also elicit therapeutic effects.”

In the present study, we implanted vital and lethally irradiated Rhesus macaque EHM in the same nude rat model with ischemia/reperfusion injury to assess “... feasibility of Rhesus EHM implantation in a widely used athymic nude rat model with ischemia/reperfusion (I/R)-injury.” (lines 115-116). The key observations were cell retention with no evidence for residual pluripotent stem cells (**Extended Data Figure 2a**). Statistical testing indicated enhanced cardiac function in rats after treatment with vital EHM in comparison to the pre-EHM implant baseline (BL) values (**Extended Data Figure 2b**); a similar, but not-significant trend (with the exception of an increased stroke volume at the 14-day time-point after irradiated EHM implantation) was noted in the rats treated with lethally irradiated EHM. We consider that these observations originate from mechanical stabilization, paracrine effects or modulation of inflammatory responses, similar as in Riegler et al. 2015.¹⁰

Please note that the nude rat xenograft study was a necessary first step towards the pivotal Rhesus macaque implant study to investigate whether Rhesus macaque EHM implantation would be similarly feasible and safe as human EHM implantation in the same model. The obtained data did indeed provide assurance that Rhesus macaque EHM can be safely administered. From an animal protection perspective, this was important before moving into the by the responsible regulatory authority in Germany requested homologous Rhesus macaque model (refer also to statement in lines 92-97).

4. The statement "We have also scored immune cell infiltration using clinical pathology standards" lacks specific information about the actual scores related to immune cell infiltration. It would be helpful if the provided details or scores on immune cell infiltration could be included for clarity.

The area covered by leukocyte infiltration (in mm²) was analyzed by an expert clinical pathologist blinded to the study protocol (refer to caption of **Extended Data Figure 4f**) in agreement with standard proceedings in clinical pathology. We have changed the labelling of the ordinate from “inflammation” to “leukocyte covered area” to improve clarity.

5. Please provide more reader-friendly figures. I will provide some of the examples:

A. Include information regarding the experimental group and/or antibodies for pictures in Extended Data Figure 7A, B, D, G.

Revised as suggested (please note that previous Extended Data Figure 7 is now Extended Data Figure 4).

B. In Extended Data Figure 7, along with the raw data, consider creating a figure or table that summarizes the results for better clarity.

Extended Data Figure 4 summarizes a comprehensive set of important histopathological data (**Panel c**: EHM patch retention, **Panel d**: cardiomyocyte retention, **Panel e**: cardiomyocyte population of engrafted patch, **Panel f**: leukocyte infiltration [inflammation]; **Panel h**: osteochondral cells) from the

Point-by-Point Response (Jebran et al. 2023-03-04361B):

three investigated Cohorts. For additional clarification images from H&E or desmin/actinin stained tissue sections are included in **Panel a, b, d, g**.

We are working with color-coded bars with differently sized borders and have included more detailed information as to the differently treated groups and hope that the reviewer finds that clarity has been improved (see blow from revised **Extended Data Figure 4**):

C. Present the aggregated ejection fraction (EF) figure shown in the rebuttal within the manuscript for better integration and understanding.

In response to the reviewer's request, we are now including aggregated Target Wall Thickening Fraction and Ejection Fraction data in **Figure 2d** and **2e**.

6. Extended Data Figure 9 was confusing. Please provide clarification by specifying which cells are graft cardiomyocytes and which are host cardiomyocytes. Additionally, despite the numerous CD56 positive cells observed, the claim of "no evidence for innate immune cell (NK-cells, macrophages) involvement" appears contradictory. Addressing this discrepancy would enhance the clarity of the presented data.

In response to the reviewer's critique, we have labelled the host cardiomyocytes in all overviews (left panels) with an asterisk and in addition indicate autograft and host myocardium in the revised **Extended Data Figure 5**. We now exclude the CD56 (NCAM1) labelling, because it obviously resulted in some confusion. CD56 (NCAM1) is a rather unspecific marker, which labels NK-cells but also immature cardiomyocytes. We had pointed this out in the first revision in the Extended Data Figure 9 legend (refer to images below with clarification highlighted in green – omitted in the revised **Extended Data Figure 5**):

Point-by-Point Response (Jebran et al. 2023-03-04361B):

Extended Data Figure 9: T-cell mediated autograft rejection. Immunohistochemical staining for T-cell (CD3, TCR α/β , TCR γ/δ), B-cell (CD20), NK-cell (CD56, CD57), and macrophage (CD68) with additional staining for cardiomyocytes (desmin) to determine the mode of EHM autograft rejection in experimental animals #2483 (Cohort 1). Note that CD56 (also known as neural cell adhesion molecule [NCAM] labels immature autograft cardiomyocytes); CD56 positive NK cells could not be identified. Desmin positive and CD56 negative cells resemble host heart cardiomyocytes (marked with an asterisk). Scale bars: 100 μ m.

7. The authors utilized cyclosporin in only one animal (#2915) and did not observe either graft survival or inflammation. The conclusion drawn, suggesting graft rejection due to donor-specific antibodies (DSA), may not accurately reflect the actual graft rejection, as this determination would require the examination of a larger number of animals under cyclosporin treatment. It is recommended to conduct the analysis on multiple animals to establish a more robust conclusion regarding "graft rejection" under the influence of cyclosporin.

We understand that the reviewer is asking for more confirmatory data, but we are unable to include additional macaques in the reported study. Cyclosporin plus Methylprednisolone treatment, as an alternative to Tacrolimus plus Methylprednisolone, was tested in one animal in response to a request by the relevant regulatory authority (Paul-Ehrlich-Institute). Although we only report one case, we do find that the observations are very informative as they reveal no cardiomyocyte survival at the 6-month study endpoint, which contrasts our observations in all other macaques treated continuously with tacrolimus and methylprednisolone (healthy Cohort 1+2: n=7 and infarcted cohort 3: n=6; **red bars in Extended Data Figure 4d**). Detection of donor-specific antibodies (DSA) from 12 weeks onwards (**Extended Data Figure 4i-l - #2915; Supplementary Data 3**) in parallel to the lowering of cyclosporin plasma levels to the target trough levels (140-250 ng/mL; **Supplementary Data 1**) is highly suggestive of a (sub)acute rejection episode setting in at 12 weeks with a subsequent clearing of the cardiomyocyte allograft. In such a case, residual inflammatory cells (leukocytes) would not be expected at the 24-week study endpoint (**Extended Data Figure 4f**).

Point-by-Point Response (Jebran et al. 2023-03-04361B):

See below a composite image with the indicated panels (striped red boxed are inserted to draw the reviewer's attention to the relevant findings in the cyclosporin + methylprednisolone treated animal #2915) – please note that we have converted the DSA data from the previous x/y graphs into heat maps (Extended Data Figure 4i-l) with accompanying raw data provided in an Excel spreadsheet (Supplementary Data 3):

Extended Data Figure 4d&f

Supplementary Data 1

Extended Data Figure 4 i-l and Supplementary Data 3

8. I am not certain if the equation for “cardiomyocyte volume” presented by the authors accurately represents the actual volume. Using the term "cardiomyocyte area" might be more accurate, considering that the parameters were obtained from 2-dimensional sections. This adjustment would better align with the nature of the data obtained.

We have reevaluated all data and now include cardiomyocyte area data as requested. The data is summarized in lines 235-238 of the revised manuscript:

“Engrafted cardiomyocytes were terminally differentiated (Ki67^{neg}) and remained smaller ($1,678 \pm 163 \mu\text{m}^2$; $n=13$ animals) than LV ($4,804 \pm 172 \mu\text{m}^2$) and RV ($3,685 \pm 226 \mu\text{m}^2$) cardiomyocytes of the recipient animals ($n=20$; Figure 3c).”

9. To enhance clarity, it would be beneficial to measure and present graft size in all recipients. This additional information would provide a more comprehensive understanding of the study's results.

Data on graft size is included in Extended Data Figure 4c (now relabeled as graft area in mm^2), 4d (cardiomyocyte area inside the engrafted patch in mm^2), and in 4e the ratio of cardiomyocyte area per patch area.

Point-by-Point Response (Jebran et al. 2023-03-04361B):

We thank referee #2 for the careful review of our manuscript and the kind acknowledgment that the first revision has significantly improved our manuscript. Please, find a point-by-point response to the critiques (**verbatim in bold**) below:

Referee #2 (Remarks to the Author):

The authors are responsive. And manuscript has been significantly improved. The findings are significant and important for the field. I have the following suggestion/comments for further improve the manuscript:

We thank the reviewer for acknowledging that our findings are significant and important for the field. In fact, our study was key for approving the first-in-patient BioVAT-HF clinical trial testing sustainable cardiac remuscularization by cardiomyocyte allografts under concomitant immune suppression. The BioVAT-HF clinical trial is progressing well and will be reported after its completion (2025/2026).

1. Although authors showed that engraftment of EHM is receipt animal hearts, almost all images were shown in relative high magnification. This makes it difficult to have an overall impression of how the EHM survived in the receipt hearts, representative images like right panel of Figure 4 A shall be shown for extended data Figure 9 and extended data figure 16A&16B)

Done as requested.

2. Authors heavily used immunohistochemistry to show the engraft and immune cell infiltration in engraft. Immunofluorescence staining shall be performed to show simultaneously the engraft and immune cell. This also applies to Figure 4B, immunofluorescence staining shall be performed to show the vessels in engraft.

Please note that our study was performed in alignment with regulatory expectations for IND-approval. Accordingly, we have embedded all samples in paraffin and performed immunohistochemistries according to standards in Clinical Pathology. Immunofluorescent images are less standardized, i.e., not validated for clinical pathology diagnostics, and thus less common in Clinical Pathology, especially in the identification of immune cell infiltrates.

In response to the reviewers critique we are now providing immunofluorescence images of CD31 positive capillaries in the human EHM allograft (**Figure 4e**).

3. It was reported by authors that RiPSC-CMs used for manufacturing EHM were more than 95% pure for ACTN2. ACTN2 is not a CM specific protein, and is expressed in skeletal muscle as well. Authors shall perform cTnT or cTnl staining to determine the purity of RiPSC-CM used for EHM.

Please find below flow cytometry data with co-labelling for sarcomeric actinin (ACTN2; Sigma A7811) and cardiac Troponin T (TNNT2; abcam AB45932) from a representative cardiomyocyte batch below. Please also note that the presence of skeletal muscle cells was excluded by snRNAseq (**Extended Data Figure 1**).

Point-by-Point Response (Jebran et al. 2023-03-04361B):

FMO = fluorescence minus one controls

In addition, we now provide the flow cytometry raw data (**Source Data** and revised text in **lines 102-108**) and would like to refer the reviewer to the snRNA-seq data in **Extended Data Figure 1** and further explanations in **Supplementary Note 1**.

Lines: 102-108

All applied Rhesus macaque iPSC-lines could be differentiated into cardiomyocytes and stromal cells with fibroblast properties (**Figure 1a**) at high purities (identified by flow cytometry: 92±2% ACTN2⁺ cardiomyocytes [n=7 batches optimized protocol]; 99% VIM⁺ stromal cells [n=2 batches]) using similar protocols established for human iPSC.¹¹ Purity was further confirmed by single nuclear RNA-sequencing (snSeq; **Extended Data Figs 1a-b** and **Supplementary Note 1**). In addition, snSeq (9,994 Rhesus macaque and 5,515 human nuclei) provided no evidence for residual pluripotent stem cells contaminations.

4. It is unclear the 95% purity of RiPSC-CMs was freshly differentiated RiPSC-CMs or after purification.

Purity assessment was after metabolic selection. Please refer to Methods section **lines 725-727**.

5. It was mentioned by authors that “we tested 4 different Rhesus macaque iPSC-lines, including 2 newly generated lines to also obtain insight as to in vivo autograft responses (Extended Data Table).” However, it is unclear which cell line was used in which animal and immunosuppression drugs. Did EHM manufacture from 3 RiPSC-CM have the similar structure and contractile performance?

Please refer to revised **Supplementary Table 1** for an overview of the investigated iPSC lines and their use in allo- and autograft preparations. We have used Rhesus iPSC 43110-4 for EHM allografts in

Point-by-Point Response (Jebran et al. 2023-03-04361B):

all animals in Cohorts 1 and 2 as well as in most animals in Cohort 3 (Rhesus iPSC DPZ_iRH34.1 was used in #2884 and #16441). We have included this information in revised **Supplementary Table 3**. We have also revised **Supplementary Table 2** to contain information on contractile performance of EHM manufactured from the different iPSC-lines. EHM contractility in DPZ_iRH34.1 and the two autograft lines was lower, with however no apparent differences in cell content and structure as well as similar outcome in #2884 (iPSC DPZ_iRH34.1) and #2819 (43110-4; **Figures 2d** and **2e**). The #2500 autograft line showed the lowest contractile performance despite similar cellularity. These data point to notable variability in iPSC-lines and contractile maturity of related EHM. These observations support the use of single well-defined iPSC-starting material (such as in BioVAT-HF) rather than individually prepared iPSC-lines.

Species	Human	Rhesus	Rhesus	Rhesus	Rhesus	
iPSC-line	TC1133	43110-4	DPZ_iRH34.1	#2483	#2500	
EHM sample number	16	12	10	4	4	
Spontaneous Beating Rate (bpm)	51±3	82±5	93±4	123±7	93±6	
@1.5 Hz electrical field stimulation	max. FOC (mN)	1.1±0.1	0.64±0.13	0.26±0.04	0.28±0.02	0.04±0.01
	Resting Tension (RT in mN)	0.6±0.08	0.36±0.05	0.65±0.07	0.63±0.11	0.46±0.09
	FOC/RT	2.2±0.3	1.8±0.3	0.44±0.07	0.48±0.08	0.09±0.04
	Contraction Time (to 90% in ms)	140±4	112±3	104±3	64±3	68±6
	Relaxation Time (to 50% in ms)	114±4	99±3	88±3	65±2	86±7

6. Authors heavily used desmin as an evidence of engraft in receipt animals hearts which is unusual (Figures 2B, 3C, 4B, extended data figure 7). Desmin is also expressed in smooth muscle cells and skeletal muscles, authors relied on desmin to show the host and donor CM, which is unconvincing. CM specific protein markers, such as cTnI or cTnI, shall be performed.

We provide high and low power magnifications as well as sarcomeric actinin and troponin I and myosin light chain 2 and 4 stains in **Extended Data Figure 4** as well as **Extended Data Figures 8a** and **8b**. As to the use of desmin, we would like to clarify that desmin IHC staining is a standard procedure for the assessment of muscle, including heart muscle, in clinical pathology. In the morphometric assessments, smooth muscle cells were excluded; skeletal muscle cells are not present in EHM grafts.

7. Extended data Figure 3, merged pic of cTnT and Rhesus Mitochondria shall be provided. Currently it is difficult to assess whether cTnT and Rhesus Mitochondria were colocalized.

Done as requested (refer to revised **Extended Data Figure 2**).

8. Can individual data be presented, such as scatter dot plot?

We took care to show most data in dot plot formats. The only data which we prefer to display as x/y-plots with averaged data points are in **Figure 2c, 2d** (controls), and **2e** (controls) as well as **Extended Data Figure 2d, 3c** and **7** to not overload the images. Please note that we are providing all Source Data with the revised submission.

Point-by-Point Response (Jebran et al. 2023-03-04361B):

9. Can authors discuss/speculate what benefit can be expected from such a small and thin engraftment (μm thickness) on a heart with cm thick left ventricular wall in human clinical trials?

The presented case is from the dose escalation cohort of the ongoing BioVAT-HF-DZHK20 Phase I/II Clinical Trial (NCT04396899). The data obtained in the Rhesus macaque study provided the basis for that start of the dose escalation with a 5x EHM in human (constructed from 200 million iPSC-derived cardiomyocytes and stromal cells). After confirmation of safety in 2 patients, we were allowed to increase the dose to 10x EHM (constructed from 400 million iPSC-derived cardiomyocytes and stromal cells). This dose increase did not increase graft thickness, but enlarged the graft area from approx. 50 cm^2 to 100 cm^2 (refer to schematic in **Figure 4a**). The thickness of such patches was 1-2 mm as can be observed in **Figure 4** and expected from our experience from the Rhesus macaque study.

According to allometric scaling (factor of 10 between Rhesus macaque and human), a 5-10x EHM in macaque resemble 0.5-1x EHM in human. Taken this in consideration, an enhancement of contractility may not have been expected. Given the high cardiomyocyte numbers implanted and the lack in knowledge as to safety of such implants it was prudent to start with the in the monkey confirmed safe maximal dose (5x EHM) and carefully accelerate to 10x EHM, before increasing to larger doses.

The findings from the presented case became only available because of a successful allocation of a heart transplant after listing for heart transplantation. The obtained data from the heart transplant was invaluable, because it demonstrated for the first time that human iPSC-derived cardiomyocytes would engraft and mature in a patient with advanced heart failure without safety concerns. After review of the data and endorsement by the independent Data Safety Monitoring Board (DSMB) rapid dose escalation to the as per clinical trial protocol maximal feasible dose of 20x EHM (constructed from 800 million iPSC-derived cardiomyocytes and stromal cells) was recommended.

[Redacted text and figure]

Point-by-Point Response (Jebran et al. 2023-03-04361B):

Note that the left ventricular wall thickness in human heart is 4-10 mm¹² and it may be anticipated that grafts of similar thickness can enhance cardiac contractility.

[Redacted text]

In **lines 256-268** and related Figures we are now providing additional data and context to help readers to place the in our view important first-in-patient observations into perspective:

Lines 256-268:

The patient demonstrated a stable disease course under EHM treatment (**Extended Data Fig. 9**). T- and B-cells as well as macrophage (CD68) and minimal NK-cell (CD57) infiltrations were noted (**Extended Data Fig. 10**). Donor-specific antibodies (Luminex) were not identified. Collectively, these findings point to a local immune response against (1) the allograft, (2) the TachoSil™ support material or (3) both, despite immune suppression at high target levels (**Extended Data Fig. 9**). Collectively, the obtained clinical data confirmed translatability of heart remuscularization by EHM allograft implantation from Rhesus macaque to human patients with advanced heart failure. It also established the rationale for continuation of patient treatment in the ongoing BioVAT-HF Phase I/II clinical trial with the maximal feasible dose according to the clinical trial protocol, i.e., 20x EHM constructed from 800 million iPSC-derived cardiomyocytes and stromal cells. Immune cell infiltration is commonly observed in heart transplant patients under guideline-directed immunosuppression¹ and will require further attention to improve outcome also in EHM-transplant patients.

Point-by-Point Response (Jebran et al. 2023-03-04361B):

We thank referee #4 for the careful review of our manuscript and pointing out that that our findings mark a significant advancement. Please, find a point-by-point response to the critiques (**verbatim in bold**) below:

Referee #4 (Remarks to the Author):

Jebran et al. conducted allogeneic transplantation experiments using Rhesus macaque-derived EHM under various conditions including transplant cell volume and immunosuppressive agent usage. In an optimized cohort, they demonstrated tissue regeneration with blood perfusion and cardiac functional recovery effects six months post-transplantation. Importantly, no significant adverse effects such as arrhythmias or tumor formation were observed, marking a significant advancement in this research area. However, reviewer believes that further detailed examination of the treatment mechanism and presentation of data is necessary for authors to assert that this therapeutic effect is directly due to remuscularization. Furthermore, authors obtained a rare opportunity in this paper to analyze heart samples from recipients of the BioVAT-HF trial, proving the potential of EHM to engraft as regenerated myocardium in human hearts. While this finding holds great significance in the study, additional evaluation of these samples is desirable.

Specific comments:

1. As the authors also comment in the text, in previous similar studies, the therapeutic effects have been suggested to be "at least partially mediated by immune responses or paracrine mechanisms" (line 86-87). In contrast, the authors suggest based on the results of experiments using irradiated EHM as a control group that the therapeutic effects in this study are mediated by "muscularization-independent mechanisms" (line 89). However, the reviewer believes that further detailed examination of this therapeutic mechanism is warranted. This is because, as noted by the authors themselves (citing #20-22), in previous studies on pluripotent stem cell-derived cardiomyocyte transplantation, at least some of the therapeutic effects have been attributed to paracrine indirect effects. Therefore, even if there were therapeutic effects due to remuscularization in the EHM transplantation in this study, there should have also been paracrine effects. It is unclear from the data presented what proportion of the therapeutic effects is attributed to each mechanism. At the very least, the authors should demonstrate what indirect effects, if any, were present in this EHM transplantation. For instance, if there were angiogenic effects in the ischemic heart, it is plausible that the degree of angiogenesis would primarily occur in the border zone of ischemia rather than within the EHM itself, but data regarding this aspect are not provided. How were cardiomyocyte apoptosis, likely induced by cardiac ischemia in the border zone, affected by EHM transplantation? If most of the therapeutic effects were due to indirect paracrine effects, the therapeutic effects of remuscularization would be limited, thus evaluating this aspect is crucial.

We are in full agreement with the reviewer that paracrine effects have to be anticipated in cell-based heart repair studies. We had previously performed allograft studies in rats in which we compared the effect of contractile EHM to controls implanted with non-cardiomyocyte containing grafts with paracrine activity.⁴ In these animals, attenuation of disease progression, but no recovery of target heart wall contractility was observed. In addition to the referred to original research paper, we have discussed this intensively in the past.⁹

[Redacted text]

Point-by-Point Response (Jebran et al. 2023-03-04361B):

[Redacted figure]

Due to the nature of the Rhesus macaque model and the study design with 3- and 6-months follow-up, assessment of apoptosis and vascularization are limited to these time points. Here, we did not find any evidence for Caspase 3 activity (as a marker of apoptosis) in engrafted and host cardiomyocytes; refer to IHC for activated Caspase 3 below).

We also did not find any evidence for enhanced vascularization in the host myocardium including myocardium in close proximity to the EHM graft (refer to **lines 250-255** and new **Figure 4e**):

Point-by-Point Response (Jebran et al. 2023-03-04361B):

Lines 250-255:

Histological analyses confirmed a similar relative immaturity as observed in the Rhesus macaque model (**Extended Data Fig. 8b**) and lower capillary density ($187 \pm 5/\text{mm}^2$) in EHM graft compared to the recipient heart ($963 \pm 12/\text{mm}^2$; $n=3$ regions of interest analyzed; **Figure 4e**). No differences in capillary densities in remote myocardium and in close proximity to the EHM suggest that angiogenic paracrine effects are locally restricted to the EHM.

Figure 4e:

Despite these observations, we do not rule out that EHM implants exhibit multimodal therapeutic activity. A completely different study design would be required to in detail tease out paracrine vs remuscularization effects. In light of our previous and present data as well as the targeting of advance heart failure in patients without hibernating myocardium paracrine effects are less likely to have contributed to the observed effects. In addition, obtaining approval for a pivotal preclinical Rhesus macaque study, in which according to our own preliminary data limited therapeutic efficacy would be anticipated, is difficult to impossible. Having said this, we do not rule out that there are beneficial effects of paracrine therapeutics, which may in specific disease settings even be superior or preferable to remuscularization strategies; for example, in patients with hibernating myocardium. Clinical trials will have to give a definitive answer as to whether, when and how heart repair or protection or both would be advantageous by remuscularization or paracrine activity.

[Redacted text and figure]

2. Understanding how EHM provides mechanical contractile force to the host heart post-transplantation is a crucial point. While the authors suggest the presence of a mechano-electrically coordinated manner (line 278-279), data supporting this assertion seem to be lacking. It remains unclear if there are any mechano-sensors present in the EHM and the host heart tissue at the transplantation site. If so, as indicated in Fig. 1D, since this EHM exhibits automaticity (self-beating without external stimuli), if mechano-electrically coordinated manner is indeed the mechanical coupling pattern between EHM and the host heart, wouldn't it potentially induce ectopic contractions in the host heart immediately after transplantation, leading to arrhythmias? Why then, in actuality, did arrhythmias not occur even in responders where EHM engraftment was successful? The authors should address this point to provide clarity.

We understand the reviewers point and would like to clarify as follows: (1) mechanical communication independent of electrical cardiomyocyte:cardiomyocyte coupling has been reported and was suggested to contribute to the synchronous pumping activity of the heart³ (in the revised manuscript, we now refer to this excellent study by Nitsan et al.), (2) an epicardial EHM graft cannot immediately couple electrically to the host heart, (3) EHM exhibit a slower spontaneous beating rate than the host myocardium leading to “mechanical” overstimulation after engraftment, (4) EHM can be triggered to beat by mechanical impulses (new **Supplementary Video 3**), and (5) cyclic mechanical conditioning has an impact on spontaneous beating rate and rhythm of EHM (new **Extended Data Figure 11**). Although mechano-sensors are expressed in EHM similar to what can be observed in the normal heart (s. below from our RNAseq data - additional stretch-activated channels which are strongly expressed (FPKM >10) in EHM and heart include: TRPC1, TRPM4, TRMP7, TRPV2, KCNJ8), it is quite unclear whether and how classical mechano-sensors contribute to heart rate and rhythm (refer also to Nitsan et al.³).

Finally, there is a clear need for extended studies to resolve coupling mechanisms of cardiomyocyte grafts in general. It is well established that cardiomyocyte grafts can integrate into the functional syncytium of the heart (refer for example to our previous mouse allograft study in which we injected GFP labeled mouse cardiomyocytes into the mouse heart and performed 2P imaging to determine 1:1 coupling – Didié et al. 2013¹³):

Point-by-Point Response (Jebran et al. 2023-03-04361B):

Legend from Figure 7 in Didié et al. 2013¹³: Retention and functional integration of PCMs after intramyocardial injection. (A and B) Immunofluorescent labeling of α -actinin (red, A) and connexin43 (red, B) in adult ventricular mouse heart tissue 3 weeks after injection of PCMs (EGFP, green; nuclei, blue). (C–F) Two-photon laser scanning microscopy of intracellular Ca^{2+} transients in adult mouse hearts after injection of PCMs: 2D-scan (C) and line-scan (D) images of stimulated (3 Hz) and spontaneous Ca^{2+} transients. Arrow 1, EGFP-positive cell; arrow 2, EGFP negative cell; the dotted line indicates the location of the line scan. Bands of increased rhod-2 fluorescence intensity reflect AP-induced Ca^{2+} transients. (E) Plots of rhod-2 and GFP line-scan data in the EGFP expressing cardiomyocyte 1 and the GFP-negative (native) cardiomyocyte 2 as a function of time. (F) Superimposed tracings of AP-evoked changes in rhod-2 fluorescence as a function of time from cardiomyocytes 1 (green) and 2 (red). For each cell, the relative changes in fluorescence were normalized such that 0 represents the prestimulus fluorescence intensity (F_0), and 1 represents the peak fluorescence intensity.

More indirect evidence for EHM:heart coupling stems from previous rat study (Zimmermann et al. 2006⁴), in which we conducted a high density epicardial mapping study in spontaneous beating Langendorff-perfused heart explants (panel a [without EHT] and b [with EHT] as well as panels c/d white bars [without EHM] and black bars [with EHM]) showing no delay in epicardial impulse propagation in EHM treated hearts as well as retrograde electrical activation after stimulation of the EHT graft (panel e) and uncoupling of EHT under acidification (panel f). Collectively, these data were suggestive of electrical host-graft coupling 4 weeks after implantation of engineered heart tissue (EHT) in a rat allograft model.

Point-by-Point Response (Jebran et al. 2023-03-04361B):

Legend from Figure 3 in Zimmermann et al. 2006⁴: Electrical integration of EHTs in vivo. Representative plots of epicardial activation times in sham-operated (a) and EHT-engrafted (b) hearts. Total activation time (c) and QRS-complex voltage (d) in right, anterior, lateral and posterior segments of the investigated hearts. (e) Point stimulation of an implanted EHT with simultaneous recording of the propagated potential in the EHT and in remote myocardium showed retrograde coupling. EHT could be uncoupled after acidification of the hearts (f).

Unfortunately, such studies cannot be performed in Rhesus macaques or human. More recently, iPSC-lines expressing genetically-encoded calcium-sensors^{5,6} and voltage tracer (RH237) infusion⁷ have been used to confirm graft-host coupling in intramural cardiomyocyte and epicardial EHT grafts. Rhesus iPSC-lines with genetically-encoded calcium-sensors are not available to us. In addition, genetically-encoded sensors may be immunogenic⁸ and thus were deemed not well-suited for pivotal (IND-enabling) preclinical studies.

Although we find a mechano-electrical coupling / conditioning mechanism likely, we agree that further studies are needed and have accordingly edited **lines 284-292** of the revised manuscript:

“Tissue engineered patches, by virtue of their epicardial **engraftment**, cannot readily establish **canonical** electromechanical connections **via intercalated disks**, but appear to be mechanically entrained **over time** to contribute to myocardial performance. This hypothesis is aligned with previous findings of mechanically induced cardiomyocyte contractility,³ observations of mechanically triggered contractions in EHM (**Supplementary Video 3**), and the finding that chronic mechanical conditioning (1 Hz for 120 h) leads to adaptations of EHM beating rate and rhythm (**Extended Data Figure 11**). Extensions of these studies are required to clarify the time course, mechanism, and role of mechanical conditioning for integration of EHM grafts.”

Point-by-Point Response (Jebran et al. 2023-03-04361B):

As to ectopic contractions induced arrhythmia, we can confirm that there is no evidence for EHM graft induced arrhythmia from any animal model studied (mice, rats,

3. The patient samples transitioning to heart transplantation in the BioVAT-HF trial could provide invaluable data. However, it is conceivable that these samples were targeted for heart transplantation precisely because they were non-responders in the clinical trial itself. In other words, this heart might be considered a sample that didn't achieve the desired therapeutic effects adequately. Was the engraftment, particularly in this human heart sample, insufficient compared to responders in the Rhesus macaque allograft transplantation performed in this study? If adequate engraftment was indeed achieved, why couldn't sufficient therapeutic effects be attained? Could it be attributed to factors such as the extent of preoperative cardiac impairment? Furthermore, did arrhythmias not occur post-transplantation in this patient? While acknowledging that data obtained from this sample should include information that ought to be confidential within the BioVAT-HF trial, it is crucial for the research to demonstrate and discuss the evidence of therapeutic efficacy in humans regarding EHM allograft transplantation. This is important not only for advancing this treatment approach clinically but also for the research itself. At least, it is deemed necessary to ensure that the results in Rhesus macaque allograft transplantation are not contradictory to the data regarding therapeutic efficacy and safety in this human case.

The Rhesus macaque data are in agreement with the clinical data from BioVAT-HF, which we cannot include fully in the present study (study is ongoing with reporting regulated according to the clinical trial protocol). The patient in our manuscript was implanted with 10x EHM. In terms of allometric scaling (factor of 10), 10xEHM in human resemble 1x EHM in the macaque model. 10x EHM were safe, but did not reverse the advanced heart failure phenotype (refer to new **Extended Data Figure 9**). The young patient was listed for heart transplantation and was upon donor organ allocation transplanted. In fact, in this patient EHM served as a bridge-to-transplant alternative to a left ventricular assist device.

The heart transplant provided us with invaluable data, demonstrating for the first time that iPSC-derived cardiomyocyte allografts are retained with no safety concerns. An unanticipated observation was T- and B-cell infiltration despite target level tacrolimus concentrations (new **Extended Data Figure 10**). Donor specific antibodies (DSA) were not detected and we cannot be certain whether the inflammatory response was directed against the allograft, the TachoSil™ membrane or both. Collectively, the obtained data provided a solid basis for an accelerated dose escalation to the as per clinical trial protocol designated maximal feasible dose (20x EHM constructed from 800 million iPSC derived cardiomyocytes and stromal cells) under careful immune monitoring.

We now provide addition information on the patient, which was successfully heart transplanted (lines 256-268, new **Extended Data Figures 9 and 10**):

Lines 256-268:

The patient demonstrated a stable disease course under EHM treatment (**Extended Data Fig. 9**). T- and B-cells as well as macrophage (CD68) and minimal NK-cell (CD57) infiltrations were noted (**Extended Data Fig. 10**). Donor-specific antibodies (Luminex) were not identified. Collectively, these findings point to a local immune response against (1) the allograft, (2) the TachoSil™ support material or (3)

Point-by-Point Response (Jebran et al. 2023-03-04361B):

both, despite immune suppression at high target levels (**Extended Data Fig. 9**). Collectively, the obtained clinical data confirmed translatability of heart remuscularization by EHM allograft implantation from Rhesus macaque to human patients with advanced heart failure. It also established the rationale for continuation of patient treatment in the ongoing BioVAT-HF Phase I/II clinical trial with the maximal feasible dose according to the clinical trial protocol, i.e., 20x EHM constructed from 800 million iPSC-derived cardiomyocytes and stromal cells. Immune cell infiltration is commonly observed in heart transplant patients under guideline-directed immunosuppression¹ and will require further attention to improve outcome also in EHM-transplant patients.

We never observed EHM-induced arrhythmia in animal models or in patients treated in BioVAT-HF.

[Redacted text]

Point-by-Point Response (Jebran et al. 2023-03-04361B):

References:

- 1 Beniaminovitz, A. *et al.* Prevention of rejection in cardiac transplantation by blockade of the interleukin-2 receptor with a monoclonal antibody. *N Engl J Med* **342**, 613-619, doi:10.1056/NEJM200003023420902 (2000).
- 2 Kuschel, T. J. *et al.* Prevention of postoperative pericardial adhesions with TachoSil. *Ann Thorac Surg* **95**, 183-188, doi:10.1016/j.athoracsur.2012.08.057 (2013).
- 3 Nitsan, I., Drori, S., Lewis, Y. E., Cohen, S. & Tzlil, S. Mechanical communication in cardiac cell synchronized beating. *Nat Phys* **12**, 472+, doi:10.1038/Nphys3619 (2016).
- 4 Zimmermann, W. H. *et al.* Engineered heart tissue grafts improve systolic and diastolic function in infarcted rat hearts. *Nature medicine* **12**, 452-458, doi:10.1038/nm1394 (2006).
- 5 Shiba, Y. *et al.* Allogeneic transplantation of iPS cell-derived cardiomyocytes regenerates primate hearts. *Nature* **538**, 388-391, doi:10.1038/nature19815 (2016).
- 6 Chong, J. J. *et al.* Human embryonic-stem-cell-derived cardiomyocytes regenerate non-human primate hearts. *Nature* **510**, 273-277, doi:10.1038/nature13233 (2014).
- 7 Weinberger, F. *et al.* Cardiac repair in guinea pigs with human engineered heart tissue from induced pluripotent stem cells. *Science translational medicine* **8**, 363ra148, doi:10.1126/scitranslmed.aaf8781 (2016).
- 8 Chang Liao, M. L. *et al.* Sensing Cardiac Electrical Activity With a Cardiac Myocyte--Targeted Optogenetic Voltage Indicator. *Circulation research* **117**, 401-412, doi:10.1161/CIRCRESAHA.117.306143 (2015).
- 9 Fujita, B. & Zimmermann, W. H. Engineered Heart Repair. *Clin Pharmacol Ther* **102**, 197-199, doi:10.1002/cpt.724 (2017).
- 10 Riegler, J. *et al.* Human Engineered Heart Muscles Engraft and Survive Long Term in a Rodent Myocardial Infarction Model. *Circulation research* **117**, 720-730, doi:10.1161/CIRCRESAHA.115.306985 (2015).
- 11 Tiburcy, M. *et al.* Defined Engineered Human Myocardium With Advanced Maturation for Applications in Heart Failure Modeling and Repair. *Circulation* **135**, 1832-1847, doi:10.1161/CIRCULATIONAHA.116.024145 (2017).
- 12 Kawel, N. *et al.* Normal left ventricular myocardial thickness for middle-aged and older subjects with steady-state free precession cardiac magnetic resonance: the multi-ethnic study of atherosclerosis. *Circ Cardiovasc Imaging* **5**, 500-508, doi:10.1161/CIRCIMAGING.112.973560 (2012).
- 13 Didie, M. *et al.* Parthenogenetic stem cells for tissue-engineered heart repair. *J Clin Invest* **123**, 1285-1298, doi:10.1172/JCI66854 (2013).

Point-by-Point Response (Jebran et al. 2023-03-04361C):

We thank the referees for their careful review of our revised manuscript. The revisions have helped to improve our manuscript.

Please, find our response to the comment (**verbatim in bold**) by reviewer #1:

Referee #1 (Remarks to the Author):

The authors have addressed most of the comments. However, I have an additional comment regarding the new Supplementary Video 3, which is quite intriguing. After physical stimulation of one side of the engineered heart muscle (EHM), the other side contracts. However, the interval between stimulation and contraction is approximately 2 seconds, which is too slow to synchronize with host beating hearts. Additionally, the propagation of contraction within the EHM does not appear sequential, which is unexpected. Please provide an explanation for this.

We are delighted to have addressed most of the reviewer's comments and thank the reviewer for acknowledging that the new Supplementary Video 3 provides intriguing information. In the two demonstrated instances of mechanical stimulation-induced contractions in ring-shaped EHM 1, we see an immediate contractile response of EHM 1, demonstrating that EHM sense and react to mechanical stimuli. Ring-shaped EHM 2 is beating spontaneously and, as expected, not affected by the mechanical impulse to EHM 1. Please note that the experiment was performed at room temperature, which explains the low spontaneous beating rate and slow contraction kinetics compared to what we observe at 37 °C (refer to human EHM data in Supplementary Table 2). We have added this information to the Supplementary Video 3 Legend:

Supplementary Video 3: Mechanically triggered contraction in human EHM. Ring-shaped human EHM 1 (**mechanically stimulated**) and EHM 2 (**spontaneously contracting/not mechanically stimulated**) suspended on flexible poles of an EHM patch holding device. **Recordings were performed at room temperature.**

Referee #2 (Remarks to the Author):

The authors are very responsive. The responses are satisfactory. The manuscript is significantly improved. The findings are significant and novel. The data are solid.

We thank referee #2 for the valuable support and the kind acknowledgement that our findings are significant and novel.

Referee #4 (Remarks to the Author):

The reviewer believes that the authors have sufficiently addressed the reviewers' concerns. The reviewer also understands that a detailed evaluation of the effects of paracrine factors is challenging within the study design using Rhesus macaques. The additional data provided by the authors can serve as a valuable contribution for ongoing discussions within the scientific

Point-by-Point Response (Jebran et al. 2023-03-04361C):

community on this point. Furthermore, the reviewer agrees that continued investigation into electrical and mechanical coupling is necessary. The reviewer also appreciates the effort to present valuable human sample data derived from the BioVAT-HF trial to the extent possible.

We thank referee #4 for the kind support and acknowledgement that our data will serve as a valuable contribution to the field. We fully agree that despite of the presented evidence for mechanical conditioning of EHM (Extended Data Figure 11) further studies are required to identify the precise mechanism and time course of mechanoelectrical integration / synchronization after implantation.